# Risk-Sensitive Diffusion: Robustly Optimizing Diffusion Models with Noisy Samples

**Yangming Li, Max Ruiz Luyten, Mihaela van der Schaar**
Department of Applied Mathematics and Theoretical Physics
University of Cambridge
`yl874@cam.ac.uk`

## Abstract

Diffusion models are mainly studied on image data. However, non-image data (e.g., tabular data) are also prevalent in real applications and tend to be noisy due to some inevitable factors in the stage of data collection, degrading the generation quality of diffusion models. In this paper, we consider a novel problem setting where every collected sample is paired with a vector indicating the data quality: *risk vector*. This setting applies to many scenarios involving noisy data and we propose *risk-sensitive SDE*, a type of stochastic differential equation (SDE) parameterized by the risk vector, to address it. With some proper coefficients, risk-sensitive SDE can minimize the negative effect of noisy samples on the optimization of diffusion models. We conduct systematic studies for both Gaussian and non-Gaussian noise distributions, providing analytical forms of risk-sensitive SDE. To verify the effectiveness of our method, we have conducted extensive experiments on multiple tabular and time-series datasets, showing that risk-sensitive SDE permits a robust optimization of diffusion models with noisy samples and significantly outperforms previous baselines.

## 1 Introduction

**Prevalence of noisy non-image data.** Current studies on diffusion models (Sohl-Dickstein et al., 2015; Ho et al., 2020) (or score-based generative models (Song & Ermon, 2019; Song et al., 2021)) have primarily focused on high-quality image data, achieving promising performance (Dhariwal & Nichol, 2021) in image synthesis. However, non-image data (e.g., tabular data and time series) are in fact more popular in real applications (e.g., medicine (Johnson et al., 2016) and finance (Takahashi et al., 2019)). A survey conducted by Kaggle (Kaggle, 2017; van Breugel et al., 2023) revealed that 79% of the data scientists are mainly working on tabular data. Importantly, while image datasets are commonly of high quality, non-image data contain noisy samples in most cases. For example, sensor data are susceptible to measurement errors (Steinvall & Chevalier, 2005), and such noise can significantly degrade the performance of diffusion models.

**Introduction of risk vectors.** In this work, we are interested in a novel problem setup where every sample in the dataset is associated with a vector indicating the sample quality: *risk vector*. The purpose of setting this vector is to provide information that a potential method can use to robustly optimize diffusion models in the presence of noisy samples. While this setup might seem artificial, it applies to many real scenarios involving noisy data. For example, tabular data often contain missing values (Barnard & Meng, 1999), and practitioners typically impute those values before using the data. During this preprocessing step, many imputation methods can provide confidence values (i.e., risk information) for their predictions. Even when such risk vectors are not directly accessible, a class of methods known as uncertainty quantification (Angelopoulos & Bates, 2021) can offer viable alternatives. In Appendix D, we provide a detailed discussion and more real-world examples, showing the broad applicability of our proposed setup.

**Principled method: risk-sensitive diffusion.** To address the problem setup: noisy samples paired with risk vectors, we first study the negative impact of noisy samples on optimizing diffusion models, with a conclusion that such samples mainly cause a marginal distribution shift in the diffusion

process. In light of this finding, we introduce an error measure called *perturbation instability*, which quantifies the negative effect of noisy samples, and propose *risk-sensitive SDE*, a type of stochastic differential equation (SDE) parameterized by the risk vector, with the aim to minimize the instability measure. For both Gaussian and general non-Gaussian noise perturbation, we determine the optimal coefficients of risk-sensitive SDE, and prove that, in the case of isotropic Gaussian noises, that type of negative impact can be fully eliminated. In experiments, we show that our method is still very effective in the scenario with non-Gaussian (e.g., Cauchy) noises.

**Contributions.** In summary, the contributions of this paper are as follows:

- Conceptually, we are the first to introduce *risk vectors* to robustly optimize diffusion models with noisy samples, with a principled method: risk-sensitive SDE, to incorporate such a vector, reducing the negative impact of noisy samples: *perturbation instability*;
- Technically, we solve the analytical forms of *risk-sensitive SDE* for both Gaussian and non-Gaussian noise distributions, with a notable conclusion that, in the case of Gaussian perturbation, the negative impact of noisy samples can be fully reduced;
- Empirically, experiment results on multiple tabular and time-series datasets show that risk-sensitive SDE can effectively handle noisy samples, even when the noise distribution is mis-specified or non-Gaussian, and notably outperform previous baselines.

We have publicly released the code at https://github.com/LeePleased/rdm.

## 2 PRELIMINARIES

In this section, we first briefly introduce the background of diffusion models, with basic terminologies and notations that will also be used later. Then, we present the motivation and formulation of our problem setup: noisy samples paired with a *risk vector*.

### 2.1 BACKGROUND OF DIFFUSION MODELS

While diffusion modes (or score-based generative models) have different versions and variants, we adopt the formulation of Song et al. (2021), which generalizes DDPM (Ho et al., 2020), SMLD (Song & Ermon, 2019), VDM (Dhariwal & Nichol, 2021), etc.

At the core of diffusion models lies a *diffusion process*, which drives data samples $\mathbf{x}(0) \sim p_0(\mathbf{x}(0))$ (i.e., a finite-dimensional vector) towards noise $\mathbf{x}(T) \sim p_T(\mathbf{x}(T))$ at time $T \in \mathbb{R}^+$, and can be expressed through a stochastic differential equation (SDE) (Itô, 1944):

$$d\mathbf{x}(t) = f(t)\mathbf{x}(t)dt + g(t)d\mathbf{w}(t), \tag{1}$$

where $\mathbf{w}(t)$ is a standard Wiener process, $f(t)\mathbf{x}(t) : \mathbb{R} \times \mathbb{R}^d \to \mathbb{R}^d$ is a predefined vector-valued function that specifies the drift coefficient, and $g(t) : \mathbb{R} \to \mathbb{R}$ is a predetermined scalar-valued function that specifies the diffusion coefficient. We call $p_T, T \to \infty$ the prior distribution, which is fixed and retains no information of $p_0$ via a proper design of coefficients $f(t), g(t)$.

Interestingly, the *reverse process* (i.e., reverse version of the diffusion process) also follows an SDE. For a process of the form as Eq. (1), it shapes as:

$$d\mathbf{x}(t) = \big(f(t)\mathbf{x}(t) - g(t)^2 \nabla_{\mathbf{x}(t)} \ln p_t(\mathbf{x}(t))\big)dt + g(t)d\bar{\mathbf{w}}(t), \tag{2}$$

which runs another standard Wiener process $\bar{\mathbf{w}}(t)$ backward in time. For generative purposes, we can sample randomly from the prior distribution $p_T$ and use the reverse process to map such samples into data samples that will follow $p_0$, that is, the distribution of inputs. The challenge is to determine the expression $\nabla_{\mathbf{x}} \ln p_t(\mathbf{x})$ in the backward process (known as the score function) since the term is analytically intractable in most cases. A common practice (Song & Ermon, 2019) is to use an approximation called the score-based model $\mathbf{s}_{\boldsymbol{\theta}}(\mathbf{x}, t)$, for instance, a neural network.

To optimize the score model towards the score function, previous works (Song & Ermon, 2019; Song et al., 2021) derived the following score-matching loss:

$$\mathcal{L} = \mathbb{E}_{(t, \mathbf{x}_0, \mathbf{x}_t)} \big[\lambda(t) \|\mathbf{s}_{\boldsymbol{\theta}}(\mathbf{x}(t), t) - \nabla_{\mathbf{x}(t)} \ln p_t(\mathbf{x}(t))\|_2^2\big], \tag{3}$$

where the weight $\lambda(t) : [0, T] \to \mathbb{R}^+$ is generally set uniformly. Importantly, it is common (Song et al., 2021) to adopt an upper bound of the above loss to fit the model $\mathbf{s}_{\boldsymbol{\theta}}(\mathbf{x}(t), t)$ with the kernel $p_{t|0}(\mathbf{x}(t) \mid \mathbf{x}(0))$ (i.e., the density of $\mathbf{x}(t)$ conditioning on $\mathbf{x}(0)$).

## 2.2 PROBLEM SETUP

For standard diffusion models, the observed sample $\mathbf{x}(0) \in \mathbb{R}^D$ is implicitly assumed to be without *noise perturbation*. However, this simplification does not apply to many real applications. Fig. 1 shows an example of medical time series, which consists of irregularly spaced observations. To apply diffusion models to such data, one will first fill in the missing values with some interpolation method (Rubanova et al., 2019), resulting in noises in the form of interpolation errors.

**Misguidance effect of Noisy Samples.** Noisy sample $\widetilde{\mathbf{x}}(0)$ intuitively has a negative impact on the optimization of score-based model $\mathbf{s}_{\boldsymbol{\theta}}(\mathbf{x}, t)$, degrading the generation quality of diffusion models. For this point, a solid explanation is as below.

*Remark* 2.1. In the standard case with only clean sample $\mathbf{x}(0)$, the score-based model $\mathbf{s}_{\boldsymbol{\theta}}(\mathbf{x}, t)$ is optimized to match the score function $\nabla_{\mathbf{x}} \ln p_t(\mathbf{x})$. Since noisy sample $\widetilde{\mathbf{x}}(0)$ has a different initial distribution $\widetilde{p}_0(\mathbf{x})$ from that $p_0(\mathbf{x})$ of clean sample $\mathbf{x}(0)$, their marginal distributions $\widetilde{p}_t(\mathbf{x}), p_t(\mathbf{x})$ at time step $t$ will also be different, with the same diffusion process (i.e., Eq. (1)). As a result, we have $\nabla_{\mathbf{x}} \ln \widetilde{p}_t(\mathbf{x}) \neq \nabla_{\mathbf{x}} \ln p_t(\mathbf{x})$, indicating that noisy sample $\widetilde{\mathbf{x}}(0)$ causes a wrong objective $\nabla_{\mathbf{x}} \ln \widetilde{p}_t(\mathbf{x})$ for optimizing the model $\mathbf{s}_{\boldsymbol{\theta}}(\mathbf{x}(t), t)$.

In short, we can say that noisy sample $\widetilde{\mathbf{x}}(0)$ misleads the diffusion models in training.

**Introducing risk information.** Although noisy samples are inescapable in some situations, they are usually with additional information, estimating the potential risk of using such samples. Following the previous example, a Gaussian process (MacKay et al., 1998) that interpolates the missing samples in Fig. 1 naturally provides uncertainty information (i.e., confidence intervals) for each prediction. We can thus pair every possibly noisy sample $\widetilde{\mathbf{x}}(0) = [\widetilde{x}_1(0), \widetilde{x}_2(0), \cdots, \widetilde{x}_D(0)]^\top$ with its risk information $\mathbf{r}$, available for free. While risk $\mathbf{r}$ is defined in a very general way in Definition 3.1, its concrete form depends on the noise type. For example, in the case of *non-isotropic Gaussian perturbation*, the risk $\mathbf{r}$ is a vector $\mathbf{r} = [r_1(0), r_2(0), \cdots, r_D(0)]^\top$ of the same $D$ dimensions as the sample $\widetilde{\mathbf{x}}(0)$, indicating its entry-wise data quality. The closer to 0 the value in each entry $r_i \in \mathbb{R}^+ \bigcup \{0\}$ of $\mathbf{r}$ is, the higher the expected quality, where $r_i = 0$ indicates that entry $\widetilde{x}_i(0)$ is clean.

Provided with the risk vector $\mathbf{r}$, an ideal generative model could draw information from both risky ($\widetilde{\mathbf{x}}(0), \mathbf{r} \neq \mathbf{0}$) and clean samples ($\widetilde{\mathbf{x}}(0) = \mathbf{x}(0), \mathbf{r} = \mathbf{0}$), and importantly, this model was only optimized towards the distribution of clean samples: $p_0(\mathbf{x})$.

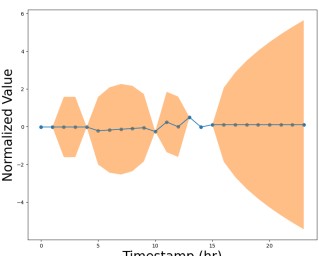

Figure 1: A segment of noisy time series from MIMIC (Johnson et al., 2016). The data points outside the orange region (i.e., 95% confidence intervals) are observed, and a Gaussian process interpolates the ones within the area.

## 3 METHOD: RISK-SENSITIVE DIFFUSION

To reduce the misguidance of noisy samples on optimizing diffusion models, we present a principled method: *risk-sensitive SDE*, a type of SDE parameterized by risk vector $\mathbf{r}$. In the following, we first define risk-sensitive SDE and some other useful concepts, and then solve its optimal coefficients for different noise perturbations. Finally, we present the training and inference algorithms for diffusion models under the framework of risk-sensitive SDE,

## 3.1 BASIC DEFINITIONS

**Risk vectors and noise distribution families.** Intuitively, risk information $\mathbf{r}$ represents the data quality of a noisy sample $\widetilde{\mathbf{x}}(0)$. To formalize this concept in a more rigorous way, we provide the below definition, which still aligns with the intuition.

**Definition 3.1** (Risk Vectors). The risk information $\mathbf{r}$ shapes as a vector that is element-wise non-negative and controls a family of continuous noise distributions:

$$\mathcal{P}_{\epsilon} = \left\{ \rho_{\mathbf{r}}(\boldsymbol{\epsilon}) : \mathbb{R}^D \to \mathbb{R}_+, \int \rho_{\mathbf{r}}(\boldsymbol{\epsilon}) d\boldsymbol{\epsilon} = 1 \,\Big|\, \mathbf{r} \neq \mathbf{0} \right\}, \tag{4}$$

with each one perturbing clean sample $\mathbf{x}(0) \sim p_0(\mathbf{x}(0))$ into noisy sample $\widetilde{\mathbf{x}}(0) = \mathbf{x}(0) + \boldsymbol{\epsilon}$, which is with respect to a distribution as $\widetilde{p}_{0,\mathbf{r}}(\widetilde{\mathbf{x}}(0)) = \int p_0(\mathbf{x}(0))\rho_{\mathbf{r}}(\widetilde{\mathbf{x}}(0) - \mathbf{x}(0))d\mathbf{x}(0)$. For zero risk $\mathbf{r} = 0$, it means the sample $\widetilde{\mathbf{x}}(0) \equiv \mathbf{x}(0)$ is noise-free.

*Remark* 3.1. This definition might not seem intuitive. For better understanding, let us take isotropic Gaussian perturbation as an example. In this case, the risk vector $\mathbf{r}$ can be simplified as a scalar $r$ and the family of noise distributions $\mathcal{P}_{\epsilon}$ is as $\{\mathcal{N}(\mathbf{0}, r\mathbf{I}) \mid r > 0\}$.

*Remark* 3.2. The operation of noise perturbation can be regarded as a form of "local averaging", which is typically not reversible. Even suppose that the reverse operation was possible, recovering the potential clean sample $\mathbf{x}(0)$ from noisy sample $\widetilde{\mathbf{x}}(0)$, would require knowledge of the probabilistic densities of samples, which are not accessible in practice.

**Motivation and definition of risk-sensitive SDE.** In light of the *misguidance effect* of noisy sample $\widetilde{\mathbf{x}}(0)$, we aim to seek an alternative diffusion process parameterized by the risk $\mathbf{r}$, such that noisy sample $(\widetilde{\mathbf{x}}(0), \mathbf{r})$ under this process has the same distribution $\widetilde{p}_{t,\mathbf{r}}(\mathbf{x})$ at some iteration $t$ in $[0, T]$ as that of clean sample $\mathbf{x}(0)$ under the ordinary diffusion process: $p_t(\mathbf{x})$. For iteration $t$ where the equality $\widetilde{p}_{t,\mathbf{r}}(\mathbf{x}) = p_t(\mathbf{x})$ holds, the score function of noisy samples: $\nabla_{\mathbf{x}} \ln \widetilde{p}_{t,\mathbf{r}}(\mathbf{x})$, can be used to safely optimize model $\mathbf{s}_{\boldsymbol{\theta}}(\mathbf{x}, t)$. The new process chosen in this spirit is a specific choice of SDE whose parameterization includes the risk vector $\mathbf{r}$. We name such an SDE as *risk-sensitive SDE*, with a strict definition as follows.

**Definition 3.2** (Risk-sensitive SDE). For a noisy sample $\widetilde{\mathbf{x}}(0)$ with risk vector $\mathbf{r}$, the *risk-sensitive SDE* is a type of SDE that incorporates the risk $\mathbf{r}$ into its coefficients, extending a sample vector $\widetilde{\mathbf{x}}(0)$ into a dynamics $\{\widetilde{\mathbf{x}}(t)\}_{t \in [0,T]}$ as

$$d\widetilde{\mathbf{x}}(t) = (\mathbf{f}(\mathbf{r}, t) \odot \widetilde{\mathbf{x}}(t))dt + \mathbf{g}(\mathbf{r}, t) \odot d\mathbf{w}(t), \tag{5}$$

where $\odot$ stands for the Hadamard product, and the coefficient functions $\mathbf{f}(\mathbf{r}, t), \mathbf{g}(\mathbf{r}, t)$ are everywhere continuous with right derivatives.

*Remark* 3.3. For zero risk $\mathbf{r} = \mathbf{0}$, the above SDE is fed with clean sample $\mathbf{x}(0)$, and thus corresponds to a standard diffusion model with risk-unaware coefficients $\mathbf{f}(\mathbf{0}, t), \mathbf{g}(\mathbf{0}, t)$. We refer to this particular case as *risk-unaware SDE*.

*Remark* 3.4. One might notice that risk-sensitive SDE is more expressive than the ordinarily defined diffusion process (i.e., Eq. (1)): The risk-sensitive coefficients $\mathbf{f}(\mathbf{r}, t), \mathbf{g}(\mathbf{r}, t)$ are vectors (i.e. non-isotropic), while risk-unaware coefficients $f(t), g(t)$ are just scalars. In Theorem 3.2, we will see this setting is essential for non-isotropic perturbation.

**Error measure: perturbation instability.** As previously discussed, we aim to find a type of risk-sensitive SDE that satisfies a nice property at some time step $t$: $\widetilde{p}_{t,\mathbf{r}}(\mathbf{x}) = p_t(\mathbf{x})$, which we define as *perturbation stability*. While this condition is indeed possible to reach for Gaussian noises, we will see in Theorem 3.1 that it is not achievable in the case of non-Gaussian perturbation. Therefore, we have to introduce a new "criterion" that generalize the stability condition, measuring how much it is violated. With this type of criterion, we can score all the coefficient candidates of a risk-sensitive SDE and search for the best candidate, which minimizes the stability violation.

Because probability densities are uniquely determined by their *cumulant-generating functions* (i.e., log-characteristics functions) (Chung, 2001), an obvious way to define the criterion is to measure the mean square error (Weisberg, 2005) between the cumulant-generating function of $\widetilde{p}_{t,\mathbf{r}}(\mathbf{x})$ and that of $p_t(\mathbf{x})$. A formal definition is in the following.

**Definition 3.3** (Measure of Perturbation Instability). For a given risk vector $\mathbf{r}$ and time step $t$, the *perturbation instability* $\mathcal{S}_t(\mathbf{r})$ of a risk-sensitive SDE (as defined in Eq. (5)) measures the discrepancy between its marginal density $\widetilde{p}_{t,\mathbf{r}}(\mathbf{x})$ for a noisy sample $\widetilde{\mathbf{x}}(0)$ and that of the ordinary diffusion process $p_t(\mathbf{x})$ for a clean sample $\mathbf{x}(0)$ as:

$$\mathcal{S}_t(\mathbf{r}) = \sup_{p_0(\mathbf{x})} \left( \int_{\mathbb{R}^D} \Omega(\mathbf{y}) \Big| \widetilde{\chi}_{t,\mathbf{r}}(\mathbf{y}) - \chi_t(\mathbf{y}) \Big|^2 d\mathbf{y} \right), \tag{6}$$

where $\Omega(\mathbf{y}) : \mathbb{R}^D \to \mathbb{R}^+$ is a positive weight function and $|\cdot|$ is the complex modulus. In particular, $\widetilde{\chi}_{t,\mathbf{r}}(\mathbf{y}), \chi_t(\mathbf{y})$ respectively stand for the cumulant-generating functions (Chung, 2001) of $\widetilde{p}_{t,\mathbf{r}}(\mathbf{x}), p_t(\mathbf{x})$, which both depends on the distribution of real samples: $p_0(\mathbf{x})$.

*Remark* 3.5. Extending our terminology, we say a risk-sensitive SDE achieves *perturbation stability* at time step $t$ if and only if it also satisfies $\mathcal{S}_t(\mathbf{r}) = 0$. The forward direction of this claim is obvious and the reverse is proved in the appendix: Lemma G.1. Importantly, we will see in the next section that such stability is not always reachable. In that case, we say a risk-sensitive SDE, which achieves the infimum of $\mathcal{S}_t(\mathbf{r})$, has the property of *minimum instability*.

*Remark* 3.6. The significance of perturbation stability is that, when this property holds, then the desired equality $\nabla_{\mathbf{x}} \ln \widetilde{p}_t(\mathbf{x}) = \nabla_{\mathbf{x}} \ln p_t(\mathbf{x})$ will also hold. In this situation, the score-based model $\mathbf{s}_\theta(\mathbf{x}, t)$ can be robustly optimized with noisy samples $(\widetilde{\mathbf{x}}(0), \mathbf{r} \neq \mathbf{0})$.

One might adopt another way to define the instability measure, considering that there are many other methods (e.g., KL divergence (Shlens, 2014)) to quantify the discrepancy of two probability distributions. However, we find that our defined measure $\mathcal{S}_t(\mathbf{r})$ leads to meaningful theoretical results and performs well in experiments. We remain the explorations of other possible measures and their implications for future work.

## 3.2 MAIN THEORY

In this part, we aim to answer the following three questions:

1. In what conditions is there a *risk-sensitive SDE* that facilitates *perturbation stability*? For example, does this depend on specific noise types or sample distribution $p_0(\mathbf{x})$?

2. If the stability property is not reachable, is there a possibility to have a analytical solution that minimally violates the stability property?

3. In the above two situations, what are the actual forms of risk-sensitive SDE? Is it generalizable to extend the current diffusion models for application?

To improve readability, we present simplified theoretical results while preserving the key ideas. The complete theory and detailed proofs can be found in Appendices E, F, G.

**Answer to the 1st question.** Our theorems provide a satisfactory answer as follows.

**Theorem 3.1** (Simplified and Reinterpreted from Theorem E.1 and Proposition F.1). *The necessary and sufficient conditions for a risk-sensisitve SDE to achieve perturbation stability: $\widetilde{p}_{t,\mathbf{r}}(\mathbf{x}) = p_t(\mathbf{x})$, include: 1) the noisy sample $\widetilde{\mathbf{x}}(0)$ is perturbed by a diagonal Gaussian noise and the risk $\mathbf{r}$ indicates its variance; 2) the time step $t$ is within the stability interval $\mathcal{T}(\mathbf{r})$.*

*In particular, suppose the Gaussian noise is isotropic, then it suffices to represent the risk vector $\mathbf{r}$ as a scalar $\mathbf{r}$ and the form of risk-sensitive SDE under this condition is as*

$$\begin{cases} f(r,t) = \dfrac{d \ln u(t)}{dt}, \forall t \in [0,T] \\ g(r,t) = u(t)^2 \dfrac{d}{dt}\left(\dfrac{v(r,t)^2}{u(t)^2}\right), \forall t \in \mathcal{T}(r), \quad g(r,t) = 0, \forall t \in \mathcal{T}(r)^c \end{cases}, \tag{7}$$

*where $u(t), v(r,t)$ are continuous functions with right derivatives, satisfying*

$$v(r,t)^2 = \max(v(0,t)^2 - r^2 u(t)^2, 0), \tag{8}$$

*and $\mathcal{T}(r) = \{t \in [0,T] \mid v(r,t) > 0\}$ is defined as the stability interval. For zero risk $r = 0$, the above equations reduce to an ordinary risk-unaware diffusion model.*

We can see that the ideal situation with perturbation stability is reachable *if and only if* the noise distribution is Gaussian and the time step is within the stability interval. This conclusion is also very intuitive from two perspectives: Firstly, since the backbones of *risk-sensitive SDE* and diffusion model are in fact a drifted Brownian motion, it is not likely that our tool can reduce the impact of a noise distribution beyond Gaussian; Secondly, noisy sample $\widetilde{\mathbf{x}}(0)$ is surely less informative than the clean sample $\mathbf{x}(0)$, so it is reasonable that noisy samples cannot be used to correctly optimize the score-based model $\mathbf{s}_\theta(\mathbf{x}, t)$ at every time step $t$. In Theorem E.1 of the appendix, one can also find a more general conclusion for non-isotropic Gaussian noises.

**Algorithm 1** Training

1: **repeat**
2: Sample $(\tilde{\mathbf{x}}(0),\ \mathbf{r}\ )$ from the training set
3: Sample $t$ from *stability interval* $\mathcal{T}(\mathbf{r})$
4: $\boldsymbol{\eta} \sim \mathcal{N}(\mathbf{0},\mathbf{I})$
5: $\tilde{\mathbf{x}}(t) = \mathbf{u}(t) \odot \tilde{\mathbf{x}}(0) + \mathbf{v}(\mathbf{r},t) \odot \boldsymbol{\eta}$
6: Update $\boldsymbol{\theta}$ with $-\nabla_{\boldsymbol{\theta}}\| \boldsymbol{\eta}\ /\ \mathbf{v}(\mathbf{r},\mathbf{t})\ + \mathbf{s}_{\boldsymbol{\theta}}(\tilde{\mathbf{x}}(t),t)\|^2$
7: **until** converged

**Algorithm 2** Sampling

1: Set time points $\{t_M = T, t_{M-1}, \cdots, t_2, t_1 = 0\}$
2: Set zero risk $\mathbf{r} = \mathbf{0}$ and $\mathbf{x}(t_M) \sim p_T(\mathbf{x})$
3: **for** $i = M, M-1, \ldots, 2$ **do**
4: $\bar{\mathbf{s}}_{\boldsymbol{\theta}}(\mathbf{x}(t_i), t_i) = \mathbf{g}(\mathbf{r},t_i)^2 \odot \mathbf{s}_{\boldsymbol{\theta}}(\mathbf{x}(t_i), t_i)$
5: $\hat{\mathbf{b}}(\mathbf{x}(t_i), t_i) = \mathbf{f}(\mathbf{r},t_i) \odot \mathbf{x}(t_i) - \bar{\mathbf{s}}_{\boldsymbol{\theta}}(\mathbf{x}(t_i), t_i)$
6: $\boldsymbol{\eta} \sim \mathcal{N}(\mathbf{0}, (t_i - t_{i-1})\mathbf{I})$
7: $\mathbf{x}(t_{i-1}) = \mathbf{x}(t_i) - \hat{\mathbf{b}}(\cdot)(t_i - t_{i-1}) - \mathbf{g}(\mathbf{r},t_i) \odot \boldsymbol{\eta}$
8: **end for**

**Answer to the 3rd question for Gaussian noises.** We have the following corollary that extends VP SDE (Song et al., 2021) (i.e., the continuous relaxation of DDPM) to *risk-sensitive VP SDE*, which supports a robust optimization with isotropic Gaussian noises.

**Corollary 3.1** (Risk-sensitive VP SDE, Simplified from Corollary G.2). *Under the setting of isotropic Gaussian perturbation, the risk-sensitive SDE for VP SDE is parameterized as follows*

$$f(r,t) = -\frac{1}{2}\beta(t), \quad g(r,t) = \mathbb{1}\big(1 > (1 + r^2)\alpha(t)\big)\sqrt{\beta(t)}, \tag{9}$$

*where $\mathbb{1}(\cdot)$ is an indicator function and the coefficient $\alpha(t)$ is defined as $\alpha(t) = \exp(-\int_0^t \beta(s)ds)$. The stability interval in this case is $\mathcal{T}(r) = \{t \in [0,T] \mid \alpha(t)^{-1} > 1 + r^2\}$. As expected, for the special case with zero risk $r = 0$, the risk-sensitive SDE reduces to an ordinary VP SDE, with $f(0,t) = -\frac{1}{2}\beta(t)$, $g(0,t) = \sqrt{\beta(t)}$, and $\mathcal{T} = [0,T]$.*

Risk-sensitive VP SDE is the same as vanilla VP SDE for optimization with clean sample $(\mathbf{x}(0), r = 0)$, otherwise it will adopt a different coefficient $g(r,t)$ and a restricted set of sampling time steps $\mathcal{T}(r)$ to reduce the negative impact of noisy sample $(\tilde{\mathbf{x}}(0), r > 0)$. We will discuss this point more in the next section, with detailed optimization and sampling algorithms. Corollary G.2 in the appendix also provides its version for non-isotropic Gaussian noises.

**Answer to the 2nd question.** This question is very important, considering that the perturbation distributions in the real world might be non-Gaussian. As shown below, we can always find the optimal parameterization of risk-sensitive SDE that minimizes the negative impact of an arbitrarily complex noise distribution on the optimization of model $\mathbf{s}_{\boldsymbol{\theta}}(\mathbf{x}, t)$.

**Theorem 3.2** (General Stability Theory, Simplified from Theorem F.1). *Suppose the risk vector $\mathbf{r}$ is element-wise positive and controls a family of continuous noise distributions, with each one formulated as $\rho_{\mathbf{r}}(\boldsymbol{\epsilon}) : \mathbb{R}^D \to \mathbb{R}_+, \int \rho_{\mathbf{r}}(\boldsymbol{\epsilon})d\boldsymbol{\epsilon} = 1$, then the optimal coefficients for the risk-sensitive SDE to minimize the instability measure $\mathcal{S}_t(\mathbf{r})$ satisfy the following equality:*

$$\begin{cases} \mathbf{v}(\mathbf{r},t)^2 = \max\big(\mathbf{0}, \mathbf{v}(\mathbf{0},t)^2 + \boldsymbol{\Psi}(\mathbf{u}(t),\mathbf{r})\big) \\ \boldsymbol{\Psi}(\mathbf{u}(t),\mathbf{r}) = 2\Big(\int \Omega(\mathbf{y})[\mathbf{y}\mathbf{y}^\top]^2 d\mathbf{y}\Big)^{-1}\Big(\int \Omega(\mathbf{y})\ln\big|\exp\big(\boldsymbol{\chi}_{\mathbf{r}}(\mathbf{u}(t) \odot \mathbf{y})\big)\big|[\mathbf{y}]^2 d\mathbf{y}\Big) \end{cases}, \tag{10}$$

*where the vectorized coefficients $\mathbf{u}(t), \mathbf{v}(\mathbf{r},t)$ come from the formal definition of risk-sensitive SDE (i.e., Definition 3.2) and the new terms $\Omega(\mathbf{y}), \boldsymbol{\chi}_{\mathbf{r}}(\cdot)$ are basic elements that defines the instability measure $\mathcal{S}_t(\mathbf{r})$ (i.e., Definition 3.3).*

We can see that the general form of perturbation distribution $\rho_{\mathbf{r}}(\boldsymbol{\epsilon})$ incurs a very complex expression $\boldsymbol{\Psi}(\mathbf{u}(t),\mathbf{r})$ in the optimal coefficient $\mathbf{v}(\mathbf{r},t)$. In particular, if the noise $\boldsymbol{\epsilon}$ follows an isotropic Gaussian $\rho_r(\boldsymbol{\epsilon}) = \mathcal{N}(\boldsymbol{\epsilon};\mathbf{0},r\mathbf{I})$, then we can verify that $\boldsymbol{\Psi}(u(t),r) = r^2u(t)^2$ regardless of the weight function $\Omega(\mathbf{y})$, which is consistent with our previous conclusion: Theorem 3.1. Another complication of non-Gaussian noise is that: even for some isotropic noise distribution $\rho_{\mathbf{r}}(\boldsymbol{\epsilon})$, the term $\boldsymbol{\Psi}(\mathbf{u}(t),\mathbf{r})$ appearing in coefficient $\mathbf{v}(\mathbf{r},t)$ might have different expressions in different dimensions. Therefore, distinct from our Theorem 3.1, the vectorized coefficients $\mathbf{u}(t), \mathbf{v}(\mathbf{r},t)$ cannot be simplified into scalar functions (e.g., $u(t), v(\mathbf{r},t)$) for isotropic noise perturbation.

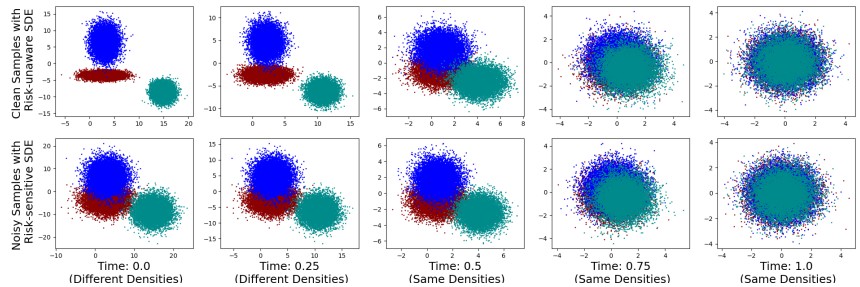

Figure 2: Comparison between the diffusion process of a standard VP SDE for clean samples (i.e., the upper 5 subfigures) and its alternative: *risk-sensitive SDE*, for Gaussian-corrupted samples (i.e., the lower 5 subfigures). With the proper risk-sensitive coefficients, the clean and noisy samples will have the same marginal densities in the *stability interval*: $t \in [0.26, 1]$.

**Answer to the 3rd question for non-Gaussian noises.** To finally answer this question for non-Gaussian noises, we have the below corollary that extends VE SDE (Song et al., 2021) (a type of diffusion model also used in Song et al. (2023)) to *risk-sensitive VE SDE*, which supports a robust optimization with Cauchy noises.

**Corollary 3.2** (Risk-sensitive VE SDE, Simplified from Corollary G.3). *For some properly defined weight function $\Omega(\mathbf{y})$ and an isotropic Cauchy perturbation specified by a scale $r$ as $\rho_r(\boldsymbol{\epsilon}) = \prod_{j=1}^{D} (\pi(r + \epsilon_j^2/r))^{-1}, \boldsymbol{\epsilon} = [\epsilon_1, \epsilon_2, \cdots, \epsilon_D]^\top$, the minimally-unstable risk-sensitive SDE for VE SDE has coefficients as*

$$f(r,t) = 0, \quad g(r,t) = \mathbb{1}\left(\sigma(t)^2 > \sigma(0)^2 + \frac{D+2}{D+5}r^2\right)\sqrt{\frac{d\sigma(t)^2}{dt}}. \tag{11}$$

*Notably, for the setting with no risk $r = 0$, risk-sensitive VE SDE reduces to the ordinary risk-unaware VE SDE, which has fixed coefficients $f(0,t) = 0, g(0,t) = \sqrt{d\sigma(t)^2/dt}$.*

With a heavy tail in the distribution, Cauchy noise has a high probability to drift a clean sample far away, exhibiting a very distinct behavior from Gaussian noises. In Sec. 5, our numerical experiments (e.g., Fig. 4) show that risk-sensitive VE SDE rarely generates outliers, indicating that the optimal risk-sensitive SDE is very effective in reducing the negative impact of Cauchy-corrupted samples. Corollary G.3 in the appendix stands as its version for non-isotropic Cauchy noises.

### 3.3 OPTIMIZATION AND SAMPLING

Similar to the score matching loss $\mathcal{L}$ of standard diffusion models, the loss function under the framework of *risk-sensitive SDE* shapes as

$$\mathcal{L}_{t,\mathbf{r}} = \mathbb{E}_{\mathbf{x} \sim \widetilde{p}_{t,\mathbf{r}}(\mathbf{x})}[\|\mathbf{s}_{\boldsymbol{\theta}}(\mathbf{x},t) - \nabla_{\mathbf{x}} \ln \widetilde{p}_{t,\mathbf{r}}(\mathbf{x})\|^2]. \tag{12}$$

Proposition G.1 in Appendix G shows that this loss function for noisy sample $(\widetilde{\mathbf{x}}, \mathbf{r})$ is equal to the score matching loss $\mathcal{L}$ for clean sample $(\mathbf{x}, \mathbf{r} = \mathbf{0})$ within the stability interval $\mathcal{T}(\mathbf{r})$, and has another form for computation in practice.

**Proposition 3.1** (Risk-free Loss, Simplified from Proposition G.1). *The loss function $\mathcal{L}_{t,\mathbf{r}}$ for risky sample $(\widetilde{\mathbf{x}}(0), \mathbf{r} \neq \mathbf{0})$ is equivalent to the below expression:*

$$\mathbb{E}_{\widetilde{\mathbf{x}}(0)\widetilde{p}_{0,\mathbf{r}}(\mathbf{x}), \boldsymbol{\eta} \sim \mathcal{N}(\mathbf{0},\mathbf{I})}\left[\|\boldsymbol{\eta} / \mathbf{v}(\mathbf{r},t) + \mathbf{s}_{\boldsymbol{\theta}}(\mathbf{u}(t) \odot \widetilde{\mathbf{x}}(0) + \mathbf{v}(\mathbf{r},t) \odot \boldsymbol{\eta}, t)\|^2\right], \tag{13}$$

*up to a constant. Here $\mathbf{u}(t), \mathbf{v}(\mathbf{r},t)$ are vectorized versions of terms $u(t), v(r,t)$ that appear in Eq. (7), with their formal definitions in Theorem E.1.*

We respectively show the training and sampling procedures in Algorithm 1 and Algorithm 2. We also highlight in blue the terms that differ from vanilla diffusion models. For the optimization algorithm, when $\mathbf{r} = \mathbf{0}$, the algorithm reduces to the optimization procedure of a vanilla diffusion model, with a trivial stability interval of $\mathcal{T}(\mathbf{r}) = [0, T]$. When the random variable $\mathbf{r}$ is non-zero, the risk-sensitive

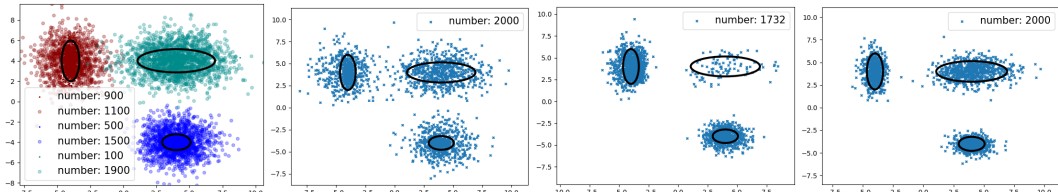

(a) Training data, polluted by Gaussian noises. (b) Samples from a standard diffusion model. (c) Samples from the risk-conditional baseline. (d) Samples from our risk-sensitive VP SDE.

Figure 3: Comparison on a Gaussian mixture data (Fig. 3(a), three-sigma regions as ellipses), with part of Gaussian-corrupted samples. Our model (Fig. 3(d)) mostly samples within the ellipses, while the samples from standard diffusion model (Fig. 3(b)) typically fall out of them, and conditional generation leads to an unbalanced generation distribution (Fig. 3(c)).

coefficient $\mathbf{v}(\mathbf{r}, t)$ and interval $\mathcal{T}(\mathbf{r})$ will guarantee that $\nabla_{\mathbf{x}} \ln p_t(\mathbf{x}) = \nabla_{\mathbf{x}} \ln \widetilde{p}_{t,\mathbf{r}}(\mathbf{x})$ for $t \in \mathcal{T}(\mathbf{r})$, such that the noisy sample $(\widetilde{\mathbf{x}}(0), \mathbf{r} \neq \mathbf{0})$ can be used to safely train the model $\mathbf{s}_{\boldsymbol{\theta}}(\mathbf{x}, t)$.

For the sampling algorithm, by setting zero risk $\mathbf{r} = \mathbf{0}$, the coefficients $\mathbf{f}(\mathbf{r}, t), \mathbf{g}(\mathbf{r}, t)$ become compatible with the model $\mathbf{s}_{\boldsymbol{\theta}}(\mathbf{x}, t)$ and together generate high-quality sample $\mathbf{x}(0)$. Our model will generate only clean samples $(\mathbf{x}(0), \mathbf{r} = \mathbf{0})$, but it was already able to capture the rich distribution information contained in noisy sample $(\widetilde{\mathbf{x}}(0), \mathbf{r} \neq \mathbf{0})$ during optimization.

## 4 RELATED WORK

**Similar setups.** To our knowledge, we are the first to study the problem setup of pairing noisy samples with *risk vectors* in the field of diffusion models. Some previous works (Ouyang et al., 2023; Kim et al., 2024) also focused on diffusion models with noisy data, though under different settings. For example, Unbiased Diffusion Model (Kim et al., 2024) considered the presence of both a biased dataset and a clean dataset and, thus, tackled a particular case of our setting: assigning risk 1 to the samples of biased dataset and risk 0 to those of the clean dataset. However, this model cannot be adapted to the common situation where different noisy samples might have different risks. Ambient Diffusion (Daras et al., 2024) aimed to handle a situation where the images are with missing pixel patches, which largely differs from our setting. As discussed in Appendix D, missingness is also a typical use case of our method: *risk-sensitive SDE*. Another related work is Na et al. (2024), which considered noisy labels, instead of noisy samples.

**Potential risk-conditional baseline.** Our proposed *risk-sensitive SDE* is the first method to address the problem setup of this paper. An alternative way is to adopt *conditional diffusion models* (Dhariwal & Nichol, 2021; Ho & Salimans, 2021), though there is surely no such work in the literature. One can treat the risk vector as that "conditional information" and apply these techniques to guide diffusion models to generate low-risk samples. We name this method as *risk-conditional baseline* in this paper and provide three different implementations in Appendix A. The main problem with risk-conditional diffusion models is that it might lead to a biased sampling distribution. To understand this point, note that conditional models essentially learn a joint distribution of samples and risk vectors. If one applies risk-conditional generation, which means a preference is imposed towards less noisy samples during generation, then the regions that are correlated with a high noise level in the sampling space tend to be ignored, yielding an unbalanced distribution of generated samples. In Sec. 5, our experiment results (e.g., Fig. 3) confirm this claim.

## 5 EXPERIMENTS

In this section, we provide two groups of empirical results: one is to verify the validity of our theorems in practice and the other is to apply our method: *risk-sensitive diffusion*, to real datasets. Due to the limited space, *we put other experiment results in Appendix B, which involve more baselines (e.g., Unbiased Diffusion Model), a different evaluation metric, and noisy images.*

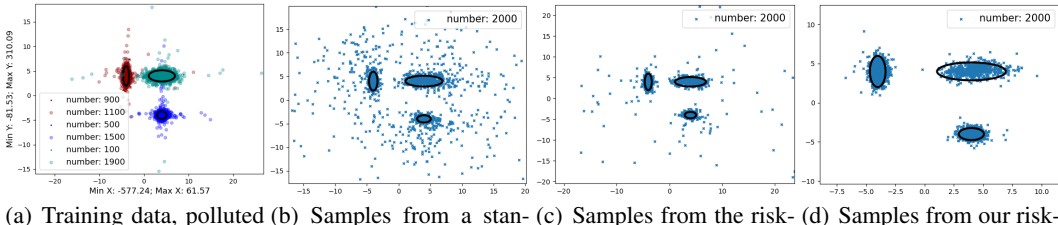

(a) Training data, polluted by Cauchy noises.
(b) Samples from a standard diffusion model.
(c) Samples from the risk-conditional baseline.
(d) Samples from our risk-sensitive VE SDE.

Figure 4: Comparison on Gaussian mixture data (Fig. 4(a)), with part of Cauchy-corrupted samples. *Despite minimal instability, our model still recovers the potential sample distribution* (Fig. 4(d)), while both baselines (Fig. 4(c) and Fig. 4(b)) incorrectly produce many outliers.

## 5.1 PROOF-OF-CONCEPT STUDIES

**Existence of stability interval.** With $T = 1, \beta(t) = 0.1 + 19.9t$, Fig. 2 shows an experiment, where VP SDE runs for clean samples $(\mathbf{x}(0), \mathbf{r} = \mathbf{0})$ while its risk-sensitive SDE (i.e., Corollary 3.1) operates on Gaussian-corrupted samples $(\tilde{\mathbf{x}}(0), \mathbf{r} = \mathbf{1})$. We can see that the clean and noisy samples follow the same distributions for step $t$ in the stability interval $\mathcal{T}(\mathbf{r}) = [0.26, 1]$. This experiment verifies that Theorem 3.1 is effective in practice.

**Risk-sensitive VP SDE under Gaussian perturbation.** Fig. 3 shows a comparison between our model (i.e., Corollary 3.2) and the risk-conditional baseline on a Gaussian-corrupted dataset Fig. 3(a). The conditional model underrepresents (Fig. 3(c)) the upper-right component at low-risk generation because it contains many more (i.e., $95\%$) noisy samples than other components. Instead, the generated samples of our model (Fig. 3(d)) are mostly unbiased, with no preference for a specific mixture component. This experiment confirms the weakness of the risk-conditional baseline and verifies that our model is more robust in practice.

**Risk-sensitive VE SDE under Gauchy perturbation.** With heavy tails, Cauchy distributions can usually distort a clean sample far away, exhibiting a distinct behavior from Gaussian noises. Fig. 4 shows an experiment on a Gaussian-mixture data (Fig. 3(a)), but with Cauchy noises corrupting samples. While risk-sensitive SDE cannot achieve *perturbation stability* in this case, our model still nicely recovers the distribution of clean samples and is robust to outliers (Fig. 4(d)). In contrast, the generated distributions of both standard (Fig. 4(c)) and conditional models (Fig. 4(b)) are seriously biased by outliers. This experiment highlights the flexibility of risk-sensitive SDEs and indicates that it can still be very effective under *minimally instability*.

## 5.2 APPLIED STUDIES

We now assess the Gaussian versions of *risk-sensitive SDE* (e.g., Corollary 3.1) on multiple real-world non-image datasets. We will find that our models still perform very well even when the data is highly noisy and the perturbation noise is in fact not Gaussian.

**Noisy time series.** As depicted in Fig. 1, time-series data might have irregularly spaced observations. To reshape such data into proper training samples for diffusion models, common practices are to first interpolate missing observations, resulting in noisy training samples. For this scenario, we adopt 2 medical time series datasets: MIMIC-III (Johnson et al., 2016) and WARDS (Alaa et al., 2017). For every time series in a dataset, we extract the observations of the first $48$ hours and select their top $5$ features with the highest variance, leading to a $240$-dimensional vector. To impute the missing values in a highly noisy manner, we apply a primitive method: Gaussian process, to interpolate them and estimate the variances, which are treated as the risk information.

**Noisy tabular data.** Tabular data is naturally composed of fixed-dimensional vectors, though they usually contain missing values, and imputations of those values introduce noise. For this scenario, we adopt 3 UCI datasets (Asuncion & Newman, 2007): Abalone, Telemonitoring, and Mushroom.

| Model | Time Series | | Tabular Data | | |
|---|---|---|---|---|---|
| | MIMIC-III | WARDS | Abalone | Telemonitoring | Mushroom |
| Standard VE SDE | 10.083 | 9.116 | 1.032 | 8.140 | 5.196 |
| VE SDE w/ Risk Regressor | 7.721 | 7.923 | 0.797 | 4.983 | 4.636 |
| VE SDE w/ Risk Variable | 6.549 | 7.314 | 0.853 | 5.161 | 4.970 |
| VE SDE w/ Risk Conditional | 5.926 | 5.951 | 0.612 | 3.159 | 4.101 |
| Our Model: Risk-sensitive VE SDE | **1.865** | **2.513** | **0.089** | **1.582** | **0.713** |
| Standard VP SDE | 9.135 | 8.765 | 0.925 | 9.935 | 6.238 |
| VP SDE w/ Risk Regressor | 7.981 | 7.832 | 0.732 | 4.197 | 5.327 |
| VP SDE w/ Risk Variable | 6.723 | 7.515 | 0.899 | 5.159 | 5.583 |
| VP SDE w/ Risk Conditional | 5.637 | 6.292 | 0.585 | 3.785 | 4.850 |
| Our Model: Risk-sensitive VP SDE | **1.625** | **2.584** | **0.077** | **1.462** | **0.852** |

Table 1: Wasserstein distances of different models on 5 datasets across 2 tasks. Part of model performances with another metric: MMD, are in Table 2 of Appendix B. The results not only show that our model significantly outperforms the baselines, but also indicate: *when the potential noise type is unknown, the assumption of Gaussian perturbation works well in practice*.

Since these datasets are initially complete, we force the missingness by randomly masking 5% of the entries in each dataset. For a data instance with missing values, we first apply k-nearest neighbors (KNN) algorithm (Peterson, 2009), to find the 10 closest samples. Then, we impute the missing value with their median and treat their absolute median deviation as the risk. Admittedly, the data generated in this way will be very noisy since KNN is certainly very inaccurate.

**Experiment setup and results.** Following commonly practices (Ho et al., 2020), we adopt the commonly used Wasserstein Distance (Heusel et al., 2017; Kolouri et al., 2019; Colombo et al., 2021) to evaluate the generative models, which measures the discrepancy of two distributions. For baselines, we adopt two standard diffusion models (VE SDE and VP SDE) and three risk-conditional models (details in Appendix A). In Table 1, we can see that our models significantly outperform all baselines regardless of the backbone model and the dataset. For example, with VE SDE as the backbone, our model has Wasserstein distances lower than Risk Conditional by 1.577 on the Telemonitoring dataset and 4.063 on MIMIC-III.

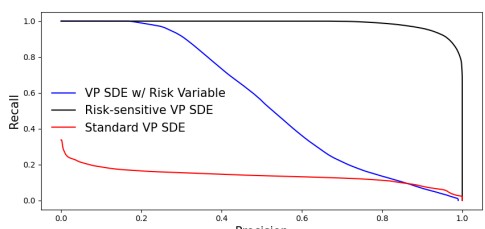

Figure 5: PRD curves (i.e., precision and recall scores) of our model and baselines on Telemonitoring dataset.

We also depict the PRD curves (Sajjadi et al., 2018; Razavi et al., 2019) of our model and two baselines on the Telemonitoring dataset. The PRD curve is similar to the precision-recall curve (Davis & Goadrich, 2006) used in testing classification models: the curves that locate more at the upper right corner indicate better performances. From Fig. 5, we can see that our model consistently achieves better recall scores than the baselines at identical precision scores. Plus, the PRD curve of our model is very close to the upper right corner, indicating the generation distribution of our model is almost consistent with the distribution of clean samples.

## 6 CONCLUSION

In this paper, we consider a novel problem setup to robustly train the diffusion models on noisy datasets: pairing noisy samples with *risk vectors*. To address this setup, we propose a principled method: *risk-sensitive SDE*, in the spirit of minimizing a defined measure: perturbation instability, which measures the negative effect of noisy samples. We have studied both the Gaussian and non-Gaussian noise perturbations, providing the optimal coefficients of risk-sensitive SDE in both cases. We have conducted extensive experiments on multiple real datasets, showing that risk-sensitive SDE can effectively handle noisy samples and significantly outperform previous baselines, even when the potential noise distribution might be non-Gaussian or mis-specified as Gaussian.

ACKNOWLEDGEMENTS

We sincerely appreciate the constructive and professional feedback from the ICLR reviewers, which has greatly improved this work. We also thank Jonathan Crabbe for several discussions. He gave suggestions on conditional generation baselines and time series datasets.

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

# Appendix

# A  RISK-CONDITIONAL DIFFUSION MODELS

An obvious way to adapt current diffusion models to the extra risk information $\mathbf{r}$ is conditional generation (Dhariwal & Nichol, 2021). The main drawback of conditional diffusion models is that it might have a biased sampling distribution. For example, suppose a regressor-guided diffusion model (Dhariwal & Nichol, 2021) is trained on a dataset composed of blurry pictures of dogs and clear pictures of cats, then the conditional model will generate very few images of dogs, given that they are associated with a higher risk than cats. In the following, we present three different implementations under this scheme.

## A.1  RISK AS THE VARIABLE

A naive implementation is first to let diffusion models learn the joint distribution of samples and risk vectors: $p_0(\mathbf{x}, \mathbf{r})$, and then regularize the reverse process for drawing samples of a low risk: $\mathbf{r} \approx \mathbf{0}$, from the trained model.

**Risk variable.**  In the optimization stage, we concatenate the sample and risk vectors as $\mathbf{z}(0) = \widetilde{\mathbf{x}}(0) \oplus \mathbf{r}$ in a column-wise manner, with Eq. (1) and Eq.(3) to train a vanilla diffusion model. We draw low-risk samples from the trained model at inference time through an improved backward SDE. Considering the technique of classifier guidance (Dhariwal & Nichol, 2021), we set a parameter-free regressor $-\| \cdot \|_2$: the minus square norm, which takes the last $D$ entries of variable $\mathbf{z}_{D+1:2D}(t) := \mathbf{r}(t)$ as the input and has a derivate as $-\nabla_{\mathbf{r}(t)} \|\mathbf{r}(t)\|_2 = -\mathbf{r}(t)/\|\mathbf{r}(t)\|_2$. With this regressor, the backward process (i.e., Eq. (2)) is updated as follows:

$$
\begin{aligned}
d\mathbf{z}(t) &= \Big(\mathbf{f}(\mathbf{z}(t), t) - g(t)^2(\nabla_{\mathbf{z}(t)} \ln p_t(\mathbf{z}(t)) - \nabla_{\mathbf{r}(t)}\|\mathbf{r}(t)\|_2)\Big) + g(t)d\bar{\mathbf{w}}(t) \\
&= \Big(\mathbf{f}(\mathbf{z}(t), t) - g(t)^2\Big(\nabla \ln p_t(\mathbf{z}(t)) - \frac{\mathbf{r}(t)}{\|\mathbf{r}(t)\|_2}\Big)\Big) + g(t)d\bar{\mathbf{w}}(t) \\
&\approx \Big(\mathbf{f}(\mathbf{z}(t), t) - g(t)^2\Big(\mathbf{s}_{\boldsymbol{\theta}}(\mathbf{z}(t), t) - \frac{\mathbf{r}(t)}{\|\mathbf{r}(t)\|_2}\Big)\Big) + g(t)d\bar{\mathbf{w}}(t),
\end{aligned}
\tag{14}
$$

where some redundant parts are omitted as $(\cdot)$. In practice, the gradient $-\nabla_{\mathbf{r}(t)}\|\mathbf{r}(t)\|_2$ is re-scaled with a positive coefficient $\gamma$, which trades diversity for quality. Intuitively, the regressor $-\|\cdot\|_2$ gradually reduces the norm of $\mathbf{r}(t)$ as decreasing iteration $t$ such that the final sample $\mathbf{x}(0) = \mathbf{z}_{1:D}(0)$ will be paired with low risk $\mathbf{r}(0)$.

## A.2  RISK AS THE CONDITIONAL

Another type of implementation treats the risk vector $\mathbf{r}$ as a generation conditional for diffusion models. Ideally, we can draw clean samples from a trained model by setting $\mathbf{r} = \mathbf{0}$.

**Risk conditional.**  There are two types of conditional diffusion models. The easier one is classifier-free (Ho & Salimans, 2021), which adds the risk vector $\mathbf{r}$ as an input to the score-based model: $\mathbf{s}_{\boldsymbol{\theta}}(\mathbf{x}, t, \mathbf{r})$. Eq. (1) and Eq. (3) are the same to train the new model, but the input $\mathbf{r}$ is randomly masked with a dummy variable $\varnothing$ to permit unconditional generation. For inference, the score function $\nabla_{\mathbf{x}(t)} \ln p_t(\mathbf{x}(t))$ in the backward SDE (i.e., Eq. (2)) is replaced with

$$
(1 + \gamma)\mathbf{s}_{\boldsymbol{\theta}}(\mathbf{x}(t), t, \mathbf{r} = \mathbf{0}) - \gamma \mathbf{s}_{\boldsymbol{\theta}}(\mathbf{x}(t), t, \varnothing),
\tag{15}
$$

where $\gamma$ is a non-negative number that plays a similar role to the first model.

**Risk regressor.**  The other one is just classifier-guided sampling. While the diffusion model remains the same, we separately train a regressor $\mathbf{h}$ to predict the risk of a sample:

$$
\widehat{\mathbf{r}} = \ln(1 + \exp(\mathbf{h}(\mathbf{x})),
\tag{16}
$$

where the SoftPlus function $\exp(1+\ln(\cdot))$ is to ensure that the final output $\widehat{\mathbf{r}}$ is positive and variable $\mathbf{x}$ is either raw sample $\widetilde{\mathbf{x}}(0)$ or its noisy version $\widetilde{\mathbf{x}}(t), t > 0$ (obtained by Eq. (1)). For implementation, the regressor $\mathbf{h}$ can be any neural network and is optimized with a square loss: $\|\mathbf{r} - \widehat{\mathbf{r}}\|_2^2$. To have

| Dataset | Abalone | Telemonitoring |
|---|---|---|
| Standard VP SDE | 0.01056 | 0.01667 |
| VP SDE w/ Risk Conditional | 0.00766 | 0.01267 |
| Our Model: VP SDE w/ Risk-sensitive Diffusion | **0.00198** | **0.00717** |

Table 2: Model performances measured by another metric: MMD.

a scalar outcome, we wrap the regressor as $-\sum_{i=1}^{D} \ln(1 + \exp(h_i(\mathbf{x})))$, with expansion $\mathbf{h}(\mathbf{x}) = [h_1(\mathbf{x}), h_2(\mathbf{x}), \cdots, h_D(\mathbf{x})]^\top$ and derivative

$$\nabla_{\mathbf{x}}\Big(-\sum_{i=1}^{D} \ln(1 + \exp(h_i(\mathbf{x})))\Big) = -\Big[\sum_{i=1}^{D} \sigma(h_i(\mathbf{x}))\frac{\partial h_i(\mathbf{x})}{\partial x_1}, \cdots, \sum_{i=1}^{D} \sigma(\cdot)\frac{\partial h_i(\mathbf{x})}{\partial x_D}\Big]^\top, \qquad (17)$$

where $\sigma$ is the Sigmoid function. Similar to our first model, we apply this derivate to regularize the backward process of a trained diffusion model as

$$d\mathbf{x}(\mathbf{t}) \approx (\mathbf{f}(\mathbf{x}(t), t) - g(t)^2(\mathbf{s}_{\boldsymbol{\theta}}(\mathbf{x}(t), t) - \gamma\nabla_{\mathbf{x}}\Big(-\sum_{i=1}^{D} \ln(\cdot)\Big) + g(t)d\bar{\mathbf{w}}(t), \qquad (18)$$

where $\gamma \in \mathbb{R}^+ \bigcup\{0\}$ is non-negative.

# B ADDITIONAL EXPERIMENTS

We have performed extra experiments to further confirm the effectiveness of our method: *risk-sensitive SDE*, including comparisons with other baselines (e.g., Ambient Diffusion), the introduction of another evaluation metric: MMD, and a study on the image dataset.

**Another evaluation metric: MMD.**   Recognizing the importance of diverse evaluation criteria, we introduced the Maximum Mean Discrepancy (MMD) metric (Jia et al., 2017) in our analysis. This additional metric further affirms the great effectiveness of our method to bridge the gap between generated and real distributions across different noisy datasets. As shown in Table 2, our model still significantly outperforms the baselines in terms of the new metric: MMD. Overall, given the consistent best results of our method across different datasets and evaluation metrics (including Wasserstein Distance, Precision-Recall Curves, and MMD), we believe the effectiveness of our method: risk-sensitive diffusion, is well justified.

**Other baselines from similar settings.**   While the works of Ambient Diffusion (Daras et al., 2024) and Unbiased Diffusion Model (Kim et al., 2024) also discuss the training of diffusion models on imperfect data, their settings are either a particular case of ours or differ altogether. In the following, we compare the settings and purpose of each of these works to risk-sensitive diffusion and empirically compare our method to these baselines:

- Ambient Diffusion addressed the setting of images with missing pixels, which, although it is one of the applications of risk-sensitive diffusion, our method is more broadly applicable to the general case of "noisy pixels";

- Unbiased Diffusion Model considered the presence of both a biased dataset and a clean dataset and, thus, tackled a particular case of our method. If one attributes risk 1 to the biased dataset and risk 0 to the clean dataset, then our method can be used to learn an unbiased diffusion model. However, our method encompasses a much more broad range of settings. In particular, this method does not admit varying risks for different features or risk distributions, as risk-sensitive diffusion does.

The experiments of our paper include tabular data with missing values, which constitute a valid benchmark for our method against the baselines. Table 3 summarizes the obtained new results. The performances of Ambient Diffusion and Unbiased Diffusion Model are comparable to our baseline VP SDE w/ Risk Conditional, and our method: risk-sensitive diffusion, significantly outperforms all of them, thereby showing the value of our contribution.

| Dataset | Abalone | Telemonitoring |
|---|---|---|
| Standard VP SDE | 0.925 | 9.935 |
| Ambient Diffusion | 0.482 | 4.253 |
| Unbiased Diffusion Models | 0.679 | 4.991 |
| VP SDE w/ Risk Conditional | 0.585 | 3.785 |
| Our Model: VP SDE w/ Risk-sensitive Diffusion | **0.077** | **1.462** |

Table 3: Comparison between our method and the baselines from similar settings.

| Model | 20% noisy images | 40% noisy images |
|---|---|---|
| Standard VP SDE | 8.31 | 13.29 |
| Unbiased Diffusion Model | 5.37 | 8.53 |
| VP SDE w/ Risk Conditional | 6.57 | 9.15 |
| Our Model: VP SDE w/ Risk-sensitive Diffusion | **4.89** | **6.97** |

Table 4: Model performances on CIFAR-10 with certain numbers of noisy images.

**Study on noisy images.** We also explored the performance of our risk-sensitive diffusion framework on image data: CIFAR-10 (Krizhevsky et al., 2009) images with pixel noises. Specifically, we perturb certain portions of CIFAR-10 images with Gaussian noises and compare our method with two baselines (i.e., risk-conditional diffusion model and Unbiased Diffusion Model). Table 4 contains the results in terms of FID scores, which show that risk-sensitive diffusion outperforms other methods in this setting, too. Our finding reveals that our method outperforms conventional and recent approaches even in domains outside our primary focus, further underscoring its general applicability and robustness.

## C   BACKWARD RISK-SENSITIVE SDE

Although the backward risk-sensitive SDE is not necessary for our method and theorems, we still provide it for reference. Let a risk-sensitive SDE be of the form as

$$d\mathbf{x}(t) = (\mathbf{f}(t) \odot \mathbf{x}(t))dt + \mathrm{diag}(\mathbf{g}(\mathbf{r}, t))d\mathbf{w}(t),$$

where both coefficient functions $\mathbf{f}(t), \mathbf{g}(\mathbf{r}, t)$ are everywhere continuous with right derivatives. According to (Anderson, 1982), we can get the corresponding backward SDE as

$$d\mathbf{x} = \Big(\mathbf{f}(t) \odot \mathbf{x}(t) - \nabla_{\mathbf{x}} \cdot \mathrm{diag}(\mathbf{g}(\mathbf{r}, t)^2) - \mathrm{diag}(\mathbf{g}(\mathbf{r}, t)^2)\nabla_{\mathbf{x}} \ln p_t(\mathbf{x} \mid \mathbf{r})\Big)dt + \mathrm{diag}(\mathbf{g}(\mathbf{r}, t))d\bar{\mathbf{w}}(t)$$

$$= \Big(\mathbf{f}(t) \odot \mathbf{x}(t) - \mathbf{g}(\mathbf{r}, t)^2 \odot \nabla_{\mathbf{x}} \ln p_t(\mathbf{x} \mid \mathbf{r})\Big)dt + \mathbf{g}(\mathbf{r}, t) \odot d\bar{\mathbf{w}}(t).$$
(19)

While $\mathbf{g}(\mathbf{r}, t) = \mathbf{0}$ is possible for $t \in \{t \in [1, T] \mid \bar{\mathbf{g}}(\mathbf{0}, t)^2 - \bar{\mathbf{f}}(\mathbf{r}, t)^2 \odot \tilde{\mathbf{r}}^2 \neq \bar{\mathbf{g}}(\mathbf{r}, t)^2\}$, our above conclusion still applies and the backward SDE is the same as the forward one in that case.

## D   WIDE APPLICATIONS

The problem formulation of our paper: noisy sample $\tilde{\mathbf{x}}(0)$ paired with risk vector $\mathbf{r}$ (i.e., accessible information of data quality), is not only rarely seen in the field of generative models, but also highly motivated by real-world applications.

**Data with Accessible Risks.** In many cases, data are naturally born with information indicating their quality. Here are some examples from the biological and sensor domains:

- Polymerase chain reaction (PCR) is widely used In DNA sequencing to produce genomic data. Since the accuracy of PCR is largely affected by the Guanine-Cytosine content (GC-content), researchers typically regard this information as a primary indicator of the data quality (Kumar & Kaur, 2014; Laursen et al., 2017);

- Laser radars resort to laser beams to generate data, indicating the spatial positions of physical objects. This type of sensor data tends to be very noisy, so the radars also provide the engineers with other data sources, such as light strength (Steinvall & Chevalier, 2005) and device states (e.g., excessive voltage and temperature) (Carmer & Peterson, 1996), which reflect the data quality;

- Gyroscopes are commonly used in navigation and robotics, which measure the angular velocity of an object. This type of sensor can inherently estimate the data quality to provide engineers with more information, including bias (offset from true value) (Kirkko-Jaakkola et al., 2012) and scale factor (deviation from the expected sensitivity) (Tang et al., 2017).

Recently, there is a growing trend towards applying generative models to scentific (e.g., AI for Science) (Chung & Ye, 2022; Huang et al., 2023) and industrial data (e.g., Smart Manufacturing) (Kapelyukh et al., 2023; Sridhar et al., 2023). Including the above examples, those types of data are generally noisy and come with risk information, where our proposed *risk-sensitive SDE* will play a key role.

**Data without Available Risks.** There are also situations where the risk information $\mathbf{r}$ for noisy sample $\tilde{\mathbf{x}}(0)$ is not available. However, since our definition of the risk vector $\mathbf{r}$ is not limited, it is very likely that one can find an alternative to the vector in a low-effort manner, without resorting to manual annotation and expert knowledge. Typical examples are time series and tabular data in the medical domain (e.g., MIMIC dataset (Johnson et al., 2016)). Specifically, because these two types of medical data either are irregular (Sun et al., 2020) or have missing values (Lin & Tsai, 2020), one will preprocess the data with interpolation and imputation before using them. Such preprocessing techniques are commonly not fully accurate, leading the final data to be noisy. In this situation, there are at least two very efficient ways to harvest the risk information:

- Some interpolation and imputation models can inherently quantify the uncertainties of their predictions. For example, Gaussian Process (MacKay et al., 1998) and MissForest (Stekhoven & Bühlmann, 2012). The uncertainties provided by these models can be treated as the risk information;

- There are a number of approaches (e.g., Bayesian Dropout (Gal & Ghahramani, 2016)) in the field of Bayesian Deep Learning (Kendall & Gal, 2017), which estimate the prediction uncertainty of a black-box model. If a preprocessing tool provides no extra information for its output, one can apply such a method to construct the risk vector.

Even for high-quality image data, the concept of risk information still applies and it is convenient to find the risk vector. For example, images in the ImageNet dataset (Deng et al., 2009) are of various sizes. To train a deep learning model (e.g., GAN (Goodfellow et al., 2020)) on that dataset, one has to first let all images have the same shape. In this way, a clear image of a small shape $H \times W$ will be expanded to a fuzzy image of a big shape $H' \times W'$. This type of sample is certainly noisy for the model and one can regard this ratio $\sqrt{(H'W')/(HW)} - 1$ as the risk information.

**Determination of the Noise Types.** A question might arise: How can we determine the noise type for applying the risk-sensitive SDE? In some cases, we can infer it based on the mechanism that generates risk vectors. For example, the arrival time of an unobserved sample in a continuous-time Markov Chain (Anderson, 2012) has an exponential distribution. In other scenarios where the risk-generating mechanism is unknown, we can suppose the noise is Gaussian, similar to the treatments in Conformal Prediction (Zaffran et al., 2023; Angelopoulos & Bates, 2021) and Kalman Filter (Kim et al., 2018). In Appendix 5, our numerical experiments (e.g., Table 1) show that this assumption works quite well.

## E    STABILITY FOR GAUSSIAN PERTURBATION

In this section, we aim to find the optimal *risk-sensitive coefficients* $\mathbf{f}(\mathbf{r}, t), \mathbf{g}(\mathbf{r}, t)$ that let the *risk-sensitive SDE* achieves stability under Gaussian perturbation. We will first prove a lemma about the kernel of risk-sensitive SDE and then dive into the main theorem.

### E.1 RISK-SENSITIVE KERNEL

For analysis purpose, we provide a lemma that determines the form of kernel $\widetilde{p}_{t|0,\mathbf{r}}(\mathbf{x} \mid \widetilde{\mathbf{x}}(0))$ (i.e., the density of $\mathbf{x}$ conditioning on noisy sample $\widetilde{\mathbf{x}}(0)$) for a given risk-sensitive SDE.

**Lemma E.1** (Kernel of Risk-sensitive SDE). *Suppose we have a risk-sensitive SDE defined as Eq. (5), then its associated kernel $\widetilde{p}_{t|0,\mathbf{r}}(\widetilde{\mathbf{x}}(t) \mid \widetilde{\mathbf{x}}(0))$ shapes as*

$$\begin{cases} \widetilde{p}_{t|0,\mathbf{r}}(\widetilde{\mathbf{x}}(t) \mid \widetilde{\mathbf{x}}(0)) = \mathcal{N}(\widetilde{\mathbf{x}}(t); \overline{\mathbf{f}}(\mathbf{r}, t) \odot \widetilde{\mathbf{x}}(0), \mathrm{diag}(\overline{\mathbf{g}}(\mathbf{r}, t)^2)) \\[2mm] \mathbf{f}(\mathbf{r}, t) = \dfrac{d \ln \overline{\mathbf{f}}(\mathbf{r}, t)}{dt} \\[3mm] \mathbf{g}(\mathbf{r}, t)^2 = \overline{\mathbf{f}}(\mathbf{r}, t)^2 \odot \dfrac{d}{dt}\left(\dfrac{\overline{\mathbf{g}}(\mathbf{r}, t)^2}{\overline{\mathbf{f}}(\mathbf{r}, t)^2}\right) \end{cases}, \qquad (20)$$

*where $\overline{\mathbf{f}}(\mathbf{r}, 0) = \mathbf{1}, \overline{\mathbf{g}}(\mathbf{r}, 0) = \mathbf{0}$ and operation* $\mathrm{diag}$ *expands a vector into a diagonal matrix.*

*Proof.* For SDE, its kernel is a Gaussian distribution and the first moment is also affine if the drift term is affine (Evans, 2012; Oksendal, 2013). Based on these facts, we can suppose that the kernel $\widetilde{p}_{t|0,\mathbf{r}}(\widetilde{\mathbf{x}}(t) \mid \widetilde{\mathbf{x}}(0))$ of risk-sensitive SDE has the following form:

$$\widetilde{p}_{t|0,\mathbf{r}}(\widetilde{\mathbf{x}}(t) \mid \widetilde{\mathbf{x}}(0)) = \mathcal{N}(\widetilde{\mathbf{x}}(t); \overline{\mathbf{F}}(\mathbf{r}, t)\widetilde{\mathbf{x}}(0), \overline{\mathbf{G}}(\mathbf{r}, t)^2), \qquad (21)$$

where $\overline{\mathbf{F}}(\mathbf{r}, t), \overline{\mathbf{G}}(\mathbf{r}, t)$ are undetermined functions that output diagonal matrices.

Considering a corner case where $t \to 0$, we can infer that $\overline{\mathbf{F}}(\mathbf{r}, 0) = \mathbf{I}$ and $\overline{\mathbf{G}}(\mathbf{r}, 0) = \mathbf{0}$. For $t > 0$, assume $\delta t > 0$ and $\delta t \approx 0$, then we have

$$\begin{aligned} \widetilde{p}_{t+\delta t|0,\mathbf{r}}(\widetilde{\mathbf{x}}(t + \delta t) \mid \widetilde{\mathbf{x}}(0)) &= \int \widetilde{p}_{t+\delta t, t|0,\mathbf{r}}(\widetilde{\mathbf{x}}(t + \delta t), \widetilde{\mathbf{x}}(t) \mid \widetilde{\mathbf{x}}(0)) d\widetilde{\mathbf{x}}(t) \\ &= \int \widetilde{p}_{t+\delta t|t,\mathbf{r}}(\widetilde{\mathbf{x}}(t + \delta t) \mid \widetilde{\mathbf{x}}(t))\widetilde{p}_{t|0,\mathbf{r}}(\widetilde{\mathbf{x}}(t) \mid \widetilde{\mathbf{x}}(0)) d\widetilde{\mathbf{x}}(t) \end{aligned}. \qquad (22)$$

Note that $\widetilde{p}_{t+\delta t|t,\mathbf{r}}(\widetilde{\mathbf{x}}(t + \delta t) \mid \widetilde{\mathbf{x}}(t), \widetilde{\mathbf{x}}(0)) = \widetilde{p}_{t+\delta t|t,\mathbf{r}}(\widetilde{\mathbf{x}}(t + \delta t) \mid \widetilde{\mathbf{x}}(t))$ because of the Markov property. For notational convenience, we represent the risk-sensitive SDE as

$$d\widetilde{\mathbf{x}}(t) = \mathbf{F}(\mathbf{r}, t)\widetilde{\mathbf{x}}(t)dt + \mathbf{G}(\mathbf{r}, t)d\mathbf{w}(t). \qquad (23)$$

where $\mathbf{F}(\mathbf{r}, t) = \mathrm{diag}(\mathbf{f}(\mathbf{r}, t))$ and $\mathbf{G}(\mathbf{r}, t) = \mathrm{diag}(\mathbf{g}(\mathbf{r}, t))$. According to Eqs. (21) and (23), we can have the following equation:

$$\begin{cases} \widetilde{\mathbf{x}}(t + \delta t) = \overline{\mathbf{F}}(\mathbf{r}, t + \delta t)\widetilde{\mathbf{x}}(0) + \overline{\mathbf{G}}(\mathbf{r}, t + \delta t)\boldsymbol{\epsilon}_1 \\ \widetilde{\mathbf{x}}(t) = \overline{\mathbf{F}}(\mathbf{r}, t)\widetilde{\mathbf{x}}(0) + \overline{\mathbf{G}}(\mathbf{r}, t)\boldsymbol{\epsilon}_2 \\ \widetilde{\mathbf{x}}(t + \delta t) = (\mathbf{I} + \delta t\mathbf{F}(\mathbf{r}, t))\widetilde{\mathbf{x}}(t) + \sqrt{\delta t}\mathbf{G}(\mathbf{r}, t)\boldsymbol{\epsilon}_3 \end{cases}, \qquad (24)$$

where $\boldsymbol{\epsilon}_1, \boldsymbol{\epsilon}_2, \boldsymbol{\epsilon}_3 \sim \mathcal{N}(\mathbf{0}, \mathbf{I})$ are independent Gaussian noises. Combining the last two equalities, we have the following equality:

$$\widetilde{\mathbf{x}}(t + \delta t) = (\mathbf{I} + \delta t\mathbf{F}(\mathbf{r}, t))\overline{\mathbf{F}}(\mathbf{r}, t)\mathbf{x}(0) + ((\mathbf{I} + \delta t\mathbf{F}(\mathbf{r}, t))\overline{\mathbf{G}}(\mathbf{r}, t)\boldsymbol{\epsilon}_2 + \sqrt{\delta t}\mathbf{G}(\mathbf{r}, t)\boldsymbol{\epsilon}_3). \qquad (25)$$

Comparing the above two equations, we have:

$$\begin{cases} \overline{\mathbf{F}}(\mathbf{r}, t + \delta t) = (\mathbf{I} + \delta t\mathbf{F}(\mathbf{r}, t))\overline{\mathbf{F}}(\mathbf{r}, t) \\ \overline{\mathbf{G}}(\mathbf{r}, t + \delta t)^2 = (\mathbf{I} + \delta t\mathbf{F}(\mathbf{r}, t))^2\overline{\mathbf{G}}(\mathbf{r}, t)^2 + \delta t\mathbf{G}(\mathbf{r}, t)^2 \end{cases}. \qquad (26)$$

Let $\delta t \to 0$, this equation can be converted into a differential form:

$$\mathbf{F}(\mathbf{r}, t) = \frac{d \ln \overline{\mathbf{F}}(\mathbf{r}, t)}{dt}, \quad \mathbf{G}(\mathbf{r}, t)^2 = \frac{d\overline{\mathbf{G}}(\mathbf{r}, t)^2}{dt} - 2\frac{d \ln \overline{\mathbf{F}}(\mathbf{r}, t)}{dt}\overline{\mathbf{G}}(\mathbf{r}, t)^2. \qquad (27)$$

If $\overline{\mathbf{G}}(\mathbf{r}, t)$ is only continuous but not differentiable, then the term $d \ln \overline{\mathbf{F}}(\mathbf{r}, t)/dt$ indicates its right derivative. Now, by converting all matrix-valued functions $\overline{\mathbf{F}}(\mathbf{r}, t), \overline{\mathbf{G}}(\mathbf{r}, t), \mathbf{F}(\mathbf{r}, t), \mathbf{G}(\mathbf{r}, t)$ into their vector forms $\overline{\mathbf{f}}(\mathbf{r}, t), \overline{\mathbf{g}}(\mathbf{r}, t), \mathbf{f}(\mathbf{r}, t), \mathbf{g}(\mathbf{r}, t)$, we have

$$\mathbf{f}(\mathbf{r}, t) = \frac{d \ln \overline{\mathbf{f}}(\mathbf{r}, t)}{dt}, \quad \mathbf{g}(\mathbf{r}, t)^2 = \overline{\mathbf{f}}(\mathbf{r}, t)^2 \odot \frac{d}{dt}\left(\frac{\overline{\mathbf{g}}(\mathbf{r}, t)^2}{\overline{\mathbf{f}}(\mathbf{r}, t)^2}\right), \qquad (28)$$

where operations $\odot$ and $\oslash$ respectively denote element-wise product and division. The initial conditions for $\overline{\mathbf{F}}(\mathbf{r}, t), \overline{\mathbf{G}}(\mathbf{r}, t)$ can also be directly transferred to $\widetilde{\mathbf{f}}(\mathbf{r}, t), \widetilde{\mathbf{g}}(\mathbf{r}, t)$. $\qquad \square$

The above lemma is very useful. We will also see it in Sec. G.

## E.2 SOLUTION FOR GAUSSIAN PERTURBATION

Provided with Lemma E.1, the following theorem gives a sufficient condition for letting the risk-sensitive SDE achieve *perturbation stability* for Gaussian noises. In Sec. F, we will also see that this condition is both sufficient and necessary.

**Theorem E.1** (Risk-sensitive SDE for Gaussian Perturbation). *Suppose that we have a Gaussian family of perturbation distributions: $\mathcal{P}_\epsilon = \{\mathcal{N}(\mathbf{0}, \mathrm{diag}(\mathbf{r}^2)) \mid \mathbf{r} \neq \mathbf{0}\}$, then the risk-sensitive SDE (as defined in Eq. (5)) parameterized as below:*

$$
\begin{cases}
\mathbf{f}(\mathbf{r}, t) = \dfrac{d \ln \mathbf{u}(t)}{dt}, \forall t \in [0, T] \\
\mathbf{g}(\mathbf{r}, t) = \mathbf{u}(t)^2 \odot \dfrac{d}{dt}\Big(\dfrac{\mathbf{v}(\mathbf{r}, t)^2}{\mathbf{u}(t)^2}\Big), \forall t \in \mathcal{T}(\mathbf{r}) \\
\mathbf{g}(\mathbf{r}, t) = \mathbf{0}, \forall t \in [0, T] \bigcap \mathcal{T}(\mathbf{r})^c
\end{cases}
\tag{29}
$$

*has the property of perturbation stability (i.e., $\mathcal{S}_t(\mathbf{r}) = 0$) for any $t$ in*

$$
\mathcal{T}(\mathbf{r}) \equiv \{t \in [0, T] \mid \mathbf{v}(\mathbf{r}, t)^2 + \mathbf{r}^2 \odot \mathbf{u}(t)^2 = \mathbf{v}(\mathbf{r}, 0)^2\},
\tag{30}
$$

*regardless of the weight function $\Omega(\mathbf{y})$. Here $\mathcal{T}(\mathbf{r})^c$ represents the complement $\mathcal{T}(\mathbf{r})$ and $\mathbf{u}(t), \mathbf{v}(\mathbf{r}, t)$ are arbitrary functions that are everywhere continuous with right derivatives.*

*In particular, for zero risk $\mathbf{r} = \mathbf{0}$, the equations correspond to the associated risk-unaware diffusion process for clean sample $(\widetilde{\mathbf{x}}(0) = \mathbf{x}(0), \mathbf{r} = \mathbf{0})$.*

*Proof.* According to Lemma E.1, the kernel of risk-sensitive SDE shapes as

$$
\widetilde{p}_{t|0,\mathbf{r}}(\widetilde{\mathbf{x}}(t) \mid \widetilde{\mathbf{x}}(0)) = \mathcal{N}(\widetilde{\mathbf{x}}(t); \overline{\mathbf{f}}(\mathbf{r}, t) \odot \widetilde{\mathbf{x}}(0), \mathrm{diag}(\overline{\mathbf{g}}(\mathbf{r}, t)^2)),
\tag{31}
$$

where coefficients $\overline{\mathbf{f}}(\mathbf{r}, t), \overline{\mathbf{g}}(\mathbf{r}, t)^2$ are defined in Eq (20), $\odot$ indicates the entry-wise product, and diag converts a vector into a diagonal matrix.

Let $\mathbf{x}(0) \in \mathbb{R}^D$ be a real sample that is without noise and we perturb it as $\widetilde{\mathbf{x}}(0) = \mathbf{x}(0) + \boldsymbol{\epsilon} \odot \mathbf{r}, \boldsymbol{\epsilon} \sim \mathcal{N}(\mathbf{0}, \mathbf{I})$. We aim to first find the relation between risk-unaware kernel transition $p_{t|0}(\mathbf{x} \mid \mathbf{x}(0))$ and the expected risk-sensitive transition:

$$
\mathbb{E}_{\widetilde{\mathbf{x}}(0) \sim \mathcal{N}(\mathbf{x}(0), \mathrm{diag}(\mathbf{r}^2))} \big[ \widetilde{p}_{t|0,\mathbf{r}}(\mathbf{x} \mid \widetilde{\mathbf{x}}(0)) \big].
\tag{32}
$$

While the risk-unaware transition is simply a multivariate Gaussian:

$$
\mathcal{N}(\mathbf{x}; \overline{\mathbf{f}}(\mathbf{0}, t) \odot \mathbf{x}(0), \mathrm{diag}(\overline{\mathbf{g}}(\mathbf{0}, t)^2)),
\tag{33}
$$

we can expand the expected risk-sensitive transition as

$$
\int \mathcal{N}(\widetilde{\mathbf{x}}(0); \mathbf{x}(0), \mathrm{diag}(\mathbf{r}^2)) \widetilde{p}_{t|0,\mathbf{r}}(\mathbf{x} \mid \widetilde{\mathbf{x}}(0)) d\widetilde{\mathbf{x}}(0)
$$
$$
= \int \mathcal{N}(\widetilde{\mathbf{x}}(0); \mathbf{x}(0), \mathrm{diag}(\mathbf{r}^2)) \mathcal{N}(\mathbf{x}; \overline{\mathbf{f}}(\mathbf{r}, t) \odot \widetilde{\mathbf{x}}(0), \mathrm{diag}(\overline{\mathbf{g}}(\mathbf{r}, t)^2)) d\widetilde{\mathbf{x}}(0).
\tag{34}
$$

The second Gaussian distribution in the above equation can be reformulated as

$$
\mathcal{N}(\mathbf{x}; \cdot, \cdot) = (2\pi)^{-D/2} |\mathrm{diag}(\overline{\mathbf{g}}(\mathbf{r}, t)^2))|^{-1/2} \exp\Big( -\frac{1}{2}(\mathbf{x} - \overline{\mathbf{f}}(\mathbf{r}, t) \odot \widetilde{\mathbf{x}}(0))^\top \mathrm{diag}(\overline{\mathbf{g}}(\mathbf{r}, t)^2)^{-1}(\cdot) \Big)
$$
$$
= \frac{(2\pi)^{-D/2}}{|\mathrm{diag}(\overline{\mathbf{f}}(\mathbf{r}, t))|} \Big| \mathrm{diag}\Big(\frac{\overline{\mathbf{g}}(\mathbf{r}, t)^2}{\overline{\mathbf{f}}(\mathbf{r}, t)^2}\Big) \Big|^{-1/2} \exp\Big( -\frac{1}{2}\Big(\widetilde{\mathbf{x}}(0) - \frac{\mathbf{x}}{\overline{\mathbf{f}}(\mathbf{r}, t)}\Big)^\top \mathrm{diag}\Big(\frac{\overline{\mathbf{g}}(\mathbf{r}, t)^2}{\overline{\mathbf{f}}(\mathbf{r}, t)^2}\Big)^{-1}(\cdot) \Big)
$$
$$
= \frac{1}{|\mathrm{diag}(\overline{\mathbf{f}}(\mathbf{r}, t))|} \mathcal{N}\Big(\widetilde{\mathbf{x}}(0), \frac{\mathbf{x}}{\overline{\mathbf{f}}(\mathbf{r}, t)}, \mathrm{diag}\Big(\frac{\overline{\mathbf{g}}(\mathbf{r}, t)^2}{\overline{\mathbf{f}}(\mathbf{r}, t)^2}\Big)\Big).
$$
$$
\tag{35}
$$

where $|\cdot|$ means the determinant of a matrix. According to the product rule of multivariate Gaussians (Ahrendt, 2005), we can simplify the form of risk-sensitive transition as

$$
\int \mathcal{N}\Big(\widetilde{\mathbf{x}}(0); \mathbf{x}(0), \mathrm{diag}(\mathbf{r}^2))\Big) \frac{1}{|\mathrm{diag}(\overline{\mathbf{f}}(\mathbf{r}, t))|} \mathcal{N}\Big(\widetilde{\mathbf{x}}(0), \frac{\mathbf{x}}{\overline{\mathbf{f}}(\mathbf{r}, t)}, \mathrm{diag}\Big(\frac{\overline{\mathbf{g}}(\mathbf{r}, t)^2}{\overline{\mathbf{f}}(\mathbf{r}, t)^2}\Big)\Big) d\widetilde{\mathbf{x}}(0)
$$

$$
= \frac{1}{|\mathrm{diag}(\overline{\mathbf{f}}(\mathbf{r}, t))|} \int \mathcal{N}\Big(\frac{\mathbf{x}}{\overline{\mathbf{f}}(\mathbf{r}, t)}; \mathbf{x}(0), \mathrm{diag}\Big(\mathbf{r}^2 + \frac{\overline{\mathbf{g}}(\mathbf{r}, t)^2}{\overline{\mathbf{f}}(\mathbf{r}, t)^2}\Big)\Big) \mathcal{N}\Big(\widetilde{\mathbf{x}}(0); \cdot, \cdot\Big) d\widetilde{\mathbf{x}}(0) \qquad (36)
$$

$$
= \mathcal{N}\Big(\mathbf{x}; \overline{\mathbf{f}}(\mathbf{r}, t) \odot \mathbf{x}(0), \mathrm{diag}(\overline{\mathbf{f}}(\mathbf{r}, t)^2 \odot \mathbf{r}^2 + \overline{\mathbf{g}}(\mathbf{r}, t)^2)\Big).
$$

To let the two transitions equal, one must achieve the following two conditions:

$$
\overline{\mathbf{f}}(\mathbf{r}, t) = \overline{\mathbf{f}}(\mathbf{0}, t), \quad \overline{\mathbf{f}}(\mathbf{r}, t)^2 \odot \mathbf{r}^2 + \overline{\mathbf{g}}(\mathbf{r}, t)^2 = \overline{\mathbf{g}}(\mathbf{0}, t)^2. \qquad (37)
$$

The first condition indicates that the term $\overline{\mathbf{f}}$ is independent of risk $\mathbf{r}$, while the second condition implies that there might exist some iteration $t$ that the two transitions are not identical. Plus, because this term $\overline{\mathbf{g}}(\mathbf{r}, t)^2$ is always non-negative, we have

$$
\overline{\mathbf{g}}(\mathbf{r}, t)^2 = \max(\overline{\mathbf{g}}(\mathbf{0}, t)^2 - \overline{\mathbf{f}}(\mathbf{0}, t)^2 \odot \mathbf{r}^2, \mathbf{0}), \qquad (38)
$$

which means the risk-sensitive SDE has an initial period of pure contraction, but after that, its transition kernel is equal to the real one. Note that operation $\max$ is applied in an element-wise manner. $\overline{\mathbf{g}}(\mathbf{r}, t)^2$ might not be differentiable everywhere, but we can either locally smooth the curve or take its right derivative.

With the above derivation, we see that the following equation holds:

$$
p_{t|0}(\mathbf{x} \mid \mathbf{x}(0)) = \mathbb{E}_{\widetilde{\mathbf{x}}(0) \sim \mathcal{N}(\mathbf{x}(0), \mathrm{diag}(\mathbf{r}^2))}\big[\widetilde{p}_{t|0,\mathbf{r}}(\mathbf{x} \mid \widetilde{\mathbf{x}}(0))\big], \qquad (39)
$$

if Eq. (37) holds. For the left hand, we then have

$$
\mathbb{E}_{\mathbf{x}(0)}[p_{t|0}(\mathbf{x} \mid \mathbf{x}(0))] = \int p_0(\mathbf{x}(0)) p_{t|0}(\mathbf{x} \mid \mathbf{x}(0)) d\mathbf{x}(0) = p_t(\mathbf{x}). \qquad (40)
$$

Similarly, we apply the expectation operation $\mathbb{E}_{\mathbf{x}(0)}$ to the risk-sensitive transition:

$$
\mathbb{E}_{\mathbf{x}(0)}\big[\mathbb{E}_{\widetilde{\mathbf{x}}(0) \sim \mathcal{N}(\mathbf{x}(0), \mathrm{diag}(\mathbf{r}^2))}\big[\widetilde{p}_{t|0,\mathbf{r}}(\mathbf{x} \mid \widetilde{\mathbf{x}}(0))\big]\big]
$$

$$
= \int_{\mathbf{x}(0)} \int_{\widetilde{\mathbf{x}}(0)} p_0(\mathbf{x}(0)) p(\widetilde{\mathbf{x}}(0) \mid \mathbf{x}(0)) \widetilde{p}_{t|0,\mathbf{r}}(\mathbf{x} \mid \widetilde{\mathbf{x}}(0)) d\widetilde{\mathbf{x}}(0) d\mathbf{x}(0) \qquad (41)
$$

$$
= \int_{\mathbf{x}(0)} \widetilde{p}_{0,\mathbf{r}}(\widetilde{\mathbf{x}}(0)) \widetilde{p}_{t|0,\mathbf{r}}(\mathbf{x} \mid \widetilde{\mathbf{x}}(0)) d\widetilde{\mathbf{x}}(0) = \widetilde{p}_{t,\mathbf{r}}(\mathbf{x}).
$$

Therefore, we finally get $\widetilde{p}_{t,\mathbf{r}}(\mathbf{x}) = p_t(\mathbf{x})$ (i.e., perturbation stability) for $t$ in

$$
\mathcal{T}(\mathbf{r}) \equiv \{t \in [0, T] \mid \overline{\mathbf{f}}(\mathbf{r}, t) = \overline{\mathbf{f}}(\mathbf{0}, t), \overline{\mathbf{f}}(\mathbf{r}, t)^2 \odot \mathbf{r}^2 + \overline{\mathbf{g}}(\mathbf{r}, t)^2 = \overline{\mathbf{g}}(\mathbf{0}, t)^2\}. \qquad (42)
$$

Now, if we replace $\overline{\mathbf{f}}(\mathbf{0}, t), \overline{\mathbf{g}}(\mathbf{r}, t)$ by $\mathbf{u}(t), \mathbf{v}(\mathbf{r}, t)$, then we get the theorem proved. $\qquad \square$

*Remark* E.1. From this conclusion, we can easily see that, for a fixed coefficient $\mathbf{u}(t)$, the setup $\mathbf{v}(\mathbf{r}, t)^2 = \max(\mathbf{v}(\mathbf{r}, 0)^2 - \mathbf{r}^2 \odot \mathbf{u}(t)^2, \mathbf{0}^2)$ (where $\max$ is an element-wise operation) maximizes the period of perturbation stability: $|\mathcal{T}(\mathbf{r})|$, leading to optimal coefficients.

*Remark* E.2. It is apparent that *stability interval* $\mathcal{T}(\mathbf{r})$ shrinks as the risk $\mathbf{r}$ increases, indicating that a noisier sample is less valuable for training. Therefore, under Gaussian perturbation, the ratio $|\mathcal{T}(\mathbf{r})|/T$ reflects how much information is contained in noisy sample $(\widetilde{\mathbf{x}}(0), \mathbf{r})$.

*Remark* E.3. If one limits the perturbation noise to be isotropically Gaussian, then the risk vector $\mathbf{r}$ reduces to a scalar $r$, with $\mathcal{P}_\epsilon = \{\mathcal{N}(\mathbf{0}, r\mathbf{I}) \mid r > 0\}$. Eq. (29) and Eq. (30) in the theorem can also be simplified in terms of $\mathbf{r} = r\mathbf{1}, \mathbf{u}(t) = u(t)\mathbf{1}, \mathbf{v}(\mathbf{r}, t) = v(t)\mathbf{1}$.

In Sec. G, we apply the above theorem to diffusion models (e.g., Risk-sensitive VP SDE in Corollary G.2) and develop tools (e.g., simplified loss in Proposition G.1) for efficient optimization. In Appendix 5, our numerical experiments confirm the validity of the theorem (i.e., Fig. 2) and show that its suggested coefficients let diffusion models be robust to Gaussian-corrupted samples (i.e., Fig. 3).

# F MINIMUM INSTABILITY FOR GENERAL NOISES

The goal of this section is to find the optimal coefficients $\mathbf{f}(\mathbf{r}, t), \mathbf{g}(\mathbf{r}, t)$ of risk-sensitive SDE for general noise perturbation. Importantly, one will see that the property of *perturbation stability*: $\mathcal{S}_t(\mathbf{r}) = 0$ is not achievable in the case of non-Gaussian perturbation. We will first prove two other conclusions and then provide the main theorem.

## F.1 SPECTRAL REPRESENTATION

The first conclusion is to study the form of cumulant-generating function $\widetilde{\chi}_{t,\mathbf{r}}(\boldsymbol{\omega})$.

**Lemma F.1** (Spectral Form of the Marginal Density). *Suppose that we have a risk-sensitive SDE as defined in Eq. (5), then the cumulant-generating function of its marginal density $\widetilde{p}_{t,\mathbf{r}}(\mathbf{x})$ at time step $t \in [0, T]$: $\widetilde{\chi}_{t,\mathbf{r}}(\boldsymbol{\omega})$, has a form as*

$$\widetilde{\chi}_{t,\mathbf{r}}(\boldsymbol{\omega}) = \widetilde{\chi}_{0,\mathbf{r}}(\overline{\mathbf{f}}(\mathbf{r}, t) \odot \boldsymbol{\omega}) - \frac{1}{2} \sum_{i=1}^{D} w_i^2 \overline{g}_i(\mathbf{r}, t)^2, \tag{43}$$

*where terms $\overline{\mathbf{f}}(\mathbf{r}, t), \overline{\mathbf{g}}(\mathbf{r}, t)$ are defined in Eq. (20).*

*Proof.* Based on Fokker-Planck equation (Øksendal & Øksendal, 2003), the marginal distribution $\widetilde{p}_{t,\mathbf{r}}(\mathbf{x})$ at time step $t$ satisfies the following partial differential equation (PDE):

$$\frac{\partial \widetilde{p}_{t,\mathbf{r}}(\mathbf{x})}{\partial t} = -\sum_{i=1}^{D} f_i(\mathbf{r}, t) \widetilde{p}_{t,\mathbf{r}}(\mathbf{x}) - \sum_{i=1}^{D} x_i f_i(\mathbf{r}, t) \frac{\partial \widetilde{p}_{t,\mathbf{r}}(\mathbf{x}))}{\partial x_i} + \frac{1}{2} \sum_{i=1}^{D} g_i(\mathbf{r}, t)^2 \frac{\partial^2 \widetilde{p}_{t,\mathbf{r}}(\mathbf{x}))}{\partial x_i^2}. \tag{44}$$

Because $\widetilde{p}_{t,\mathbf{r}}(\mathbf{x})$ belongs to the function space $L^1(\mathbb{R}^D) = \{h : \mathbb{R}^D \to \mathbb{R} \mid \int |h(\mathbf{x})| d\mathbf{x} < \infty\}$, we can apply the continuous-time Fourier transform (Krantz, 2018),

$$\mathcal{F}(h(\mathbf{x}))(\boldsymbol{\omega}) = \int h(\mathbf{x}) \exp(\mathrm{i}\boldsymbol{\omega}^\top \mathbf{x}) d\mathbf{x}, \boldsymbol{\omega} \in \mathbb{R}^D,$$

where $\mathrm{i}$ is the imaginary unit, to the PDE as

$$\frac{\partial \mathcal{F}(\widetilde{p}_{t,\mathbf{r}}(\mathbf{x}))(\boldsymbol{\omega})}{\partial t} = -\sum_{i=1}^{D} f_i(\mathbf{r}, t) \mathcal{F}(\widetilde{p}_{t,\mathbf{r}}(\mathbf{x}))(\boldsymbol{\omega})$$

$$- \sum_{i=1}^{D} f_i(\mathbf{r}, t) \mathcal{F}\left(x_i \frac{\partial \widetilde{p}_{t,\mathbf{r}}(\mathbf{x})}{\partial x_i}\right)(\boldsymbol{\omega}) + \frac{1}{2} \sum_{i=1}^{D} g_i(\mathbf{r}, t)^2 \mathcal{F}\left(\frac{\partial^2 \widetilde{p}_{t,\mathbf{r}}(\mathbf{x})}{\partial x_i^2}\right)(\boldsymbol{\omega}). \tag{45}$$

According to some basic properties of the Fourier transform, we have

$$\begin{cases} \mathcal{F}\left(x_i \frac{\partial \widetilde{p}_{t,\mathbf{r}}(\mathbf{x})}{\partial x_i}\right)(\boldsymbol{\omega}) = -\mathrm{i}\frac{\partial}{\partial w_i}\left(\mathcal{F}\left(\frac{\partial \widetilde{p}_{t,\mathbf{r}}(\mathbf{x})}{\partial x_i}\right)(\boldsymbol{\omega})\right) = -\frac{\partial}{\partial w_i}\left(w_i \mathcal{F}(\widetilde{p}_{t,\mathbf{r}}(\mathbf{x})))(\boldsymbol{\omega})\right) \\ \mathcal{F}\left(\frac{\partial^2 \widetilde{p}_{t,\mathbf{r}}(\mathbf{x})}{\partial x_i^2}\right)(\boldsymbol{\omega}) = -w_i^2 \mathcal{F}(\widetilde{p}_{t,\mathbf{r}}(\mathbf{x}))(\boldsymbol{\omega}) \end{cases}, \tag{46}$$

By denoting $\mathcal{F}(\widetilde{p}_{t,\mathbf{r}}(\mathbf{x}))(\boldsymbol{\omega})$ as $\boldsymbol{\varphi}_{t,\mathbf{r}}(\boldsymbol{\omega})$, we can cast Eq. (44) as

$$\frac{\partial \boldsymbol{\varphi}_{t,\mathbf{r}}(\boldsymbol{\omega})}{\partial t} = \sum_{i=1}^{D} w_i f_i(\mathbf{r}, t) \frac{\partial \boldsymbol{\varphi}_{t,\mathbf{r}}(\boldsymbol{\omega})}{\partial w_i} - \left(\frac{1}{2} \sum_{i=1}^{D} w_i^2 g_i(\mathbf{r}, t)^2\right) \boldsymbol{\varphi}_{t,\mathbf{r}}(\boldsymbol{\omega}). \tag{47}$$

In terms of the characteristic curves, we set ordinary differential equations (ODE) as

$$\frac{dt}{ds} = 1, \quad \frac{dw_i}{ds} = -w_i f_i(\mathbf{r}, t), i \in [1, D] \bigcup \mathbb{N}, \quad \frac{d\boldsymbol{\varphi}}{ds} = \left(-\frac{1}{2} \sum_{i=1}^{D} w_i^2 g_i(\mathbf{r}, t)^2\right) \boldsymbol{\varphi}. \tag{48}$$

where $s$ and $\mathbb{N}$ are respectively a dummy variable and the set of all natural numbers. Considering the initial condition: $t(0) = 0, w_i(0) = \xi_i, \boldsymbol{\varphi}(0) = \boldsymbol{\varphi}_{0,\mathbf{r}}(\boldsymbol{\xi}), \boldsymbol{\xi} = [\xi_1, \xi_2, \cdots, \xi_D]^\top$, the solutions to these ODE are formulated as

$$t(s) = s, \quad w_i(s) = \xi_i \overline{f}_i(\mathbf{r}, s)^{-1}, \quad \boldsymbol{\varphi}(s) = \boldsymbol{\varphi}_{0,\mathbf{r}}(\boldsymbol{\xi}) \exp\left(-\frac{1}{2} \sum_{i=1}^{D} \xi_i^2 \frac{\overline{g}_i(\mathbf{r}, s)^2}{\overline{f}_i(\mathbf{r}, s)^2}\right), \tag{49}$$

where coefficient functions $\bar{f}_i(\mathbf{r}, s), \bar{g}_i(\mathbf{r}, s)$ are of the forms as

$$\bar{f}_i(\mathbf{r}, s) = \exp\left(\int_0^s f_i(\mathbf{r}, s')ds'\right), \quad \bar{g}_i(\mathbf{r}, s) = \bar{f}_i(\mathbf{r}, s)\sqrt{\int_0^t \frac{g_i(\mathbf{r}, s')^2}{\bar{f}_i(\mathbf{r}, s')^2}ds'}. \tag{50}$$

Based on the above results, we can get the solution of $\boldsymbol{\varphi}_{t,\mathbf{r}}(\boldsymbol{\omega})$ as

$$\boldsymbol{\varphi}_{t,\mathbf{r}}(\boldsymbol{\omega}) = \boldsymbol{\varphi}_{0,\mathbf{r}}(\bar{\mathbf{f}}(\mathbf{r}, t) \odot \boldsymbol{\omega})\exp(-\frac{1}{2}\sum_{i=1}^D w_i^2 \bar{g}_i(\mathbf{r}, t)^2), \tag{51}$$

in which $\bar{\mathbf{f}}(\mathbf{r}, t) = [\bar{f}_1(\mathbf{r}, t), \bar{f}_2(\mathbf{r}, t), \cdots, \bar{f}_D(\mathbf{r}, t)]^\top$ and $\bar{\mathbf{g}}(\mathbf{r}, t) = [\bar{g}_1(\mathbf{r}, t), \bar{g}_2(\mathbf{r}, t), \cdots, \bar{g}_D(\mathbf{r}, t)]^\top$. The lemma is proved by taking logarithms on both sides of the equation. $\qquad\square$

We can also get a similar conclusion for the term $\boldsymbol{\chi}_t(\boldsymbol{\omega})$ by setting $\mathbf{r} = \mathbf{0}$.

### F.2 Necessary Condition for Achieving Stability

The second conclusion is about the necessary condition to achieve perturbation stability.

**Proposition F.1** (Necessary Condition for Perturbation Stability). *Given the definition (i.e., Eq. (5)) of risk-sensitive SDE, then a necessary condition for it to have the property of perturbation stability is that the perturbation results from diagonal Gaussian noises.*

*Proof.* Let the noise distribution be of a free form $q(\boldsymbol{\epsilon}) : \int q(\boldsymbol{\epsilon})d\boldsymbol{\epsilon} = 1; q(\boldsymbol{\epsilon}) > 0, \forall\boldsymbol{\epsilon} \in \mathbb{R}^D$, then the distribution of clean data $p_0(\mathbf{x})$ will be perturbed into a noisy one

$$\widetilde{p}_{0,\mathbf{r}}(\mathbf{x}) = \int p_0(\mathbf{x}')q(\mathbf{x} - \mathbf{x}')d\mathbf{x}' = (p_0(\cdot) * q(\cdot))(\mathbf{x}), \tag{52}$$

where $*$ represents the convolution operation. Through Fourier transform, we have

$$\boldsymbol{\varphi}_{0,\mathbf{r}}(\boldsymbol{\omega}) = \mathcal{F}(p_{0,\mathbf{r}}(\mathbf{x}))(\boldsymbol{\omega}) = \mathcal{F}(p_{0,\mathbf{r}}(\mathbf{x}))(\boldsymbol{\omega}) \cdot \mathcal{F}(q(\mathbf{x}))(\boldsymbol{\omega}) = \boldsymbol{\varphi}_{0,\mathbf{r}}(\boldsymbol{\omega})\phi(\boldsymbol{\omega}), \tag{53}$$

where $\phi(\boldsymbol{\omega}) := \mathcal{F}(q(\mathbf{x}))(\boldsymbol{\omega})$. Here we also suppose that $q(\boldsymbol{\epsilon}) \in L^1(\mathbb{R}^D)$.

With the above results and applying the Lemma F.1, we can get the form of risk-unaware marginal distribution $p_{t,\mathbf{r}}(\mathbf{x})$ in the frequency domain as

$$\boldsymbol{\chi}_t(\boldsymbol{\omega}) = \chi_0(\bar{\mathbf{f}}(\mathbf{0}, t) \odot \boldsymbol{\omega}) - \frac{1}{2}\sum_{i=1}^D w_i^2 \bar{g}_i(\mathbf{0}, t)^2, \tag{54}$$

and the one for risk-sensitive marginal distribution $\widetilde{p}_{t,\mathbf{r}}(\mathbf{x})$ is as

$$\widetilde{\boldsymbol{\chi}}_{t,\mathbf{r}}(\boldsymbol{\omega}) = \chi_{0,\mathbf{r}}(\bar{\mathbf{f}}(\mathbf{r}, t) \odot \boldsymbol{\omega}) + \chi_q(\bar{\mathbf{f}}(\mathbf{r}, t) \odot \boldsymbol{\omega}) - \frac{1}{2}\sum_{i=1}^D w_i^2 \bar{g}_i(\mathbf{r}, t)^2, \tag{55}$$

where $\chi_q(\cdot)$ are the cumulant-generating function of noise distribution $q(\boldsymbol{\epsilon})$.

Because the Fourier transform $\mathcal{F}$ is injective in the domain of definition $L^1(\mathbb{R}^D)$, the property $\widetilde{p}_{t,\mathbf{r}}(\mathbf{x}) = p_t(\mathbf{x})$ is equivalent to the condition $\boldsymbol{\chi}_t(\boldsymbol{\omega}) = \widetilde{\boldsymbol{\chi}}_{t,\mathbf{r}}(\boldsymbol{\omega})$. Considering function $p_0(\mathbf{x}) \in L^1(\mathbb{R}^D)$ and variable $\boldsymbol{\omega} \in \mathbb{R}^D$ are arbitrarily selected, the above two equations are equivalent indicate that the below two conditions are satisfied:

$$\bar{\mathbf{f}}(\mathbf{0}, t) = \bar{\mathbf{f}}(\mathbf{r}, t), \quad \phi(\boldsymbol{\omega}) = \exp\left(-\frac{1}{2}\boldsymbol{\omega}^\top\text{diag}\left(\frac{\bar{\mathbf{g}}(\mathbf{0}, t)^2}{\bar{\mathbf{f}}(\mathbf{0}, t)^2} - \frac{\bar{\mathbf{g}}(\mathbf{r}, t)^2}{\bar{\mathbf{f}}(\mathbf{r}, t)^2}\right)\boldsymbol{\omega}\right). \tag{56}$$

The shape of characteristic function $\phi(\boldsymbol{\omega})$ indicates that its form $q(\mathbf{x})$ in the spatial domain is a multivariate Gaussian (DasGupta & DasGupta, 2011), with the following moments:

$$\mathbb{E}_{\mathbf{x}\sim q(\mathbf{x})}[\mathbf{x}] = \mathbf{0}, \quad \text{diag}(\mathbf{r}^2) := \mathbb{E}[\mathbf{x}\mathbf{x}^\top] = \text{diag}\left(\frac{\bar{\mathbf{g}}(\mathbf{0}, t)^2}{\bar{\mathbf{f}}(\mathbf{0}, t)^2} - \frac{\bar{\mathbf{g}}(\mathbf{r}, t)^2}{\bar{\mathbf{f}}(\mathbf{r}, t)^2}\right). \tag{57}$$

Therefore, perturbation stability is only possible for Gaussian noise $\mathcal{N}(\mathbf{0}, \mathrm{diag}(\mathbf{r}^2)), \mathbf{r} \in \mathbb{R}^D$. To achieve this stability, we have to pick up a risk-sensitive SDE of the form:

$$\begin{cases} d\mathbf{x}(t) = \left(\frac{d\ln\overline{\mathbf{f}}(t)}{dt} \odot \mathbf{x}(t)\right)dt + \left(\overline{\mathbf{f}}(t)^2 \odot \frac{d}{dt}\left(\frac{\overline{\mathbf{g}}(\mathbf{r},t)^2}{\overline{\mathbf{f}}(t)^2}\right)\right)d\boldsymbol{\omega}(t) \\ \overline{\mathbf{g}}(\mathbf{r},t)^2 = \max(\overline{\mathbf{g}}(\mathbf{0},t)^2 - \mathbf{r}^2 \odot \overline{\mathbf{f}}(t)^2, \mathbf{0}) \end{cases}, \tag{58}$$

which is risk-unaware for iteration $t$ in $\{t \in [1,T] \mid \overline{\mathbf{g}}(\mathbf{r},t)^2 = \overline{\mathbf{g}}(\mathbf{0},t)^2 - \mathbf{r}^2 \odot \overline{\mathbf{f}}(t)^2\}$. $\qquad\square$

Paired with Theorem E.1, this proposition indicates that the necessary and sufficient conditions for a risk-sensitive SDE to achieve perturbation stability $\widetilde{p}_{t,\mathbf{r}}(\mathbf{x}) = p_t(\mathbf{x})$ are: 1) noises follow diagonal Gaussian distributions; 2) the time step $t$ is within the stability interval.

### F.3 Solution for General Noises

Provided with former two conclusions, we can now prove the below main theorem.

**Theorem F.1** (General Theory of Perturbation Stability). *Suppose we have a family of continuous noise distributions: $\mathcal{P}_\epsilon = \{\rho_{\mathbf{r}}(\boldsymbol{\epsilon}) : \mathbb{R}^D \to \mathbb{R}^+, \int \rho_{\mathbf{r}}(\boldsymbol{\epsilon})d\boldsymbol{\epsilon} = 1 \mid \mathbf{r} \neq \mathbf{0}\}$, controlled by a risk vector $\mathbf{r}$, then the optimal coefficients of the risk-sensitive SDE (as defined in Eq. (5)) that minimizes the perturbation instability $\mathcal{S}_t(\mathbf{r})$ satisfy*

$$\mathbf{v}(\mathbf{r},t)^2 = \max\left(\mathbf{0}, \mathbf{v}(\mathbf{0},t)^2 + \boldsymbol{\Psi}(\mathbf{u}(t),\mathbf{r})\right), \tag{59}$$

*in which the term $\boldsymbol{\Psi}(\cdot)$ is defined as*

$$\boldsymbol{\Psi}(\mathbf{u}(t),\mathbf{r}) = 2\left(\int \Omega(\mathbf{y})[\mathbf{y}\mathbf{y}^\top]^2 d\mathbf{y}\right)^{-1} \otimes \left(\int \Omega(\mathbf{y})\ln\left|\exp\left(\boldsymbol{\chi}_{\mathbf{r}}(\mathbf{u}(t) \odot \mathbf{y})\right)\right|[\mathbf{y}]^2 d\mathbf{y}\right) \tag{60}$$

*where $\otimes$ stands for a matrix multiplication, $\boldsymbol{\chi}_{\mathbf{r}}(\mathbf{y})$ is the cumulant-generating function of the noise distribution $\rho_{\mathbf{r}}(\boldsymbol{\epsilon})$, and $\mathbf{u}(t), \mathbf{v}(\mathbf{r},t)$ satisfy the same conditions as in Eq. (29). Importantly, the property of perturbation stability $\mathcal{S}_t(\mathbf{r}) = \mathbf{0}$ is only possible to achieve for Gaussian perturbations of a diagonal form: $\rho_{\mathbf{r}}(\boldsymbol{\epsilon}) = \mathcal{N}(\boldsymbol{\epsilon}; \mathbf{0}, \mathrm{diag}(\mathbf{r}^2))$.*

*Proof.* Based on Lemma F.1 and Proposition F.1, we can see there is no appropriate risk-sensitive SDE to fully neutralize the negative impact of non-Gaussian noise distribution $q(\boldsymbol{\epsilon})$. Therefore, we aim to find an optimal (though not perfect) risk-sensitive SDE in this regard. Note that $p_0(\mathbf{x})$ is arbitrarily selected in space $L^1(\mathbb{R}^D)$, so condition $\overline{\mathbf{f}}(\mathbf{r},t) = \overline{\mathbf{f}}(\mathbf{0},t)$ still needs to hold. For $\overline{\mathbf{g}}(\mathbf{0},t)$, we first consider the objective function to optimize:

$$\mathcal{O}_t = \int \Omega(\boldsymbol{\omega})\left|\widetilde{\chi}_{t,\mathbf{r}}(\boldsymbol{\omega}) - \chi_t(\boldsymbol{\omega})\right|^2 d\boldsymbol{\omega}, \tag{61}$$

where $|\cdot|$ represents the magnitude of a complex number and $\Omega(\boldsymbol{\omega}) : \mathbb{R}^D \to \mathbb{R}^+$ is a predefined weight function. Considering Eq. (54) and Eq. (55), we have

$$\mathcal{O}_t = \int \Omega(\boldsymbol{\omega})\left|\frac{1}{2}\sum_{i=1}^{D}\omega_i^2 \overline{g}_i(\mathbf{0},t)^2 - \frac{1}{2}\sum_{i=1}^{D}\omega_i^2 \overline{g}_i(\mathbf{r},t)^2 + \boldsymbol{\chi}_{\mathbf{r}}(\overline{\mathbf{f}}(\mathbf{r},t) \odot \boldsymbol{\omega})\right|^2 d\boldsymbol{\omega}$$

$$= \int \Omega(\boldsymbol{\omega})\left|\frac{1}{2}\left\langle\boldsymbol{\omega}^2, \overline{\mathbf{g}}(\mathbf{0},t)^2 - \overline{\mathbf{g}}(\mathbf{r},t)^2\right\rangle + \ln\left|\boldsymbol{\phi}_{\mathbf{r}}(\overline{\mathbf{f}}(\mathbf{r},t) \odot \boldsymbol{\omega})\right| + \mathrm{i}\cdot\arg\left(\boldsymbol{\phi}_{\mathbf{r}}(\overline{\mathbf{f}}(\mathbf{r},t) \odot \boldsymbol{\omega})\right)\right|^2 d\boldsymbol{\omega}$$

$$= \int \Omega(\boldsymbol{\omega})\left(\frac{1}{2}\left\langle\boldsymbol{\omega}^2, \overline{\mathbf{g}}(\mathbf{0},t)^2 - \overline{\mathbf{g}}(\mathbf{r},t)^2\right\rangle + \ln\left|\boldsymbol{\phi}_{\mathbf{r}}(\overline{\mathbf{f}}(\mathbf{r},t) \odot \boldsymbol{\omega})\right|\right)^2 d\boldsymbol{\omega} + \int \Omega(\boldsymbol{\omega})\arg\left(\cdot\right)^2 d\boldsymbol{\omega}, \tag{62}$$

where $\arg(\cdot)$ and $<\cdot,\cdot>$ respectively represent the argument of a complex number and the inner product of two vectors. Plus, $\boldsymbol{\chi}_{\mathbf{r}}(\cdot)$ and $\boldsymbol{\phi}_{\mathbf{r}}(\cdot)$ are respectively the cumulant-generating and characteristic functions of noise distribution $\rho_{\mathbf{r}}(\boldsymbol{\epsilon})$.

Now, we denote $\overline{\mathbf{g}}(\mathbf{0},t)^2 - \overline{\mathbf{g}}(\mathbf{r},t)^2$ as a dummy variable $\mathbf{y} = [y_1, y_2, \cdots, y_D]^\top$. Then, we compute the derivative of objective $\mathcal{O}_t$ with respect to every input entry $y_j, j \in [1,D]$:

$$\frac{d\mathcal{O}_t}{dy_j} = \int \left(\Omega(\boldsymbol{\omega}) \cdot \omega_j^2\left(\frac{1}{2}\left\langle\boldsymbol{\omega}^2, \mathbf{y}\right\rangle + \ln\left|\boldsymbol{\phi}_{\mathbf{r}}(\overline{\mathbf{f}}(\mathbf{r},t) \odot \boldsymbol{\omega})\right|\right)\right)d\boldsymbol{\omega}. \tag{63}$$

Through setting $d\mathcal{O}_t/dy_j = 0$ for every $j$ in $[1, D]$, we can get

$$\left\langle \Big[ \int \Omega(\boldsymbol{\omega})\omega_j^2\omega_i^2 d\boldsymbol{\omega} \Big]_{i\in[1,D]}^\top, \mathbf{y} \right\rangle = -2 \int \Omega(\boldsymbol{\omega})w_j^2 \ln\Big|\boldsymbol{\phi}_{\mathbf{r}}(\bar{\mathbf{f}}(\mathbf{r}, t)\odot\boldsymbol{\omega})\Big| d\boldsymbol{\omega}, \tag{64}$$

where $[\cdot]_{i\in[1,D]}^\top$ represents some column vector. By combining all results, we have

$$\Big[ \int \Omega(\boldsymbol{\omega})\omega_i^2\omega_j^2 d\boldsymbol{\omega} \Big]_{i,j\in[1,D]}\mathbf{y} = -2\Big[ \int \Omega(\boldsymbol{\omega}) \ln\Big|\boldsymbol{\phi}_{\mathbf{r}}(\bar{\mathbf{f}}(\mathbf{r}, t)\odot\boldsymbol{\omega})\Big|\omega_i^2 d\boldsymbol{\omega} \Big]_{i\in[1,D]}^\top, \tag{65}$$

where $[\cdot]_{i,j\in[1,D]}$ represents some matrix. For notational convenience, we denote

$$\begin{cases} \Big[ \int \Omega(\boldsymbol{\omega})\omega_i^2\omega_j^2 d\boldsymbol{\omega} \Big]_{i,j\in[1,D]} = \int \Omega(\boldsymbol{\omega})[\boldsymbol{\omega}\boldsymbol{\omega}^\top]^2 d\boldsymbol{\omega} \\ \Big[ \int \Omega(\boldsymbol{\omega}) \ln\Big|\boldsymbol{\phi}_{\mathbf{r}}(\cdot)\Big|\omega_i^2 d\boldsymbol{\omega} \Big]_{i\in[1,D]}^\top = \int \Omega(\boldsymbol{\omega}) \ln|\boldsymbol{\phi}_{\mathbf{r}}(\cdot)|[\boldsymbol{\omega}]^2 d\boldsymbol{\omega} \end{cases}. \tag{66}$$

Considering that $\mathbf{y} = \bar{\mathbf{g}}(\mathbf{0}, t)^2 - \bar{\mathbf{g}}(\mathbf{r}, t)^2$ and $\bar{\mathbf{g}}(\mathbf{r}, t)^2$ is always non-negative, we get

$$\bar{\mathbf{g}}(\mathbf{r}, t)^2 = \max\left( \bar{\mathbf{g}}(\mathbf{0}, t)^2 + 2\Big( \int \Omega(\boldsymbol{\omega})[\boldsymbol{\omega}\boldsymbol{\omega}^\top]^2 d\boldsymbol{\omega} \Big)^{-1} \Big( \int \Omega(\boldsymbol{\omega}) \ln\Big|\boldsymbol{\phi}_{\mathbf{r}}(\bar{\mathbf{f}}(\mathbf{r}, t)\odot\boldsymbol{\omega})\Big|[\boldsymbol{\omega}]^2 d\boldsymbol{\omega} \Big), \mathbf{0} \right). \tag{67}$$

For simplification, we introduce a new symbol $\boldsymbol{\Psi}$ as

$$\begin{aligned} \boldsymbol{\Psi}(\mathbf{f}(\mathbf{r}, t), \mathbf{r}) &= 2\Big( \int \Omega(\boldsymbol{\omega})[\boldsymbol{\omega}\boldsymbol{\omega}^\top]^2 d\boldsymbol{\omega} \Big)^{-1} \Big( \int \Omega(\boldsymbol{\omega}) \ln\Big|\boldsymbol{\phi}_{\mathbf{r}}(\bar{\mathbf{f}}(\mathbf{r}, t)\odot\boldsymbol{\omega})\Big|[\boldsymbol{\omega}]^2 d\boldsymbol{\omega} \Big) \\ &= 2\Big( \int \Omega(\boldsymbol{\omega})[\boldsymbol{\omega}\boldsymbol{\omega}^\top]^2 d\boldsymbol{\omega} \Big)^{-1} \Big( \int \Omega(\boldsymbol{\omega}) \ln\Big|\exp\big(\boldsymbol{\chi}_{\mathbf{r}}(\bar{\mathbf{f}}(\mathbf{r}, t)\odot\boldsymbol{\omega})\big)\Big|[\boldsymbol{\omega}]^2 d\boldsymbol{\omega} \Big). \end{aligned} \tag{68}$$

As a result, the optimal coefficient $\bar{\mathbf{g}}(\mathbf{r}, t)$ can be formulated as

$$\bar{\mathbf{g}}(\mathbf{r}, t)^2 = \max\Big( \bar{\mathbf{g}}(\mathbf{0}, t)^2 + \boldsymbol{\Psi}(\mathbf{f}(\mathbf{r}, t), \mathbf{r}), \mathbf{0} \Big), \tag{69}$$

which finally proves the theorem. $\qquad\square$

With the above theorem, we can find the optimal risk-sensitive SDE for non-Gaussian perturbation, though its coefficient $\mathbf{g}(\mathbf{r}, t)$ might have a rather complicated form. In Sec. G, we apply this theorem to Cauchy perturbation under the architecture of VE SDE (i.e., Corollary G.3). Our numerical experiments in Appendix 5 also show that such a risk-sensitive VE SDE are very robust for optimization with Cauchy-corrupted samples (i.e., Fig. 4).

## G  APPLICATIONS TO DIFFUSION MODELS

In this section, we aim to apply *risk-sensitive SDE* and our developed theorems to the practical implementations of diffusion models. We will show how to extend a vanilla diffusion model to its risk-sensitive versions under different noise perturbations and provide an efficient algorithm for optimizing the score-based model with risk-sensitive SDE.

### G.1  EXTENSIONS UNDER GAUSSIAN PERTURBATION

In terms of Theorem E.1, we have the following corollaries that extend two widely used diffusion models to their risk-sensitive versions. One of them is VE SDE (Song et al., 2021).

**Corollary G.1** (Risk-sensitive VE SDE for Gaussian Noises). *Under the assumption of Gaussian perturbation, the risk-sensitive SDE (as defined by Eq. (5)) for VE SDE, a commonly used risk-unaware diffusion model, is parameterized as follows:*

$$\mathbf{f}(\mathbf{r}, t) = \mathbf{0}, \quad \mathbf{g}(\mathbf{r}, t) = \mathbb{1}(\sigma(t)^2\mathbf{1} \gtrsim \sigma(0)^2\mathbf{1} + \mathbf{r}^2)\sqrt{\frac{d\sigma(t)^2}{dt}},$$

*where $\mathbb{1}(\cdot)$ is an element-wise indicator function, $\gtrsim$ represents the element-wise greater-than sign. The stability interval $\mathcal{T}(\mathbf{r})$ in this case is*

$$\mathcal{T}(\mathbf{r}) = \left\{ t \in [0, T] \mid \sigma(t)^2 - \max_{i \in [1, D]} r_i^2 > 0 \right\}.$$

*In particular, for the case of zero risk $\mathbf{r} = \mathbf{0}$, the above risk-sensitive SDE reduces to the vanilla VE SDE, with $\mathbf{f}(\mathbf{0}, t) = \mathbf{0}$, $\mathbf{g}(\mathbf{0}, t) = \sqrt{d\sigma(t)^2/dt}\mathbf{1}$, and $\mathcal{T} = [0, T]$.*

*Proof.* For VE SDE, its risk-unaware kernel is as

$$\begin{cases} p_{t|0}(\mathbf{x}(t) \mid \mathbf{x}(0)) = \mathcal{N}(\mathbf{x}(t); \overline{\mathbf{f}}(t) \odot \mathbf{x}(0), \mathrm{diag}(\overline{\mathbf{g}}(\mathbf{0}, t)^2)) \\ \overline{\mathbf{f}}(t) = \mathbf{1}, \quad \overline{\mathbf{g}}(\mathbf{0}, t)^2 = (\sigma(t)^2 - \sigma(0)^2)\mathbf{1} \end{cases},$$

where $\sigma(t) : [0, T] \rightarrow \mathbb{R}^+$ is an exponentially increasing function. Considering our previous conclusions (i.e., Eq. (37) and Eq. (28)) for Gaussian noises, the risk-unaware and risk-sensitive SDEs for VE SDE are parameterized as follows:

$$\begin{cases} d\mathbf{x}(t) = \mathbf{g}(\mathbf{0}, t) \odot d\mathbf{w}(t), \mathbf{g}(\mathbf{0}, t) = \mathbf{1}\sqrt{\dfrac{d\sigma(t)^2}{dt}} \\ d\widetilde{\mathbf{x}}(t) = \mathbf{g}(\mathbf{r}, t) \odot d\mathbf{w}(t), \mathbf{g}(\mathbf{r}, t) = \mathbb{1}(\sigma(t)^2\mathbf{1} \gtrsim \sigma(0)^2\mathbf{1} + \mathbf{r}^2)\sqrt{\dfrac{d\sigma(t)^2}{dt}} \end{cases},$$

where $\mathbb{1}(\cdot)$ is an element-wise indicator function. The risk-sensitive SDE of VE SDE is of perturbation stability at iteration $t \in [0, T]$ iff the vector $\sigma(t)^2\mathbf{1} - \mathbf{r}^2$ is positive in each entry. $\square$

The other is VP SDE (Song et al., 2021), the continuous version of DDPM (Ho et al., 2020).

**Corollary G.2** (Risk-sensitive VP SDE for Gaussian Noises). *Under the assumption of Gaussian perturbation, the risk-sensitive SDE for VP SDE is parameterized as follows:*

$$\mathbf{f}(\mathbf{r}, t) = -\frac{1}{2}\beta(t)\mathbf{1}, \quad \mathbf{g}(\mathbf{r}, t) = \mathbb{1}\left(\mathbf{1} \gtrsim (\mathbf{1} + \mathbf{r}^2)\alpha(t)\right)\sqrt{\beta(t)},$$

*where $\alpha(t) = \exp(-\int_0^t \beta(s)ds)$. The stability interval $\mathcal{T}(\mathbf{r})$ in this case is*

$$\mathcal{T}(\mathbf{r}) = \{t \in [0, T] \mid \alpha(t)^{-1} > 1 + \max_{1 \leqslant j \leqslant D} r_j^2\}.$$

*As expected, for the situation with zero risk $\mathbf{r} = \mathbf{0}$, the risk-sensitive SDE reduces to an ordinary VP SDE, with $\mathbf{f}(\mathbf{0}, t) = -\frac{1}{2}\beta(t)\mathbf{1}$, $\mathbf{g}(\mathbf{0}, t) = \sqrt{\beta(t)}\mathbf{1}$.*

*Proof.* For VP SDE, its risk-unaware kernel is as

$$\begin{cases} p_{t|0}(\mathbf{x}(t) \mid \mathbf{x}(0)) = \mathcal{N}(\mathbf{x}(t); \overline{\mathbf{f}}(t) \odot \mathbf{x}(0), \mathrm{diag}(\overline{\mathbf{g}}(\mathbf{0}, t)^2)) \\ \overline{\mathbf{f}}(t) = \mathbf{1}\exp\left(-\dfrac{1}{2}\int_0^t \beta(s)ds\right) \\ \overline{\mathbf{g}}(\mathbf{0}, t)^2 = \mathbf{1} - \mathbf{1}\exp\left(-\int_0^t \beta(s)ds\right) \end{cases},$$

where $\beta(t) : [0, T] \rightarrow \mathbb{R}^+$ is a predefined curve. Similar to our discussion about VE SDE, by using Eq. (37) and Eq. (28), the risk-unaware and risk-sensitive SDEs for VP SDE are as

$$\begin{cases} d\mathbf{x}(t) = (\mathbf{f}(t) \odot \mathbf{x}(t))dt + \mathbf{g}(\mathbf{0}, t) \odot d\mathbf{w}(t), \mathbf{f}(t) = -\dfrac{1}{2}\beta(t)\mathbf{1}, \mathbf{g}(\mathbf{0}, t) = \sqrt{\beta(t)}\mathbf{1} \\ d\widetilde{\mathbf{x}}(t) = (\mathbf{f}(t) \odot \widetilde{\mathbf{x}}(t))dt + \mathbf{g}(\mathbf{r}, t) \odot d\mathbf{w}(t), \mathbf{g}(\mathbf{r}, t) = \mathbb{1}\left(\mathbf{1} \gtrsim (\mathbf{1} + \mathbf{r}^2)\exp\left(-\int_0^t \beta(s)ds\right)\right)\sqrt{\beta(t)} \end{cases}.$$

The stability interval $\mathcal{T}(\mathbf{r})$ is as $\{t \in [0, T] \mid \exp(\int_0^t \beta(s)ds) > 1 + \max_{1 \leqslant j \leqslant D} r_j^2\}$. $\square$

One can apply the two corollaries to isotropic Gaussian noises by simply setting $\mathbf{r} = r\mathbf{1}$, such that coefficients $\mathbf{f}(t), \mathbf{g}(\mathbf{r}, t)$ will reduce to scalar functions. In Appendix 5, our numerical experiments confirm that Risk-sensitive VP SDE can achieve stability under Gaussian perturbation (i.e., Fig. 2) and it is indeed robust to Gaussian-corrupted samples (i.e., Fig. 3).

### G.2 OPTIMIZATION AND SAMPLING

In this part, we provide essential elements for applying *risk-sensitive diffusion models* in practice, including the loss function, optimization algorithm, and sampling algorithm. Before diving into the details, we will first prove a lemma about the stability condition.

**Lemma G.1** ("Not Instable" means "Stable"). *Provided with the definition of instability measure* $\mathcal{S}_t(\mathbf{r})$ *in Eq. (6) and suppose that distributions* $p_0(\mathbf{x}), \rho_{\mathbf{r}}(\boldsymbol{\epsilon})$ *are continuous. If we have* $\mathcal{S}_t(\mathbf{r}) = 0$, *then* $\widetilde{\boldsymbol{\chi}}_{t,\mathbf{r}}(\mathbf{y}) = \boldsymbol{\chi}_t(\mathbf{y}), \forall \mathbf{y} \in \mathbb{R}^D$ *and* $\widetilde{p}_{t,\mathbf{r}}(\mathbf{x}) = p_t(\mathbf{x}), \forall \mathbf{x} \in \mathbb{R}^D$ *both hold.*

*Proof.* We prove the first equality by contradiction. Suppose that $\widetilde{\boldsymbol{\chi}}_{t,\mathbf{r}}(\mathbf{y}) \neq \boldsymbol{\chi}_t(\mathbf{y})$ for some point $\mathbf{y}' \in \mathbb{R}^D$, then there is a closed region $\mathbf{y}' \in U \subset \mathbb{R}^D$ that satisfies $|U| > 0$ and $\widetilde{\boldsymbol{\chi}}_{t,\mathbf{r}}(\mathbf{y}) \neq \boldsymbol{\chi}_t(\mathbf{y}), \forall \mathbf{y} \in U$ because probabilistic densities $\widetilde{\boldsymbol{\chi}}_{t,\mathbf{r}}(\mathbf{y}), \boldsymbol{\chi}_t(\mathbf{y})$ are both continuous everywhere. Considering that the region $U$ is a closed set, we have the below inequalities

$$\zeta_1 = \min_{\mathbf{y} \in U} |\widetilde{\boldsymbol{\chi}}_{t,\mathbf{r}}(\mathbf{y}) - \boldsymbol{\chi}_t(\mathbf{y})| > 0, \quad \zeta_2 = \min_{\mathbf{y} \in U} \Omega(\mathbf{y}) > 0. \tag{70}$$

With these results, we further have

$$\mathcal{S}_t(\mathbf{r}) \geqslant \int_U \Omega(\mathbf{y}) |\widetilde{\boldsymbol{\chi}}_{t,\mathbf{r}}(\mathbf{y}) - \boldsymbol{\chi}_t(\mathbf{y})|^2 d\mathbf{y} > \int_U \zeta_2 \zeta_1^2 = |U| \zeta_2 \zeta_1^2 > 0, \tag{71}$$

which contradict the precondition $\mathcal{S}_t(\mathbf{r}) = 0$. Hence, we have $\widetilde{\boldsymbol{\chi}}_{t,\mathbf{r}}(\mathbf{y}) = \boldsymbol{\chi}_t(\mathbf{y}), \forall \mathbf{y} \in \mathbb{R}^D$. We can also immediately get the second equality proved since probability densities are uniquely determined by their cumulant-generating functions. $\square$

It is trivial that stability condition $\widetilde{p}_{t,\mathbf{r}}(\mathbf{x}) = p_t(\mathbf{x})$ indicates zero instability measure $\mathcal{S}_t(\mathbf{r}) = 0$. Therefore, an implication of the above lemma is that the two conditions are in fact equivalent if the sample $p_0(\mathbf{x})$ and noise distributions $\rho_{\mathbf{r}}(\boldsymbol{\epsilon})$ are both continuous.

**Risk-free loss for noisy samples.** In the following proposition, we derive the loss function for risk-sensitive SDE to robustly optimize the score-based model $\mathbf{s}_{\boldsymbol{\theta}}(\mathbf{x}, t)$ with noisy sample $(\widetilde{\mathbf{x}}(0), \mathbf{r})$. We also further simplify the loss for efficient computation.

**Proposition G.1** (Risk-free Loss for Robust Optimization). *Suppose that the risky sample* $(\widetilde{\mathbf{x}}(0) = \mathbf{x}(0) + \boldsymbol{\epsilon}, \mathbf{r})$ *is generated from clean sample* $(\mathbf{x}(0), \mathbf{r} = \mathbf{0})$ *with some perturbation noise* $\boldsymbol{\epsilon} \sim \rho_{\mathbf{r}}(\boldsymbol{\epsilon})$, *then the loss of standard (i.e., risk-unaware) diffusion models at time step* $t$:

$$\mathcal{L}_{t,\mathbf{r}=\mathbf{0}} = \mathbb{E}_{\mathbf{x} \sim p_t(\mathbf{x})}[\|\mathbf{s}_{\boldsymbol{\theta}}(\mathbf{x}, t) - \nabla_{\mathbf{x}} \ln p_t(\mathbf{x})\|^2],$$

*is equal to the below new loss for non-zero risk* $\mathbf{r} \neq \mathbf{0}$:

$$\mathcal{L}_{t,\mathbf{r}} = \mathbb{E}_{\mathbf{x} \sim \widetilde{p}_{t,\mathbf{r}}(\mathbf{x})}[\|\mathbf{s}_{\boldsymbol{\theta}}(\mathbf{x}, t) - \nabla_{\mathbf{x}} \ln \widetilde{p}_{t,\mathbf{r}}(\mathbf{x})\|^2], \tag{72}$$

*if the noise distribution* $\rho_{\mathbf{r}}(\boldsymbol{\epsilon})$ *is Gaussian and the time step* $t$ *is within the stability interval* $\mathcal{T}(\mathbf{r})$. *Importantly, the alternative loss* $\mathcal{L}_{t,\mathbf{r}}$ *has another form:*

$$\mathcal{L}_{t,\mathbf{r}} = \mathcal{C}_t + \mathbb{E}_{\widetilde{\mathbf{x}}(0),\boldsymbol{\eta}}[\|\boldsymbol{\eta} / \mathbf{v}(\mathbf{r}, t) + \mathbf{s}_{\boldsymbol{\theta}}(\mathbf{u}(t) \odot \widetilde{\mathbf{x}}(0) + \mathbf{v}(\mathbf{r}, t) \odot \boldsymbol{\eta}, t)\|^2], \tag{73}$$

*where* $\widetilde{\mathbf{x}}(0) \sim \widetilde{p}_{0,\mathbf{r}}(\mathbf{x})$, *coefficients* $\mathbf{u}(t), \mathbf{v}(\mathbf{r}, t)$ *are as defined in Eq. (29),* $\boldsymbol{\eta} \sim \mathcal{N}(\mathbf{0}, \mathbf{I})$, *and* $\mathcal{C}_t$ *is a constant that does not contain the parameter* $\boldsymbol{\theta}$.

*Proof.* Based on Theorem E.1 and Lemma G.1, we know that the stability condition $\widetilde{p}_{t,\mathbf{r}}(\mathbf{x}) = p_t(\mathbf{x})$ is achieved in this setup. Therefore, we can derive

$$\mathcal{L}_{t,\mathbf{0}} = \mathbb{E}[\|\mathbf{s}_{\boldsymbol{\theta}}(\mathbf{x}, t) - \nabla_{\mathbf{x}} \ln p_t(\mathbf{x})\|_2^2] = \mathbb{E}[\|\mathbf{s}_{\boldsymbol{\theta}}(\mathbf{x}, t) - \nabla_{\mathbf{x}} \ln \widetilde{p}_{t,\mathbf{r}}(\mathbf{x})\|_2^2] = \mathcal{L}_{t,\mathbf{r}}. \tag{74}$$

Therefore, the first claim of this theorem is proved.

Secondly, we aim to derive another form of the loss $\mathcal{L}_{t,\mathbf{r}}$ to make it computationally feasible. To begin with, by expanding the definition of $\mathcal{L}_{t,\mathbf{r}}$, we have

$$\begin{aligned}
\mathcal{L}_{t,\mathbf{r}} &= \mathbb{E}_{\widetilde{\mathbf{x}}(t) \sim \widetilde{p}_{t,\mathbf{r}}(\widetilde{\mathbf{x}}(t))}\left[\|\mathbf{s}_{\boldsymbol{\theta}}(\widetilde{\mathbf{x}}(t), t) - \nabla_{\widetilde{\mathbf{x}}(t)} \ln \widetilde{p}_{t,\mathbf{r}}(\widetilde{\mathbf{x}}(t))\|_2^2\right] \\
&= \mathbb{E}\left[\|\mathbf{s}_{\boldsymbol{\theta}}(\widetilde{\mathbf{x}}(t), t)\|_2^2 + \|\nabla_{\widetilde{\mathbf{x}}(t)} \ln \widetilde{p}_{t,\mathbf{r}}(\widetilde{\mathbf{x}}(t))\|_2^2 - 2\mathbf{s}_{\boldsymbol{\theta}}(\widetilde{\mathbf{x}}(t), t)^\top \nabla_{\widetilde{\mathbf{x}}(t)} \ln \widetilde{p}_{t,\mathbf{r}}(\widetilde{\mathbf{x}}(t))\right].
\end{aligned} \tag{75}$$

Considering the following transformations:

$$\nabla_{\widetilde{\mathbf{x}}(t)} \ln \widetilde{p}_{t,\mathbf{r}}(\widetilde{\mathbf{x}}(t)) = \frac{\nabla_{\widetilde{\mathbf{x}}(t)} \widetilde{p}_{t,\mathbf{r}}(\widetilde{\mathbf{x}}(t))}{\widetilde{p}_{t,\mathbf{r}}(\widetilde{\mathbf{x}}(t))} = \frac{\nabla_{\widetilde{\mathbf{x}}(t)} \int \widetilde{p}_{0,t,\mathbf{r}}(\widetilde{\mathbf{x}}(0), \widetilde{\mathbf{x}}(t)) d\widetilde{\mathbf{x}}(0)}{\widetilde{p}_{t,\mathbf{r}}(\widetilde{\mathbf{x}}(t))}$$

$$= \frac{\nabla_{\widetilde{\mathbf{x}}(t)} \int \widetilde{p}_{0,\mathbf{r}}(\widetilde{\mathbf{x}}(0)) \widetilde{p}_{t|0,\mathbf{r}}(\widetilde{\mathbf{x}}(t) \mid \widetilde{\mathbf{x}}(0)) d\widetilde{\mathbf{x}}(0)}{\widetilde{p}_{t,\mathbf{r}}(\widetilde{\mathbf{x}}(t))}$$

$$= \frac{\int \widetilde{p}_{0,t,\mathbf{r}}(\widetilde{\mathbf{x}}(0), \widetilde{\mathbf{x}}(t)) \nabla_{\widetilde{\mathbf{x}}(t)} \ln \widetilde{p}_{t|0,\mathbf{r}}(\widetilde{\mathbf{x}}(t) \mid \widetilde{\mathbf{x}}(0)) d\widetilde{\mathbf{x}}(0)}{\widetilde{p}_{t,\mathbf{r}}(\widetilde{\mathbf{x}}(t))} \qquad (76)$$

$$= \int \widetilde{p}_{0|t,\mathbf{r}}(\widetilde{\mathbf{x}}(0) \mid \widetilde{\mathbf{x}}(t)) \nabla_{\widetilde{\mathbf{x}}(t)} \ln \widetilde{p}_{t|0,\mathbf{r}}(\widetilde{\mathbf{x}}(t) \mid \widetilde{\mathbf{x}}(0)) d\widetilde{\mathbf{x}}(0)$$

$$= \mathbb{E}_{\widetilde{\mathbf{x}}(0) \sim \widetilde{p}_{0|t,\mathbf{r}}(\widetilde{\mathbf{x}}(0)|\widetilde{\mathbf{x}}(t))} [\nabla_{\widetilde{\mathbf{x}}(t)} \ln \widetilde{p}_{t|0,\mathbf{r}}(\widetilde{\mathbf{x}}(t) \mid \widetilde{\mathbf{x}}(0))].$$

Combining the above two equations, we have

$$\mathcal{L}_{t,\mathbf{r}} = \mathbb{E}_{\widetilde{\mathbf{x}}(t)} \Big[ \|\mathbf{s}_{\boldsymbol{\theta}}(\cdot)\|_2^2 + \|\nabla_{\widetilde{\mathbf{x}}(t)} \ln \widetilde{p}_{t,\mathbf{r}}(\widetilde{\mathbf{x}}(t))\|_2^2 - 2\mathbf{s}_{\boldsymbol{\theta}}(\widetilde{\mathbf{x}}(t), t)^{\top} \mathbb{E}_{\widetilde{\mathbf{x}}(0)} [\nabla_{\widetilde{\mathbf{x}}(t)} \ln \widetilde{p}_{t|0,\mathbf{r}}(\widetilde{\mathbf{x}}(t) \mid \widetilde{\mathbf{x}}(0))] \Big]$$

$$= \mathbb{E}_{\widetilde{\mathbf{x}}(0), \widetilde{\mathbf{x}}(t)} \Big[ \|\mathbf{s}_{\boldsymbol{\theta}}(\cdot)\|_2^2 + \|\nabla_{\widetilde{\mathbf{x}}(t)} \ln \widetilde{p}_{t|0,\mathbf{r}}(\cdot)\|_2^2 - 2\mathbf{s}_{\boldsymbol{\theta}}(\widetilde{\mathbf{x}}(t), t)^{\top} \nabla_{\widetilde{\mathbf{x}}(t)} \ln \widetilde{p}_{t|0,\mathbf{r}}(\widetilde{\mathbf{x}}(t) \mid \widetilde{\mathbf{x}}(0)) \Big] + \mathcal{C}_t$$

$$= \mathbb{E}_{\widetilde{\mathbf{x}}(0) \sim \widetilde{p}_{0,\mathbf{r}}(\widetilde{\mathbf{x}}(0)), \widetilde{\mathbf{x}}(t) \sim \widetilde{p}_{t|0,\mathbf{r}}(\widetilde{\mathbf{x}}(t)|\widetilde{\mathbf{x}}(0))} \Big[ \|\mathbf{s}_{\boldsymbol{\theta}}(\widetilde{\mathbf{x}}(t), t) - \nabla_{\widetilde{\mathbf{x}}(t)} \ln \widetilde{p}_{t|0,\mathbf{r}}(\widetilde{\mathbf{x}}(t) \mid \widetilde{\mathbf{x}}(0))\|_2^2 \Big] + \mathcal{C}_t,$$
$$(77)$$

where $\mathcal{C}_t$ is a constant that does not contain parameter $\boldsymbol{\theta}$:

$$\mathcal{C}_t = \mathbb{E}_{\widetilde{\mathbf{x}}(0), \widetilde{\mathbf{x}}(t)} \Big[ \|\nabla_{\widetilde{\mathbf{x}}(t)} \ln \widetilde{p}_{t,\mathbf{r}}(\widetilde{\mathbf{x}}(t))\|_2^2 - \|\nabla_{\widetilde{\mathbf{x}}(t)} \ln \widetilde{p}_{t|0,\mathbf{r}}(\widetilde{\mathbf{x}}(t) \mid \widetilde{\mathbf{x}}(0))\|_2^2 \Big]. \qquad (78)$$

Finally, we only have to simplify the derivative term:

$$\nabla_{\widetilde{\mathbf{x}}(t)} \ln \widetilde{p}_{t|0,\mathbf{r}}(\widetilde{\mathbf{x}}(t) \mid \widetilde{\mathbf{x}}(0)) = \nabla_{\widetilde{\mathbf{x}}(t)} \Big( \ln \mathcal{N}(\widetilde{\mathbf{x}}(t); \bar{\mathbf{f}}(\mathbf{r}, t) \odot \widetilde{\mathbf{x}}(0), \mathrm{diag}(\bar{\mathbf{g}}(\mathbf{r}, t)^2)) \Big)$$

$$= \nabla_{\widetilde{\mathbf{x}}(t)} \Big( -\frac{D}{2} \ln(2\pi) - \sum_{j=1}^{D} \ln \bar{g}_j(\mathbf{r}, t) - \frac{1}{2} \Big\langle (\widetilde{\mathbf{x}}(t) - \bar{\mathbf{f}}(\mathbf{r}, t) \odot \widetilde{\mathbf{x}}(0))^2, \bar{\mathbf{g}}(\mathbf{r}, t)^{-2} \Big\rangle \Big) \qquad (79)$$

$$= \frac{\bar{\mathbf{f}}(\mathbf{r}, t) \odot \widetilde{\mathbf{x}}(0) - \widetilde{\mathbf{x}}(t)}{\bar{\mathbf{g}}(\mathbf{r}, t)^2}.$$

To conclude, the risk-free loss has a simplified form as

$$\mathcal{L}_{t,\mathbf{r}} = \mathbb{E}_{\widetilde{\mathbf{x}}(0) \sim \widetilde{p}_{0,\mathbf{r}}(\widetilde{\mathbf{x}}(0)), \widetilde{\mathbf{x}}(t) \sim \widetilde{p}_{t|0,\mathbf{r}}(\widetilde{\mathbf{x}}(t)|\widetilde{\mathbf{x}}(0))} \Big[ \Big\| \mathbf{s}_{\boldsymbol{\theta}}(\widetilde{\mathbf{x}}(t), t) - \frac{\bar{\mathbf{f}}(\mathbf{r}, t) \odot \widetilde{\mathbf{x}}(0) - \widetilde{\mathbf{x}}(t)}{\bar{\mathbf{g}}(\mathbf{r}, t)^2} \Big\|_2^2 \Big] + \mathcal{C}_t. \quad (80)$$

Similar to DDPM (Ho et al., 2020), we can further simplify this equation by Gaussian reparameterization. With the reparameterization $\widetilde{\mathbf{x}}(t) = \bar{\mathbf{f}}(\mathbf{r}, t) \odot \widetilde{\mathbf{x}}(0) + \bar{\mathbf{g}}(\mathbf{r}, t) \odot \boldsymbol{\epsilon}, \boldsymbol{\epsilon} \sim \mathcal{N}(\mathbf{0}, \mathbf{I})$, we get

$$\mathcal{L}_{t,\mathbf{r}} = \mathbb{E}_{\widetilde{\mathbf{x}}(0) \sim \widetilde{p}_{0,\mathbf{r}}(\widetilde{\mathbf{x}}(0)), \boldsymbol{\epsilon} \sim \mathcal{N}(\mathbf{0}, \mathbf{I})} \Big[ \Big\| \mathbf{s}_{\boldsymbol{\theta}} \Big( \bar{\mathbf{f}}(\mathbf{r}, t) \odot \widetilde{\mathbf{x}}(0) + \bar{\mathbf{g}}(\mathbf{r}, t) \odot \boldsymbol{\epsilon}, t \Big) + \frac{\boldsymbol{\epsilon}}{\bar{\mathbf{g}}(\mathbf{r}, t)} \Big\|_2^2 \Big] + \mathcal{C}_t, \quad (81)$$

where time step $t$ belongs to stability period $\mathcal{T}$. $\qquad \square$

The expression of risk-free loss in Eq. (73) permits efficient computation in practice. Importantly but as anticipated, for the case of zero risk $\mathbf{r} = 0$, the term reduces to the loss function of ordinary risk-unaware diffusion models: $\mathcal{L}_{t,\mathbf{0}}$, for clean sample $\mathbf{x}(0)$.

**Optimization and sampling.** We respectively show the optimization and sampling procedures in Algorithm 3 and Algorithm 4. We also highlight in blue the terms that differ from vanilla diffusion models. For the optimization algorithm, when the risk is $\mathbf{0}$, the algorithm reduces to the optimization procedure of a vanilla diffusion model, with a trivial stability interval of $\mathcal{T}(\mathbf{r}) = [0, T]$. When the risk is non-zero, the risk-sensitive coefficient $\mathbf{v}(\mathbf{r}, t)$ and interval $\mathcal{T}(\mathbf{r})$ will guarantee that $\nabla_{\mathbf{x}} \ln p_t(\mathbf{x}) = \nabla_{\mathbf{x}} \ln \widetilde{p}_{t,\mathbf{r}}(\mathbf{x})$ for $t \in \mathcal{T}(\mathbf{r})$, such that the noisy sample $(\widetilde{\mathbf{x}}(0), \mathbf{r} \neq \mathbf{0})$ can be used to safely optimize the score-based model $\mathbf{s}_{\boldsymbol{\theta}}(\mathbf{x}, t)$.

---

**Algorithm 3** Optimization Algorithm

---

1: **repeat**
2:   Sample $(\widetilde{\mathbf{x}}(0),\ \mathbf{r}\ )$ from the dataset
3:   Sample time step $t$ from *stability interval* $\mathcal{T}(\mathbf{r})$
4:   $\widetilde{\mathbf{x}}(t) = \mathbf{u}(t) \odot \widetilde{\mathbf{x}}(0) + \mathbf{v}(\mathbf{r}, t) \odot \boldsymbol{\eta}\ , \boldsymbol{\eta} \sim \mathcal{N}(\mathbf{0}, \mathbf{I})$
5:   Update $\boldsymbol{\theta}$ with $-\nabla_{\boldsymbol{\theta}} \| \boldsymbol{\eta}\ /\ \mathbf{v}(\mathbf{r}, \mathbf{t})\ + \mathbf{s}_{\boldsymbol{\theta}}(\widetilde{\mathbf{x}}(t), t)\|^2$
6: **until** converged

---

**Algorithm 4** Sampling Algorithm

---

1: Set time points $\{t_M = T, t_{M-1}, \cdots, t_2, t_1 = 0\}$
2: Set zero risk $\mathbf{r} = \mathbf{0}$
3: $\mathbf{x}(t_M) \sim p_T(\mathbf{x}) \approx \mathcal{N}(\mathbf{x}; \mathbf{0}, \mathbf{I})$
4: **for** $i = M, M-1, \ldots, 2$ **do**
5:   $\widehat{\mathbf{b}}(\mathbf{x}(t_i), t_i) = \mathbf{f}(\mathbf{r}, t_i)\ \odot \mathbf{x}(t_i) - \mathbf{g}(\mathbf{r}, t_i)^2 \odot \mathbf{s}_{\boldsymbol{\theta}}(\mathbf{x}(t_i), t_i)$
6:   $\boldsymbol{\eta} \sim \mathcal{N}(\mathbf{0}, (t_i - t_{i-1})\mathbf{I})$
7:   $\mathbf{x}(t_{i-1}) = \mathbf{x}(t_i) - \widehat{\mathbf{b}}(\mathbf{x}(t_i), t_i)(t_i - t_{i-1}) - \mathbf{g}(\mathbf{r}, t_i) \odot \boldsymbol{\eta}$
8: **end for**
9: **return** $\mathbf{x}(0)$

---

For the sampling algorithm, by setting zero risk $\mathbf{r} = \mathbf{0}$, the coefficients $\mathbf{f}(\mathbf{r}, t), \mathbf{g}(\mathbf{r}, t)$ become compatible with the model $\mathbf{s}_{\boldsymbol{\theta}}(\mathbf{x}, t)$ and together generate high-quality sample $\mathbf{x}(0)$. Our model will generate only clean samples $(\mathbf{x}(0), \mathbf{r} = \mathbf{0})$, but it was already able to capture the rich distribution information contained in noisy sample $(\widetilde{\mathbf{x}}(0), \mathbf{r} \neq \mathbf{0})$ during optimization.

### G.3   EXTENSION UNDER CAUCHY PERTURBATION

In terms of Theorem F.1, we provide a corollary that applies *risk-sensitive diffusion* to VE SDE, letting it be robust to Cauchy-corrupted samples.

**Corollary G.3** (Risk-sensitive VE SDE for Cauchy Noises)**.** *For the weight function $\Omega(\mathbf{y}) = |\boldsymbol{\chi}_t(\mathbf{y})|^2$ and a Cauchy noise distribution $\rho_{\mathbf{r}}(\boldsymbol{\epsilon})$ that is specified by scales $\mathbf{r}$ (i.e., risk vector):*

$$\rho_{\mathbf{r}}(\boldsymbol{\epsilon}) = \prod_{j=1}^{D} \left( \pi r_j \left( 1 + \frac{\epsilon_j^2}{r_j^2} \right) \right)^{-1},$$

*the minimally-unstable risk-sensitive SDE (defined by Eq. (5)) for VE SDE has a drift coefficient as $\mathbf{f}(\mathbf{r}, t) = \mathbf{0}$ and a diffusion coefficient as*

$$\mathbf{g}(\mathbf{r}, t) = \mathbb{1}(\sigma(t)^2 \mathbf{1} \gtrsim \sigma(0)^2 \mathbf{1} + \boldsymbol{\psi}(\mathbf{r})) \sqrt{\frac{d\sigma(t)^2}{dt}},$$

*where the term $\boldsymbol{\psi}(\mathbf{r})$ is defined as*

$$\boldsymbol{\psi}(\mathbf{r}) = \left( \mathbf{r}^{-2}(\mathbf{r}^{-2})^{\top} + \mathrm{diag}(5\mathbf{r}^{-4}) \right)^{-1} \left( \mathbf{r}^{-1}\mathbf{1}^{\top}\mathbf{r}^{-1} + 2\mathbf{r}^{-2} \right).$$

*The vector $\mathbf{r}$ is element-wise non-negative. For $\mathbf{r} = \mathbf{0}$, the risk-sensitive SDE reduces to VE SDE, with $\mathbf{f}(\mathbf{0}, t) = \mathbf{0}, \mathbf{g}(\mathbf{0}, t) = \sqrt{d\sigma(t)^2/dt}$.*

*Proof.* Based on Theorem F.1, we aim to derive risk-sensitive SDEs for the noises sampled from a multivariate Cauchy distribution. We first suppose a noise $\boldsymbol{\epsilon} = [\epsilon_1, \epsilon_2, \cdots, \epsilon_D]^{\top}$, with every dimension $j \in [1, D]$ being independent and following a univariate Cauchy distribution $\rho_j(\epsilon_j)$ that is parameterized by scale $\kappa_j$:

$$\rho_j(\epsilon_j) = \frac{1}{\pi \kappa_j (1 + \epsilon_j^2/\kappa_j^2)}, \quad \phi_j(\omega_j) = \exp\left( -\kappa_j |\omega_j| \right), \tag{82}$$

where $\phi_j(\omega_j)$ is the characteristic function of distribution $\rho_j(\epsilon_j)$. Then, because random variables $\epsilon_1, \epsilon_2, \cdots, \epsilon_D$ are mutually independent, their joint distribution $\rho(\boldsymbol{\epsilon})$ is equal to $\prod_{j=1}^{D} \rho_j(\epsilon_j)$ and we can derive its characteristic function $\phi(\boldsymbol{\omega})$ as

$$
\begin{aligned}
\phi(\boldsymbol{\omega}) &= \mathbb{E}_{\boldsymbol{\epsilon} \sim \rho(\boldsymbol{\epsilon})}[\exp(i\boldsymbol{\epsilon}^\top \boldsymbol{\omega})] = \int \rho(\boldsymbol{\epsilon}) \exp(i\boldsymbol{\epsilon}^\top \boldsymbol{\omega}) d\boldsymbol{\epsilon} \\
&= \prod_{j=1}^{D} \left( \int \rho_j(\epsilon_j) \exp(i\epsilon_j \omega_j) d\omega_j \right) = \exp\left( -\boldsymbol{\kappa}^\top |\boldsymbol{\omega}| \right),
\end{aligned}
\tag{83}
$$

where $\boldsymbol{\kappa} = [\kappa_1, \kappa_2, \cdots, \kappa_D]$. Now, we can convert Eq. (66) into the following form:

$$
\begin{aligned}
&\left[ \int \Gamma(\boldsymbol{\omega}) \omega_i^2 \omega_j^2 d\boldsymbol{\omega} \right]_{i,j\in[1,D]} \left( \bar{\mathbf{g}}(\mathbf{0},t)^2 - \bar{\mathbf{g}}(\mathbf{r},t)^2 \right) = -2 \left[ \int \Gamma(\boldsymbol{\omega}) \ln \left| \phi(\bar{\mathbf{f}}(t) \odot \boldsymbol{\omega}) \right| \omega_i^2 d\boldsymbol{\omega} \right]_{i\in[1,D]}^\top \\
&= 2 \left[ \int \Gamma(\boldsymbol{\omega}) \left( \bar{\mathbf{f}}(t) \odot \boldsymbol{\kappa} \right)^\top |\boldsymbol{\omega}| \omega_i^2 d\boldsymbol{\omega} \right]_{i\in[1,D]}^\top = 2 \left[ \int \Gamma(\boldsymbol{\omega}) |\omega_i| \omega_j^2 d\boldsymbol{\omega} \right]_{i,j\in[1,D]} \left( \bar{\mathbf{f}}(t) \odot \boldsymbol{\kappa} \right).
\end{aligned}
\tag{84}
$$

We set the weight function $\Gamma(\boldsymbol{\omega})$ as the magnitude of the characteristic function $|\phi(\boldsymbol{\omega})|$ since it indicates the importance of value $\boldsymbol{\omega}$. In this regard, consider the below five integrals:

$$
\begin{aligned}
&\int \exp(-\kappa_j|\omega_j|) d\omega_j = \frac{2}{\kappa_j}, \quad \int \exp(-\kappa_j|\omega_j|) w_j^2 d\omega_j = \frac{4}{\kappa_j^3}, \quad \int \exp(-\kappa_j|\omega_j|) |\omega_j| d\omega_j = \frac{2}{\kappa_j^2}, \\
&\int \exp(-\kappa_j|\omega_j|) w_j^4 d\omega_j = \frac{48}{\kappa_j^5}, \quad \int \exp(-\kappa_j|\omega_j|) w_j^2 |w_j| d\omega_j = \frac{12}{\kappa_j^4}.
\end{aligned}
\tag{85}
$$

we can solve that linear equation as

$$
\bar{\mathbf{g}}(\mathbf{0},t)^2 - \bar{\mathbf{g}}(\mathbf{r},t)^2 = \left[ \frac{1 + 5\mathbb{1}(i=j)}{\kappa_i^2 \kappa_j^2} \right]_{i,j\in[1,D]}^{-1} \left[ \frac{1 + 2\mathbb{1}(i=j)}{\kappa_i \kappa_j^2} \right]_{i,j\in[1,D]} \left( \bar{\mathbf{f}}(t) \odot \boldsymbol{\kappa} \right).
\tag{86}
$$

For demonstration purposes, let's consider the case of $D = 1$:

$$
\bar{g}(r,t)^2 = \bar{g}(0,t)^2 - \frac{1}{2} \bar{f}(t) \kappa^2.
\tag{87}
$$

For $D > 1$, we can only have the below form with an inverse matrix:

$$
\bar{\mathbf{g}}(\mathbf{r},t)^2 = \bar{\mathbf{g}}(\mathbf{0},t)^2 - \bar{\mathbf{f}}(t) \odot \left( \left( \boldsymbol{\kappa}^{-2} (\boldsymbol{\kappa}^{-2})^\top + \text{diag}(5\boldsymbol{\kappa}^{-4}) \right)^{-1} \left( \boldsymbol{\kappa}^{-1} \mathbf{1}^\top \boldsymbol{\kappa}^{-1} + 2\boldsymbol{\kappa}^{-2} \right) \right),
\tag{88}
$$

where $\boldsymbol{\kappa}^{-n} = [\kappa_1^{-n}, \kappa_2^{-n}, \cdots, \kappa_D^{-n}], n \in \mathcal{N}^+$. $\qquad \square$

For risk-sensitive VE SDE under Cauchy Perturbation, the concept of stability interval does not apply, though its coefficient $\mathbf{g}(\mathbf{r},t)$ is optimal in a sense that the instability measure $\mathcal{S}_t(\mathbf{r})$ is minimized. For optimization, one can simply set $\mathcal{T}(\mathbf{r}) = [0,T]$ and apply Algorithm 3. In Appendix 5, our numerical experiments show that the risk-sensitive VE SDE are very robust for optimization with Cauchy-corrupted samples (i.e., Fig. 4).

