# OpenReview forum: "Risk-Sensitive Diffusion: Robustly Optimizing Diffusion Models with Noisy Samples"
_ICLR.cc/2025/Conference — ICLR 2025 Poster_

### Official Review · Reviewer_YG5L · 2024-11-02

**Soundness:** 3
**Presentation:** 4
**Contribution:** 3
**Rating:** 8
**Confidence:** 4

**Summary:**

When we train a diffusion model, what if not all samples from the training dataset were made equal? Some contain no noise - risk of zero - and some are noisy and come with so-called risk vectors that indicate how corrupted they are. The paper proposes modifications to the standard diffusion training and sampling algorithms that account for this noise in the training data. Samples generated by the resulting model are [almost] the same as if the model were trained on entirely clean/uncorrupted data. The authors show that their modifications are grounded in theory and are expected to work in many realistic scenarios.

**Strengths:**

I buy into the concept, I think that this is an interesting twist on the original idea. Even though the changes to the diffusion algorithms are rather minor, the theory section persuades me that this is indeed the right modification in the presence of noise. The toy examples are equally persuasive - it's pretty clear that the proposal works.

**Weaknesses:**

I'm less impressed with the empirical results, as they too seem rather artificial and closer in spirit to the synthetic data than to a real-world problem being solved.

**Questions:**

I don't have questions - everything in the paper is pretty clear. I suggest improving empirical results on the real-world data. A low-hanging fruit would be promoting the vision results from the Appendix to the main body: this methods seems very general to me and needs not be confined to just the time series or tabular data setting. Generally, authors should think about/look for settings where this risk/corruption is front and center. Perhaps from vision and video domains. Or from finance - you mention applicability, but don't attempt to solve any finance-flavored problems. Perhaps volatility can proxy for noise/risk. Right now, this paper is more like a [rather elegant] solution looking for a problem it can solve.

---

> ### Author Response · Authors · 2024-11-23
> **Rebuttal**
>
> We thank the reviewer for such kind and insightful feedback.
>
> ## Part-1: More Risk-centric Application: Handling Fuzzy Training Images.
> As you recommended, we found a risk-centric problem in the image domain: fuzzy training images, which is mostly neglected by practitioners and lacks an elegant solution at present. We conduct the following systematic study, showing that risk-sensitive diffusion can effectively handle this problem.
>
> ### **Problem**
> Many image datasets contain fuzzy training images that are resized from a smaller size. Typical examples are ImageNet and CelebA: their images are with very diverse shapes, so practitioners need to first resize them to a fixed shape before training diffusion models, though some originally low-resolution images will get burry in this process, resulting in low-quality training samples. Besides just ignoring this problem, an obvious way to get around is to exclude such blurry images, though this means losing the valuable information they contain.
>
> ### **Approach with Risk-sensitive Diffusion**
> Suppose that the required shape is $H \times W$ and an image sample x(0) is sized as $H’ \times W’$, then we define the risk vector r in that case as a vector full of $\sqrt{\max(H / H’ - 1, 0) \max(W / W’ - 1, 0)}$. This expression can be interpreted as below:
> - If the original image $x(0)$ is larger than expected in both height H and width W, then the resized image $\tilde{x}(0)$ remains still clear and the risk vector is full of zero;
> - If the image is smaller than expected in either height or width, then the resized one will be fuzzy, incurring a non-zero risk vector.
>
> With pairs of fuzzy image $\tilde{x}(0)$ and risk vector r, we apply the Gaussian variant of risk-sensitive diffusion (i.e., Theorem 3.1) to robustly train a vanilla diffusion model (i.e., VP SDE), forming the Risk-sensitive VP SDE (i.e., Corollary 3.1). It is for sure that the potential noise perturbation in this setting is not Gaussian, but we will see that risk-sensitive can still effectively handle the fuzziness, similar to the cases of our experiments in Sec. 5: the Gaussian assumption empirically works well.
>
> ### **Experiment Setup**
> ImageNet would be an ideal benchmark due to the diverse shapes of its images. Given the limited time, we simulate fuzziness on CIFAR-10 by resizing a percentage of the training images (32 x 32) to smaller dimensions, chosen uniformly from the range [10, 20] x [10, 20], and then mapping them back to the original size. The selected percentages are 5%, 10%, and 20%, resulting in three simulated benchmarks. Following standard practice, we adopt FID as the evaluation metric, where a lower value indicates better performance.
>
> ### **Results**
> The experiment results are in the following table.
>
> | Method  | 5% fuzziness | 10% fuzziness | 20% fuzziness |
> |----|---|---|----|
> | Standard VP SDE     | 4.07 | 4.58    | 5.22   |
> | Standard VP SDE w/ Discarding Fuzzy Images  | 3.92  | 4.51   | 5.42   |
> | Risk Conditional VP SDE  | 3.82  | 4.25 | 5.01   |
> | Our Model: Risk-sensitive VP SDE  | **3.58** | **3.79** | **4.15**    |
>
> We can see that the baseline: simply excluding fuzzy images, fails to scale effectively to high levels of fuzziness, as it ignores much valuable information from the discarded images. In contrast, risk-sensitive diffusion models robustly exploit the rich information contained in fuzzy images, significantly outperforming both baselines.
>
> ### **Significance**
> We believe that this study is significant in the following aspects:
> - Highlighting that risk-sensitive SDE can robustly exploit the rich information contained in fuzzy training images, further improving the training of diffusion models;
> - Diffusion-based image foundation models are highly popular now, though relying on vast amounts of training data. Risk-sensitive diffusion could potentially be used to expand the scope of their training datasets;
> - Similar to this problem setup, risk-sensitive diffusion also holds promise for diffusion-based video generation, permitting training the models on videos with varying frame rates.

---

> > ### Comment · Reviewer_YG5L · 2024-11-23
> >
> > I think this is a good paper, I have raised my score.

---

> > > ### Author Response · Authors · 2024-11-24
> > >
> > > Thank you for your positive feedback, and thanks again for reviewing our paper.
> > >
> > > We wish you a pleasant weekend!

---

### Official Review · Reviewer_pAqa · 2024-11-03

**Soundness:** 2
**Presentation:** 1
**Contribution:** 3
**Rating:** 5
**Confidence:** 3

**Summary:**

This paper introduces a framework for optimizing diffusion models in the presence of noisy data by using risk-sensitive stochastic differential equations (SDEs) guided by a “risk vector” for each data sample. This risk vector quantifies sample quality, allowing the model to adapt its optimization process to noisy conditions, minimizing instability and improving robustness. The authors derive analytical solutions for risk-sensitive SDEs under both Gaussian and non-Gaussian noise and validate the framework’s effectiveness through experiments on tabular and time-series data, showing significant improvements over baseline diffusion models.

**Strengths:**

* The paper provides a comprehensive characterization of each case encountered within the risk-sensitive framework
* The proposed methodology intrinsically outperforms approaches that condition the score function solely on the risk value or vector, demonstrating better handling of noise.
* Simple, low-dimensional experiments effectively illustrate the method’s properties: stability intervals for Gaussian-corrupted data, enhanced robustness to class imbalance relative to a risk-conditional baseline, and flexibility of the framework with "minimal instability".

**Weaknesses:**

* The paper would benefit from improved mathematical precision, as some definitions and notations create ambiguity, which affects readability. The structure can feel confusing. Some theoretical claims could provide interpretative insights, which would strengthen the theoretical exposition.
* The experimental setup appears tailored to emphasize the proposed method’s strengths, which raises questions about the generalizability and fairness of the evaluation. For instance, it is likely that the proposed method includes noisy imputed data only very late in its training process, thus excluding them in practice, which would explain most of the performance gap.

**Questions:**

I am willing to reconsider my scores if my concerns are addressed.

* Lines 56-57: The claim that isotropic Gaussian noise can completely eliminate negative impacts is maybe a bit of an over-statement. Could you provide a brief explanation (the data being already noised, we juste use it after some noising time)
* Lines 93 and 119 contain redundant expressions such as “the reverse process (i.e., reverse version of the diffusion process)” and the entire “Remark 2.1.” Simplifying these would improve readability.
* Definition 3.1: The phrase “The risk information $r$ shapes as a vector” is mathematically unclear, alike the definition of the set in (4). The use of $P_{\epsilon}$ for a family of distributions that should depend on $r$ is confusing. Furthermore, on line 172, $P_r$ is used instead of $P_{\epsilon}$, while it still does not depend on $r$ (all isotropic centred Gaussian distributions?). This is frustrating to follow.
* In Theorem 3.1, it would be nice to add some interpretations for the mathematics. What can we expect the stability interval to look like? Are we sure $u(t)$ or $f(t) (= u' / u)$ does not diverge at $t=0$ or on the boundary of the stability interval? What can we say about the fact that $f(r, t)$ does not depend on $r$? You can then articulate it better with Corollary 3.1, where you can focus on explaining that, essentially, a noised sample $\Tilde{x}(0)$ can be used in the training process at the moment when the noising process would have added "equivalent" noise from a clean sample. Is this not a good interpretation?
* Line 340: For non-Gaussian noise, it would be interesting to specify any required conditions on the noise distribution for the derivation of tractable expressions.
* Experiments (Tabular Data): Given that the imputed data is noisy (lines 506-507 "the data generated in this way will be very noisy since KNN is certainly very inaccurate."), it will mostly not be used to train the model, since the stability interval will be a small set around $T$. To properly assess the technique, you would need an experiment where you train a model only on the clean samples.
* Experiments (Noisy Time Series): In the same way, we would need more information on how the interpolation is constructed, what is the kernel used for the Gaussian process, on which part of the dataset is the generative performance measured etc., so we can fully understand the performance gap. A comparison based on your customized evaluation methods, which appear designed to highlight the strengths of your approach, may not fully capture a fair performance assessment.

---

> ### Author Response · Authors · 2024-11-23
> **Rebuttal, Section 1**
>
> We thank the reviewer for such comprehensive and constructive feedback.
>
> ## Part-1: A Recap of Our Theoretical Study
>
> We would like to provide a brief yet systematic review of our theoretical study, supported by concrete examples. We believe this review will help your understanding and address related concerns.
>
> ### **Part 1.1: Basic Concepts**
>
> ***Risk Vector $\mathbf{r}$ and Noise Distribution Family $P_{\epsilon}$:*** In our problem setting, every noisy training sample $\tilde{\mathbf{x}}(0)$ originates from a clean sample $\mathbf{x}(0)$ and a certain noise distribution $\rho_{\mathbf{r}}$. **The family $P_{\epsilon}$ is a countable set that collects all such noise distribution $\rho_{\mathbf{r}}$, with the risk vector $\mathbf{r}$ serving as a set index to identify the specific noise distribution $\rho_{\mathbf{r}} \in P_{\epsilon}$** that results in noisy sample $\tilde{\mathbf{x}}(0)$.
>
> To better understand the above explanation, let us first see the below two concrete examples:
> - Suppose that there is a set of 1-dimensional training sample $\tilde{x}(0)$ corrupted by Gaussian noises, then the noise distribution family is $P_{\epsilon} = \\{ N(0, r) \mid r > 0 \\}$. A risk scalar $r$ that pairs a noisy sample $\tilde{x}(0)$ indicates that the sample is corrupted by a noise drawn from $N(0, r)$;
> - Same as above, but what if every training sample $\tilde{x}(0)$ is corrupted by a uniform noise? The noise distribution family $P_{\epsilon}$ in this case is $\\{ U\\{ b, a \\} \mid a > 0 > b \\}$. Every noisy sample $\tilde{x}(0)$ is paired with a 2-dimensional risk vector $\mathbf{r} = [a, b]^T$, identifying the type of uniform noise.
>
> While the first example is very intuitive: the risk scalar $r$ indicates our uncertainty about the data quality of sample $\tilde{x}(0)$, this is not the case in the second example: a 1-dimensional sample x corresponds to a 2-dimensional risk vector $\mathbf{r} = [a, b]^T$. **We could make the second example as interpretable as the first one by simplifying the noise distribution family $P_{\epsilon}$ to $\\{ U\\{-a, a\\} \mid a > 0 \\}$, though we realized that some complicated cases do exist**. For example, a more complex example can be constructed as follows:
> - Still with 1-dimensional sample $\tilde{x}(0)$, but now we suppose that the noise distribution is either Gaussian or uniform. The risk vector in this case shapes as $\mathbf{r} = [a, b, c]^T$, where $r = [0, 0, c]^T$ is for the Gaussian sub-family $P_{\epsilon}^{gauss} = \\{ N(0, c) \mid c > 0 \\}$ and $r = [a, b, 0]^T$ is for the uniform sub-family $P_{\epsilon}^{uniform} = \\{ U\\{ b, a \\} \mid a > 0 > b \\}$. The whole noise distribution family is as $P_{\epsilon} =P_{\epsilon}^{gauss} \cup P_{\epsilon}^{uniform}$.
>
> Therefore, **Definition 3.1 was intentionally made as broad as possible to accommodate some less straightforward cases**, though it might seem too general at first glance. We will improve that part in the final version, incorporating the above discussion and examples.
>
> ***Risk-sensitive SDE and Its Optimality:*** Risk-sensitive SDE can be regarded as an extension of vanilla diffusion process, with its drift $\mathbf{f}(\mathbf{r}, t)$ and volatility coefficients $\mathbf{g}(\mathbf{r}, t)$ depending on the risk vector $\mathbf{r}$. The core idea is to determine its optimal coefficients that minimize the negative effect (i.e., *perturbation instability* as formalized in Definition 3.3) of noisy sample $\tilde{\mathbf{x}}(0)$ on training vanilla diffusion models. *The following Part 1.2 will provide a concrete example of such optimal risk-sensitive SDE*.
>
> ### **Part 1.2: Gaussian Risk-sensitive Diffusion**
>
> Theorem 3.1 provided the optimal risk-sensitive SDE for Gaussian perturbation in the 1-dimensional case. Since the theorem might not be very straightforward, let us first understand the essential information it was trying to convey:
> 1. Given non-zero risk $r > 0$ and vanilla diffusion process $d\tilde{x}(t) = f(0, t) \tilde{x}(t) dt + g(0, t) dw$, we can always find another diffusion process $d\tilde{x}(t) = f(r, t) \tilde{x}(t) dt + g(r, t) dw$ that minimizes the *negative effect* of noisy sample $\tilde{x}(0)$. Simply speaking, the optimal coefficients $f(r, t),  f(r, t)$ can be derived from risk $r$ and zero-free coefficients $f(0, t), g(0, t)$；
> 2. For any non-zero risk $r > 0$, the optimal risk-sensitive SDE can fully eliminate the negative effect of noisy sample $\tilde{x}(0)$ at *certain* time point $t$. All such points form a set called stability interval $\Gamma(r) \subseteq [0, T]$.

---

> ### Author Response · Authors · 2024-11-23
> **Rebuttal, Section 2**
>
> Because of the continuity assumption, **the optimal coefficients $f(r, t), g(r, t)$ are bounded on the closed set $[0, T]$, and thus do not diverge at any time point**, including $t = 0$ and the boundaries of stability interval $T(r)$. Besides, it is obvious from the theorem that this set is Borel measurable, and in practice, we find that it might be formed by multiple disjoint points and continuous lines in some wild cases. While the stability interval $\Gamma(r)$ can be very complicated in shape, we also find that, **for commonly known diffusion models (e.g., VP SDE [1]), it is simply a continuous line** starting from a middle point $s > 0$ and ending at $t = T$: (s, T].
>
> ***Some Key Interpretations:*** As you recommended, we provide some interpretations that make Theorem 3.1 more understandable:
> - For non-zero risk $r > 0$, the stability interval $\Gamma(r)$ is typically a strict subset of the entire time domain $[0, T]$. An explanation for this fact is that noisy sample $\tilde{x}(0)$ inherently involves certain information loss: **While its negative impact can be eliminated through risk-sensitive SDE, the sample provides limited noiseless information, which is reflected in the length of stability interval $\Gamma(r) \subseteq [0, T]$**. In light of the above point, we can interpret the normalized length of stability interval $|\Gamma(r)| / T$ as some type of signal-to-noise ratio;
> - As the reviewer pointed out, the drift coefficient $f(r, t)$ in fact does not depend on $r$ in its optimal form. The reason for this fact is that the perturbation noise is centered at zero by convention [2]: noisy samples are unbiased in expectation, so risk-sensitive drift $f(r, t)$ remains the same for different risks, introducing no bias;
> - We agree with the viewpoint of the reviewer: risk-sensitive SDE is a different way of adding noise to the noisy sample $\tilde{x}(0)$, such that its noisier version $\tilde{x}(t)$ can achieve the same noise level as in the risk-free diffusion process. From this perspective, we can infer that **the boundary $t = 0$ is not contained in stability interval $\Gamma(r)$, as the noisy sample $\tilde{x}(0)$ cannot get noisier at the initial time**. In other words, the interval $\Gamma(r)$ is at most a continuous line that starts from a middle time $s > 0$ and ends at $t = T$.
>
> ***Concrete Example:*** We take VP SDE as a real case for better understanding. **This example is similar to Corollary 3.1, but with more concrete computational details**. A commonly adopted form of VP SDE is as
> $$d\tilde{x}(t) = -(0.05 + 9.95t) \tilde{x}(t) dt + \sqrt{0.1 + 19.9t} dw, t \in [0, T=1].$$
> The risk-free coefficients in this case are as $f(0, t) = -(0.05 + 9.95t), g(0, t) = \sqrt{0.1 + 19.9t}$. Based on Eq. (8) and Eq. (9) of Theorem 3.1, the optimal risk-sensitive SDE for this pair of risk-free coefficients and any given non-zero risk $r > 0$ shapes as
> $$d\tilde{x}(t) = -(0.05 + 9.95t) \tilde{x}(t) dt + 1(\exp(0.1t + 9.95t^2) > 1 + r^2) \sqrt{0.1 + 19.9t} dw, t \in [0, 1],$$
> where the volatility coefficient $g(r, t) = 1(\cdot) \sqrt{0.1 + 19.9t}$ is truly risk-sensitive. Importantly, We can get the stability interval as
> $$\Gamma(r) = [(-0.1 + \sqrt{0.01 + 39.8\ln(1 + r^2)}) / 19.9, 1],$$
> where the risk-sensitive SDE can fully reduce the negative impact of noisy sample $\tilde{x}(0)$.
>
> We can see that a higher risk $r$ results in a shorter interval $\Gamma(r)$. A kind reminder is that **samples with different risk levels (including $r = 0$) are equally likely to be sampled for training**, though they have various sets of available diffusion time steps for computing the score matching loss.
>
> ### **Part 1.3: Non-Gaussian Risk-sensitive Diffusion**
>
> Theorem 3.2 provided the optimal coefficients of risk-sensitive SDE for non-Gaussian perturbation. This theorem has the same spirit as Theorem 3.1, despite the following differences:
> - The negative effect of noisy sample $\tilde{\mathbf{x}}(0)$ can still be minimized by certain risk-sensitive coefficients $\mathbf{f}(\mathbf{r}, t), \mathbf{g}(\mathbf{r}, t)$, but it cannot be fully eliminated as in the Gaussian case. Therefore, the concept of stability interval $\Gamma(r)$ does not apply in the non-Gaussian case;
> - The expression of volatility coefficient $\mathbf{g}(\mathbf{r}, t)$ involves a complicated integral, which might not have a closed-form solution for all noise types. A way to get around this issue is to **numerically solve the integral, and since the solution only depends on risk $\mathbf{r}$ and time $t$, it can be pre-computed before training models: trading space for time**. While this method may not be the most elegant, we leave it as an open question for future work.
>
> This whole part aims to answer your concerns in Weakness-1 and Question-1,3,4,5,6 and we will include it in the final version.
> We welcome any further questions you might have.

---

> ### Author Response · Authors · 2024-11-23
> **Rebuttal, Section 3**
>
> ## Part-2:  Clarifying Some Experiment Details
>
> ### **Part-2.1: Tabular Experiments**
>
> As highlighted at the end of the previous "Concrete Example" paragraph, regardless of the risk level $\mathbf{r}$, both noisy sample $\tilde{\mathbf{x}}(0)$ and clean sample $\mathbf{x}(0)$ are equally likely to be sampled for training, though they have different stability interval $\Gamma(r)$ to compute the score matching loss. In other words, **even the noisiest sample is treated the same as a clean sample for batch sampling in training**, and noisy samples are involved throughout the entire process of model training, not just its final stage.
>
> Another useful fact is that, for risk-sensitive VP SDE, *the length of stability interval $|\Gamma(r)|$ slowly decreases at a logarithmic speed with increasing risk $r$*. In terms of the previous "Concrete Example", the interval length is as
> $$|\Gamma(r)| = 1 - \Big(-0.1 + \sqrt{0.01 + 39.8\ln(1 + r^2)} \Big) / 19.9.$$
> The largest risk entry in our tabular datasets is $r = 5.41$, corresponding to the interval length as $|\Gamma(5.41)| = 0.42$, which is almost half of the total length $|[0, 1]| = |1 - 0| = 1$. This finding indicates that **even the noisiest sample provides much valuable information for training diffusion models**, so again: there is no need to worry that noise sample $\mathbf{x}(0)$ might be ignored during training.
>
> As the reviewer recommended, we have conducted **a new experiment to compare our model with the baseline that excludes all noisy samples**. Below are the experiment results.
>
> | Model / Dataset  | Abalone | Telemonitoring  |
> |-----|-----|-----|
> | Standard VP SDE (All Training Samples)   | 0.925 | 9.935 |
> | Standard VP SDE (**Only Clean Samples**)     | 0.792     | 6.331 |
> | VP SDE w/ Risk Conditional (All Training Samples) | 0.585 | 3.785  |
> | Our Model: Risk-sensitive VP SDE  (All Training Samples)  | **0.077** | **1.462**  |
>
> We can see that the recommended baseline indeed incurs some performance gains because it is only trained with clean samples. However, such gains are much less significant than the risk-conditional baseline and our model, both of which two can exploit the rich information contained in noisy samples.
>
> ### **Part-2.2: Time-series Experiments**
>
> Regarding *time series interpolation*, we adopted a non-parametric variant of the Gaussian process [3] that is commonly used in data curation. Specifically, given some time series $X = \\{ (x_1, t_1), (x_2, t_2), \cdots, (x_n, t_n) \\}, t_1 < t_2 < \cdots < t_n$, there are three cases:
> - If the missing point $x_s$ at time $s < t_1$ is to the left of all known points, then it is with respect to follows $N(x_1, t_1 - s)$ because the stochastic process $\\{x_t\\}_{s \le t < t_1}$ is assumed to be a martingale;
> - If the missing point $x_s$ at time $s > t_n$ is to the right of all known points, then it follows $N(x_n, s - t_n)$ since the stochastic process $\\{x_t\\}_{t_n < t \le s}$ is also supposed to be a martingale;
> - If the missing point  $x_s$ at time $s \in (t_i, t_j), j = i + 1$ is in the middle of two adjacent known points, then it follows  $N(x_{t_i} + x_{t_j} (s - t_i) / (t_j - t_i) , (t_j - s) (s - t_i) / (t_2 - t_1))$ as the stochastic process $\\{x_t\\}_{t_i \le t \le t_j}$ is assumed to be a Brownian bridge.
>
> The missing point $x_s$ is finally imputed with a sample that is drawn from the distribution specified by the above procedure.
>
> Regarding the *performance evaluation*, we mainly adopted the Wasserstein distance that measures the distribution gap between the test set and model-generated time series. **This metric can naturally handle missing values through some masking mechanism [4]**, so we can apply the whole time-series test set, which contains missing values, to evaluate the generative models. **Most importantly, we carefully considered the diversity of evaluation metrics**, with PRD curves depicted in Fig. 5 and MMD shown in Table 2. We also adopted a widely recognized metric: FID [5], for our image experiment in Table 4 and a new application study in our Part-1 answer to Reviewer YG5L. Therefore, we believe that our experiment results with such diverse standard metrics are very convincing.
>
> This part aims to answer your concerns in Weakness-2 and Question-6,7.
> We welcome any further questions you might have.
>
> ## Part-3: Other Concerns in Question-2,3
>
> We thank the reviewer for pointing out some mistakes (e.g., redundant expressions and notation error) in our paper. We will correct them in the final version.
>
> ## References
>
> [1] Score-based Diffusion Models via Stochastic Differential Equations, ICLR-2021
>
> [2] Signal Processing and Linear Systems, Oxford University Press-1998
>
> [3] Extrapolation and Interpolation of Stationary Gaussian Processes, The Annals of Mathematical Statistics-1970
>
> [4] Time Series Diffusion in the Frequency Domain, ICML-2024
>
> [5] The Fréchet distance between multivariate normal distributions, Journal of Multivariate Analysis-1982

---

> ### Author Response · Authors · 2024-11-26
> **A Brief Guide to Our Rebuttal**
>
> Dear Reviewer pAqa,
>
> We would like to thank you again for your comprehensive and insightful feedback!
>
> *While our rebuttal was intended to both address your concerns and help you understand, we realize that its length might cause inconvenience to your re-evaluation, especially considering the large volume of submissions*.
>
> Below is a table that links your concerns and questions to our answers in the rebuttal:
>
> ---
> | **Specific Questions** | **General Concern** | **Answers in Our Rebuttal** |
> | -------------------- | ------------------------------------ | ------------------------ |
> | ***Question-1***: Explanation for the seemingly over-stated Gaussian case? | Weakness-1  | The clarification of Gaussian risk-sensitive SDE is at ***the beginning of Part 1.2***, with an interpretation in the ***1st point of paragraph "Some Key Interpretations" (Part 1.2)***. |
> | ***Question-3.1***: Unclear definitions of risk vectors $r$ and noise family $P_{\epsilon}$. | Weakness-1 | The whole ***Part 1.1***, providing more straightforward definitions, detailed explanations, and many examples. |
> | ***Question-4.1***: What does the stability interval $\Gamma(r)$ look like? | Weakness-1 | The answer is in the unnamed ***paragraph preceding a named one "Some Key Interpretations" (Part 1.2)***, with a real case in paragraph  ***"Concrete Example" (Part 1.2)***. |
> | ***Question-4.2***: Properties (e.g., divergence) of risk-sensitive coefficients $f(t, r), g(t, r)$? | Weakness-1 | The answers are in the unnamed ***paragraph preceding a named one "Some Key Interpretations" (Part 1.2)*** and the ***2nd point of that named paragraph***. |
> | ***Question-4.3***: Interpretations for Theorem 3.1 and Corollary 3.1? | Weakness-1 | The ***3rd point of paragraph "Some Key Interpretations" (Part 1.2)***, with a real case in paragraph  ***"Concrete Example" (Part 1.2)***.  |
> | ***Question-5***: Tractability of the non-Gaussian cases. | Weakness-1 | Answers are provided in ***2nd point of "Part 1.3”***. |
> | ***Question-6***: Sample fairness in tabular experiments. | Weakness-2 | The whole ***Part-2.1***, with detailed clarifications and new experiment results.  |
> | ***Question-7***: Missing point interpolation and model evaluation in time-series experiments. | Weakness-2 |  The whole ***Part-2.2***, providing detailed clarifications. |
> | ***Question-2,3.2***: Notation error, redundant expression, etc. | Weakness-1 | ***Part-2.3***: We will correct the mistakes in the final version. |
> ---
>
> Hope this table will be helpful.
>
> We sincerely look forward to hearing from you and welcome any further concerns!
>
> Best Regards,
>
> The Authors

---

> ### Author Response · Authors · 2024-11-29
> **Looking Forward to Your Feedback for Our Rebuttal**
>
> Dear Reviewer pAqa,
>
> We thank you again for your time and effort in reviewing our paper!
>
> We have provided a very detailed rebuttal addressing your concerns, along with an extra brief guide to help you understand it.
> While the discussion stage is approaching its extended deadline, we have not yet heard from you.
> ***We are sincerely looking forward to your feedback for our rebuttal***, *particularly as you mentioned that you would reconsider your rating*.
>
> We would appreciate your continued attention!
>
> Best Regards,
>
> The Authors

---

> > ### Comment · Reviewer_pAqa · 2024-12-01
> >
> > Dear authors,
> >
> > I warmly thank you for your detailed rebuttal. Your comments address most of my concerns, especially about the experimental results. The mathematical details seem convincing. I will raise my evaluation as promised, but lower my confidence score.
> >
> > Best regards

---

> > > ### Author Response · Authors · 2024-12-01
> > >
> > > Glad to know that our answers were helpful, and we really appreciate your decision to raise your rating in the end!

---

### Official Review · Reviewer_Rd4g · 2024-11-03

**Soundness:** 4
**Presentation:** 3
**Contribution:** 3
**Rating:** 6
**Confidence:** 3

**Summary:**

This paper extends diffusion models to non-image data, like tabular data, which often contains noise that degrades model performance. The authors introduce a "risk vector" paired with each sample to indicate data quality and propose a risk-sensitive stochastic differential equation (SDE) that leverages this vector to minimize the effects of noise during model optimization. With specifically chosen coefficients, the risk-sensitive SDE supports more stable diffusion model training, accommodating both Gaussian and non-Gaussian noise types.

**Strengths:**

1 This paper ntroduces the novel concept of a "risk vector" to improve the robustness of diffusion models against noisy samples. This approach uses a principled method, risk-sensitive SDE, to incorporate the risk vector and reduce the adverse effects of noise, specifically perturbation instability.

2 This paper provides analytical solutions for risk-sensitive SDE with both Gaussian and non-Gaussian noise. A key finding is that Gaussian noise can be fully mitigated, eliminating its negative impact on model performance.

3 Extensive experiments on tabular and time-series datasets demonstrate the model's effectiveness in handling noisy samples, even when noise is mis-specified or non-Gaussian, outperforming prior baseline models.

**Weaknesses:**

1. The second section suggests that noise interference can cause a bias in the neural network's estimation of the score function. A visual experiment could be added to illustrate the extent of this bias.

2. The construction of Risk-Sensitive Diffusion inherently requires extensive prior information about noise. Could it be possible to denoise the data directly instead? The article lacks relevant exploration and comparative experiments on this aspect.

3. The Risk-Sensitive Diffusion approach is limited to Gaussian and Cauchy noise. However, is this limitation applicable to most real world datasets? The article lacks an explanation regarding this point. Although the experiments in the article show that their method can be applied to unknown noise under the Gaussian assumption, there is a lack of experiments demonstrating whether the noise itself actually has Gaussian characteristics.

**Questions:**

Please see weakness.

---

> ### Author Response · Authors · 2024-11-23
> **Rebuttal**
>
> We thank the reviewer for his or her kind and constructive feedback.
>
> ## Part-1: Visualizing the Bias Introduced by Noisy Samples
>
> Our previous proof-of-concept experiment, **as shown in Fig. 2 of our paper, both visualize the negative effect of noisy samples** and depict how risk-sensitive diffusion eliminates it over time. Specifically, the upper subfigures display the marginal distributions of clean samples at $5$ diffusion time steps (i.e., $t = 0.0, 0.25, 0.5, 0.75, 1.0$), while the lower subfigures show those of noisy samples at the same time steps under risk-sensitive diffusion. We can see that noisy samples largely differ from clean samples in their marginal distributions in the beginning (i.e., t = $0.0, 0.25$), though this gap is eventually bridged by risk-sensitive diffusion as time progresses (i.e., $t = 0.5, 0.75, 1.0$). *A kind reminder: it is the difference in marginal distributions that introduces bias in score estimation*.
>
> This part aims to answer your concern in Weakness-1.
> We welcome any further questions you might have.
>
> ## Part-2: Directly Denoising the Noisy Samples
>
> To the best of our knowledge, the deconvolution formula ( https://en.wikipedia.org/wiki/Convolution ) can be used to directly denoise a noisy sample $\tilde{x}(0)$ based on the noise distribution $\rho$, but **this formula requires the density information $p(\tilde{x}(0))$**, which is typically inaccessible.
>
> ***Brainstorming:*** A more feasible way might be to train a denoising neural network from clean samples. Specifically, we first add some noise to a clean sample $x(0)$, and then condition the neural network on the new noisy sample $\tilde{x}(0)$ and the risk level $r$ to predict the original clean sample $x(0)$. With the trained denoising neural network, vanilla diffusion models can be safely trained with noisy samples as they can be first denoised before training.
>
> We performed **a new experiment on tabular data to compare our method with this baseline**. The noisy sample $\tilde{x}(0)$ in the real dataset were still simulated, but we trained the denoising neural network with all underlying clean samples to maximize its potential.  The experiment results are as below.
>
> | Model / Dataset  | Abalone | Telemonitoring  |
> |-----|-----|-----|
> | Standard VP SDE   | 0.925 | 9.935 |
> | **Standard VP SDE w/ Denoising Neural Network** | 0.631 | 4.357 |
> | VP SDE w/ Risk Conditional | 0.585 | 3.785  |
> | **Our Model: Risk-sensitive VP SDE**  | **0.077** | **1.462**  |
>
> We can see that, while the denoising neural network indeed contributes to more robust training of diffusion models with noisy samples, its performance improvements are much less significant than our method. One possible explanation for these results: the denoising neural network trained on the full clean data is still not accurate enough.
>
> This part aims to answer your concern in Weakness-2.
> We welcome any further questions you might have.
>
> ## Part-3: Applicable to most real-world datasets?
>
> ***Real Image Experiments:*** Besides tabular and time-series data, **we have also applied risk-sensitive diffusion to two real-world problems in the image domain**. One involves noisy pixels, as *previously* presented in Table 4 of Appendix B, while the other addresses fuzzy training images, which corresponds to our response in Part-1 to Reviewer YG5L. The results for the latter application are presented below.
>
> | Method  | 5% fuzziness | 10% fuzziness | 20% fuzziness |
> |----|---|---|----|
> | Standard VP SDE     | 4.07 | 4.58    | 5.22   |
> | Standard VP SDE w/ Discarding Fuzzy Images  | 3.92  | 4.51   | 5.42   |
> | Risk Conditional VP SDE  | 3.82  | 4.25 | 5.01   |
> | Our Model: Risk-sensitive VP SDE  | **3.58** | **3.79** | **4.15**    |
>
> We can see that risk-sensitive diffusion significantly outperforms the baselines in terms of the FID metric, indicating that it can robustly exploit the rich information contained in fuzzy images.
>
> ***Non-Gaussian Noise Type:***  The noisy sample $\tilde{x}(0)$ in our tabular datasets are simulated, so we can directly get the noise $\epsilon$ by a simple subtraction $\tilde{x}(0) - x(0)$. The easiest way to check whether noise $\epsilon$ is Gaussian-like is to compute its 3rd-order central moment, which should be $0$ in the Gaussian cases. Below are the results of our three tabular datasets.
>
> | Abalone | Telemonitoring  |  Mushroom  |
> |-----|-----|-----|
> | 1.21 | 2.57 | 0.98 |
>
> We can see that the noises in all datasets have significantly non-zero 3rd-order central moments, so they are not Gaussian-like.
>
> This part aims to answer your concern in Weakness-3.
> We welcome any further questions you might have.

---

> > ### Comment · Reviewer_Rd4g · 2024-11-25
> >
> > I would like to thank the authors for their comprehensive rebuttal, which addresses most of my concerns. I vote to accept this paper.

---

> > > ### Author Response · Authors · 2024-11-25
> > >
> > > Many thanks for your support, and we thank you again for your time and efforts in reviewing our paper!

---

### Official Review · Reviewer_8F9f · 2024-11-05

**Soundness:** 3
**Presentation:** 3
**Contribution:** 3
**Rating:** 6
**Confidence:** 2

**Summary:**

This paper introduces a method called risk-sensitive SDE to improve diffusion model performance on noisy non-image data, like tabular and time-series datasets, by pairing noisy samples with risk vectors. Applying the standard diffusion process on noisy samples causes a marginal distribution shift and degrades generation quality. To mitigate this, the risk-sensitive SDE aims to minimize noise impact and improve robustness by minimizing a stability measure, termed Perturbation Instability. If perturbation stability holds—which is achievable in cases where the noise distribution is Gaussian—the noisy sample will have the same distribution as that of the corresponding clean sample at some iteration t in the Stability Interval and this could be used to optimize the score-based model. When this condition does not hold, the model seeks to minimize perturbation instability. The authors provide sufficient and necessary conditions for achieving perturbation stability and conduct extensive experiments demonstrating the effectiveness of risk-sensitive SDE over baseline models in both Gaussian and non-Gaussian noise settings.

**Strengths:**

1. The paper is well-written and well-structured, addressing an important problem and clearly articulating its contributions and findings.

2.  The work provides a well-developed mathematical foundation, including solutions for both Gaussian and non-Gaussian noise, enabling a robust approach to noisy data.

3. The paper includes well-designed experiments that validate the theoretical results and demonstrate the effectiveness of the proposed approach in handling noisy samples.

**Weaknesses:**

1. The paper focuses primarily on tabular and time-series data, which may limit insights into the method's performance on other data types, especially noisy image data. Extending experiments to image datasets could demonstrate the method’s generalizability across diverse noisy data contexts.

2. The effectiveness of the proposed approach heavily relies on accurate estimations of the risk vector. If these estimations are inaccurate—either overestimating or underestimating noise levels—the model may apply inappropriate adjustments, which could hinder its ability to adequately suppress noise. This reliance on accurate risk estimation introduces a vulnerability to the approach.

3. The paper could benefit from a discussion on the limitations of the proposed method and the areas that could be improved in future work.

**Questions:**

1. As mentioned above, while this paper focuses on noisy non-image data, such as tabular data and time series, noisy image data is also a common challenge in real-world applications. How would the proposed method perform on image datasets with high levels of noise?

2. Could you provide more details on how you derive the sensitivity interval in practice and what are the challenges you might encounter in deriving this?

---

> ### Author Response · Authors · 2024-11-23
> **Rebuttal**
>
> We thank the reviewer for such kind and comprehensive feedback.
>
> ## Part-1: Image Experiments
>
> ***Our Previous Experiment on Noisy Images:*** We have previously explored applying risk-sensitive diffusion to noisy images. In Appendix B of our paper, Table 4 presents the experiment results for images with up to 40% noisy pixels.
>
> ***New Experiment on Fuzzy Training Images:*** We also have newly applied risk-sensitive diffusion to handle fuzzy training images in our Part-1 answer to Reviewer YG5L. Please refer to that answer for the experiment setup, and we present the experiment results as below.
>
> | Method  | 5% fuzziness | 10% fuzziness | 20% fuzziness |
> |----|---|---|----|
> | Standard VP SDE     | 4.07 | 4.58    | 5.22   |
> | Standard VP SDE w/ Discarding Fuzzy Images  | 3.92  | 4.51   | 5.42   |
> | Risk Conditional VP SDE  | 3.82  | 4.25 | 5.01   |
> | Our Model: Risk-sensitive VP SDE  | **3.58** | **3.79** | **4.15**    |
>
> We can see that risk-sensitive diffusion significantly outperforms the baselines in terms of the FID metric, indicating that it can robustly exploit the rich information contained in fuzzy images.
>
> This part aims to answer your concerns in Weakness-1 and Question-1.
> We welcome any further questions you might have.
>
> ## Part-2: Estimation of Risk Vectors
>
> The estimation of risk vectors or more generally: the identification of the noise type of a noisy sample, is indeed a problem with risk-sensitive diffusion. Therefore, we have previously conducted extensive experiments (i.e., Table 1, Fig. 5, Table 2, Table 3 and Table 4) **with the noise type mis-specified as Gaussian, showing that risk-sensitive diffusion can still yield promising performance**. Another example is the above new experiment on fuzzy training images, where the underlying noise distribution is almost surely non-Gaussian, but our model performs very well, with much higher performance gains than the baselines.
>
> Our current solution to this problem is similar to the spirit of Kalman Filter: while the real-world noise distribution is typically non-Gaussian, a Gaussian assumption is often not bad. This solution might not be ideal, and we will mention it as a limitation in the final version, leaving an open question for future work.
>
> This part aims to answer your concerns in Weakness-2,3.
> We welcome any further questions you might have.
>
> ## Part-3: Derivation of the Stability Interval
>
> We take a widely used diffusion model: VP SDE, as an example to show how we derive the stability interval $\Gamma(r)$ in practice. Firstly, a commonly adopted form of VP SDE is as
> $$d\tilde{x}(t) = -(0.05 + 9.95t) \tilde{x}(t) dt + \sqrt{0.1 + 19.9t} dw, t \in [0, T=1].$$
> The risk-free coefficients in this case are as $f(0, t) = -(0.05 + 9.95t), g(0, t) = \sqrt{0.1 + 19.9t}$. Then, based on Eq. (8) and Eq. (9) of Theorem 3.1, the optimal risk-sensitive SDE for this pair of risk-free coefficients and any given non-zero risk $r > 0$ shapes as
> $$d\tilde{x}(t) = -(0.05 + 9.95t) \tilde{x}(t) dt + 1(\exp(0.1t + 9.95t^2) > 1 + r^2) \sqrt{0.1 + 19.9t} dw, t \in [0, 1],$$
> where the volatility coefficient $g(r, t) = 1(\cdot) \sqrt{0.1 + 19.9t}$ is risk-sensitive. Lastly, we can get the stability interval as
> $$\Gamma(r) = [(-0.1 + \sqrt{0.01 + 39.8\ln(1 + r^2)}) / 19.9, 1],$$
> where the risk-sensitive SDE can fully reduce the negative impact of noisy sample $\tilde{x}(0)$.
>
> As shown above, Theorem 3.1 is very elegant as it provides closed-form solutions for all components. The only challenge might be to perform some pen-and-paper calculations when dealing with a different diffusion model.
>
> This part aims to answer your Question-2.
> We welcome any further questions you might have.

---

> ### Comment · Reviewer_8F9f · 2024-12-03
>
> I thank the authors for the rebuttal. I would like to keep my score and support accepting the paper.

---

> > ### Author Response · Authors · 2024-12-03
> >
> > Thank you for your positive feedback! ***We also hope that our previous rebuttal had sufficiently addressed your concerns***; *If that is the case, we would greatly appreciate any improvement in your rating*. Such continued support might more clearly reflect the quality of this paper :)

---

### Author Response · Authors · 2024-11-25
**Looking Forward to Your Feedback**

Dear Reviewers,

We greatly appreciate the thoughtful and constructive reviews received so far, including the new feedback from Reviewer YG5L.

*As the discussion stage nears its conclusion, we look forward to hearing from all other reviewers and welcome any further concerns*.

Thank you, and we wish you a pleasant day!

Best Regards,

The Authors

---

### Meta-Review · Area_Chair_iqmh · 2024-12-21

**Metareview:**

The paper introduces a novel framework for training diffusion models on noisy data by pairing samples with risk vectors indicating data quality. It proposes risk-sensitive stochastic differential equations to minimize the negative impact of noisy samples, providing theoretical guarantees and demonstrating robust performance across Gaussian and non-Gaussian noise in various datasets, outperforming baselines.

The paper was rated positively by the reviewers as a whole. The appropriateness of some of the settings and the mathematical notation and theoretical implications were noted, and the authors provided appropriate rebuttals to these points. The content of these rebuttals should be appropriately reflected in the paper.

**Additional Comments On Reviewer Discussion:**

8F9f pointed out the limitations of the type of data handled and the instability of some of the procedures of the method; Rd4g mentioned the realism of the noise setup and the need for additional validation of the scenarios in the paper; pAqa indicated the need to explain mathematical notations and theoretical implications. Overall, the evaluation is positive.

---

### Decision · Program_Chairs · 2025-01-22

Accept (Poster)